# NAA60 facilitates LRRC8A- and LRRC8D-mediated platinum drug uptake

Carmen Alexandra Widmer [1,2], Anna Moyseos [1,2], Ismar Klebic [1,3], Martina Dettwiler [1], Martín González-Fernández [1,2,4], Ewa Gogola [5], Myriam Siffert [1], Natasha Buchs [6], Sophie Braga-Lagache [6], Anne-Christine Uldry [6], Jos Jonkers [5], Manfred Heller [6] & Sven Rottenberg [1,2,3,4,5] ✉

The platinum-based drugs cis- and carboplatin, which are crucial for treating cancers with DNA repair defects, like those caused by *BRCA1/2* mutations, rely on the volume-regulated anion channel subunits LRRC8A and LRRC8D for about 50% of cellular drug uptake. Yet, the precise mechanisms of how LRRC8A and LRRC8D mediate this function are largely unknown. Here, we identify NAA60, an N-terminal acetyltransferase, which localizes to the Golgi apparatus to affect LRRC8A and LRRC8D function. Our data suggest that NAA60 acetylates the LRRC8A/D N-termini, and its loss decreases cis- and carboplatin uptake resulting in drug resistance of otherwise hypersensitive BRCA1;p53-deficient cells and tumors. Furthermore, we mimicked the absence of the neutralizing acetyl moiety that is observed after loss of NAA60 by introducing positively charged amino acids at the N-termini of LRRC8A/D, which indeed decreased cis- and carboplatin sensitivity. Our findings highlight the importance of N-terminal acetylation by NAA60 for effective platinum drug uptake, offering new insights into overcoming drug resistance.

The platinum (Pt)-based agents cis- and carboplatin are among the most frequently used anti-cancer drugs. As their main mechanism of action is causing DNA damage, they are particularly useful to treat patients with tumors that are deficient in DNA repair by homologous recombination (HR), e.g. due to inactivating mutations of the BRCA1 or BRCA2 genes [1–5]. Cis- and carboplatin are standard of care for *BRCA1/2*-mutated ovarian cancers, and clinicians use Pt drug sensitivity to infer an HR defect and apply poly(ADP-ribose) polymerase (PARP) inhibition as maintenance therapy [6,7]. Several *BRCA1/2*-mutated high-risk breast cancer patients could even be cured by intensified carboplatin-based chemotherapy [8,9]. We have previously shown that the LRRC8A and LRRC8D subunits of the volume-regulated anion channel (VRAC) mediate cis- and carboplatin uptake [10,11]. In different human and mouse cancer cells, we observed that about 50% of the cis- and carboplatin enters the cells via LRRC8A- or LRRC8D-containing VRAC. Although recent studies revealed the structure of the individual LRRC8A/D domains important for gating, substrate uptake, and substrate specificity, we only have limited knowledge about the regulation of LRRC8A/D function [12–15].

Moreover, various attempts to get structural details of the LRRC8A/D N-termini failed.

The N-terminal (Nt) acetylation is a highly abundant co- and post-translational protein modification in eukaryotes, affecting protein subcellular localization [16,17], protein stability and degradation [18,19], and protein–protein interactions [20,21]. N-terminal acetyltransferases (NATs) hold important roles in cell survival in mammalians and stress response in plants [22–25]. Aberrant Nt acetylation has also been identified in different pathologies such as neurodegenerative or genetic diseases [26–28]. NATs are often overexpressed in cancers and aid tumorigenesis and cell proliferation [29–31], suggesting them as interesting drug targets. Their function has not been linked to the emergence of drug resistance, however [31,32]. The N-termini of LRRC8 proteins have previously been shown to affect pore properties and gating of VRACs [33], but the precise mechanism of how the N-terminus preceding the first LRRC8 transmembrane domain regulates channel function has been elusive.

Here, we identified the Nt acetyltransferase 60 (NAA60/NatF) to be crucial for LRRC8A/D function. We demonstrate that NAA60 is epistatic

[1]Institute of Animal Pathology, Vetsuisse Faculty, University of Bern, Bern, Switzerland. [2]Bern Center for Precision Medicine, University of Bern, Bern, Switzerland. [3]COMPATH, Institute of Animal Pathology, Vetsuisse Faculty, University of Bern, Bern, Switzerland. [4]Cancer Therapy Resistance Cluster, Department for Bio-Medical Research (DBMR), University of Bern, Bern, Switzerland. [5]Division of Molecular Pathology, The Netherlands Cancer Institute, Amsterdam, The Netherlands. [6]Proteomics and Mass Spectrometry Core Facility, Department for Biomedical Research (DBMR), University of Bern, Bern, Switzerland. ✉e-mail: sven.rottenberg@unibe.ch

with LRRC8A/D and its loss reduces cis- and carboplatin uptake, resulting in Pt-based drug resistance. Our data show that the neutralizing Nt acetylation of the LRRC8A/D VRAC subunits regulates the substrate permeability of molecules including cisplatin, carboplatin, and blasticidin S.

## Results

### A genome-scale CRISPR/Cas9 screen identifies *Naa60* as a mediator of cisplatin sensitivity

To get new mechanistic insights into Pt drug uptake, we employed a CRISPR/Cas9-mediated screening approach. For this purpose, we transduced cells from our genetically engineered mouse model (GEMM) for hereditary *Brca2*- and *Trp53*- mutated breast cancer (K14cre;Brca2[F/F];Trp53[F/F]) (KB2P)[34,35] using the GeCKO v2 library. The cell pool was treated with either 0.2 μM or 0.25 μM cisplatin for 7 days. The surviving cell population was harvested, its genomic DNA isolated, sequenced and the data was analyzed by the *Model-based Analysis of Genome-wide CRISPR-Cas9 Knockout* (MAGeCK) tool (Fig. 1a)[36]. The results display the analysis of 3 biological replicates for each of the individual treatment conditions (Fig. 1b, c and Supplementary Data 1, 2). *Naa60* scored amongst the top 2 hits in both treatment conditions, alongside *Lrrc8a*, which we previously found to mediate cis- and carboplatin uptake[10,11]. NAA60 is a NAT that has been suggested to post-translationally modify membrane proteins, while they are passing through the Golgi[37]. Of note, NAA60/NatF-depleted cells were found to be significantly enriched ($p = 0.028$) in a CRISPR screen for suppressors of cisplatin sensitivity in the human *BRCA1*-mutant ovarian cancer cell line COV362[38]. To validate the screen results, we generated monoclonal *Naa60* knockout cells in our BRCA1;p53-deficient KB1PM5 mouse mammary tumor cell line[39] that is highly sensitive to Pt drugs (Fig. 1d–f and Supplementary Data 2). The monoclonal *Naa60* knockout cell lines harbor big deletions in exons 4 and 5 (*Naa60*_12A6, 87 coding nucleotides deletion), which encode for functional domains of the NAA60 protein. The deletions of *Naa60*_24A4 (119 coding nucleotides deleted) and *Naa60*_13C3 (197 coding nucleotides deleted), additionally lead to a frameshift. Phenotypic confirmation of the successful NAA60 depletion came from the more fragmented composition of the Golgi apparatus (Supplementary Fig. 1 and Supplementary Data 3), a feature that has been observed previously following *NAA60* knockdown[37]. Indeed, loss of *Naa60* caused resistance towards cisplatin and carboplatin (Fig. 1g, h, j, k and Supplementary Data 2). The sensitivity towards cisplatin and carboplatin could be rescued upon reintroduction of *Naa60* cDNA (Supplementary Fig. 2 and Supplementary Data 3). The rescue constructs were either N-or C-terminally tagged with HA and localized to the Golgi apparatus as expected (Supplementary Fig. 2b). After successful in vitro validation, we generated NAA60;BRCA1;p53-deficient mammary tumor organoids, which also showed resistance towards cisplatin and carboplatin, when treated with two cycles of the indicated Pt drug (Supplementary Fig. 3 and Supplementary Data 3). To study the lack of *Naa60* expression in the in vivo context, we then injected these organoids into mice to force the development of mouse mammary tumors. These mice were then treated with two cycles (day 0 + 14) of either cisplatin (6 mg/kg) or carboplatin (50 mg/kg). Mice harboring NAA60-deficient tumors survived significantly shorter than mice that had tumors derived from the non-targeting sgRNA control organoids (cisplatin $p = 0.0018$, carboplatin $p = 0.0045$) (Fig. 1i, l and Supplementary Data 2), indicating that loss of NAA60 results in reduced efficacy of the Pt-based agents cisplatin and carboplatin and therefore faster progression of the tumor followed by poor survival of the mice.

Hence, our genome-scale CRISPR/Cas9 screen identified NAA60 as a factor that contributes to the efficacy of cis- and carboplatin both in vitro and in vivo.

### Loss of NAA60 decreases Pt drug uptake and reduces the subsequent accumulation of DNA damage

To gain further insights into the mechanism by which NAA60 facilitates cis- and carboplatin sensitivity, we exposed *Naa60* knockout and non-targeting control KB1PM5 cells to a panel of different anti-cancer treatments (Supplementary Fig. 4 and Supplementary Data 3). This included another

Pt-based drug (oxaliplatin); a DNA-crosslinking agent which mainly induces interstrand crosslinks (nimustine hydrochloride); ionizing irradiation, which induces DNA damage independent of any uptake mechanisms; the microtubule-targeting drug docetaxel; the PARP inhibitor (PARPi) olaparib, which increases single-strand DNA breaks; and blasticidin S, which disrupts protein synthesis. Strikingly, we did not observe any alterations in the response of the *Naa60* knockout cells towards oxaliplatin, nimustine hydrochloride, irradiation, docetaxel, or olaparib (Supplementary Fig. 4a–e). Loss of *Naa60* only caused resistance towards one additional drug—blasticidin S (Supplementary Fig. 4f). The sensitivity towards blasticidin S was also restored by the reintroduction of *Naa60* cDNA into the monoclonal knockout cell lines (Supplementary Fig. 2f, g). Blasticidin S is known to be taken up into the cell by the VRAC[40]. The subunit Leucine-rich repeat containing protein 8D (LRRC8D), which by the latest crystal structures has been shown to widen the pore diameter of the channel to allow for the uptake and release of bigger osmolytes[14,41], seems crucial for the uptake of blasticidin S via VRAC[40]. In fact, the cross-resistance pattern of NAA60-deficient cells is identical to the one of LRRC8A- or LRRC8D-deficient cells[11]. To determine whether the resistance is caused by decreased cellular uptake of cisplatin, we measured intracellular Pt levels by cytometry-based mass spectrometry (CyTOF). Indeed, NAA60-deficient cell lines accumulated significantly less cisplatin compared to the wild-type control cells (Fig. 2a and Supplementary Data 2). This also translated into decreased levels of DNA damage, which was determined by staining for the DNA damage marker γH2AX (Fig. 2b, c and Supplementary Data 2) after cisplatin treatment. The levels of Pt drug accumulation as well as DNA damage could be restored following the reintroduction of *Naa60* cDNA (Fig. 2d–g and Supplementary Data 2).

This suggests an epistatic interaction of NAA60 with the LRRC8A and LRRC8D VRAC subunits and further hints towards a defective uptake of cis- and carboplatin as the underlying resistance mechanism.

### Investigation of LRRC8A and LRRC8D N-termini as a substrate for NAA60

In a previous Nt acetylome analysis, the Arnesen laboratory detected LRRC8A and LRRC8C N-termini to be acetylated, however, the investigated N-Acetyl transferase B (NatB) was found not to be responsible for this modification[42]. We therefore tested whether the LRRC8s were a substrate for NAA60 instead. For this purpose, we set up an in vitro acetylation assay (Supplementary Fig. 5 and Supplementary Data 3). The NAA60 enzyme was expressed and isolated from the clonal rescue cell line *Naa60*_12A6 +pOZ *Naa60*-HA C10 using anti-HA magnetic beads via the C-terminal HA tag (Supplementary Fig. 5a). Peptides containing the first seven Nt amino acids of LRRC8A (MIPVTEL), LRRC8D (MFTLAEV), and the control peptides stemming from the heterogeneous nuclear ribonucleoprotein F (hnRNP F, MLGPEGG, positive control), and the high mobility group protein A1 (HMGB1, SESSSKS, negative control) were incubated with NAA60 in a reaction mix containing acetyl-coenzyme A. Subsequently, the reactions were quenched by the addition of acetic acid. Mass spectrometry analysis of the reaction supernatants revealed that the NAA60 enzyme-incubated samples indeed contained acetylated LRRC8A and LRRC8D, as well as hnRNP F peptides, which fit into the substrate category of NAA60[37] (Supplementary Fig. 5a). The N-terminus of HMGB1 lacks the initiator methionine and therefore should not be a substrate for NAA60[37]. Accordingly, the levels of detected acetylation were lower than for the other tested peptides. However, the differences observed using this assay did not reach statistical significance.

Furthermore, we isolated C-terminally MYC-tagged LRRC8A from isogenic NAA60-expressing or *Naa60* knockout cells by pulldown using anti-MYC magnetic beads. The beads, enriched with LRRC8A-MYC were subjected to on-bead digestion and subsequent mass spectrometry analysis (Fig. 3a and Supplementary Data 2). We identified the Nt peptide in both groups (parental and *Naa60* KO), but only in the NAA60-proficient (parental) samples the Nt peptide was acetylated (Fig. 3a). For the analysis of

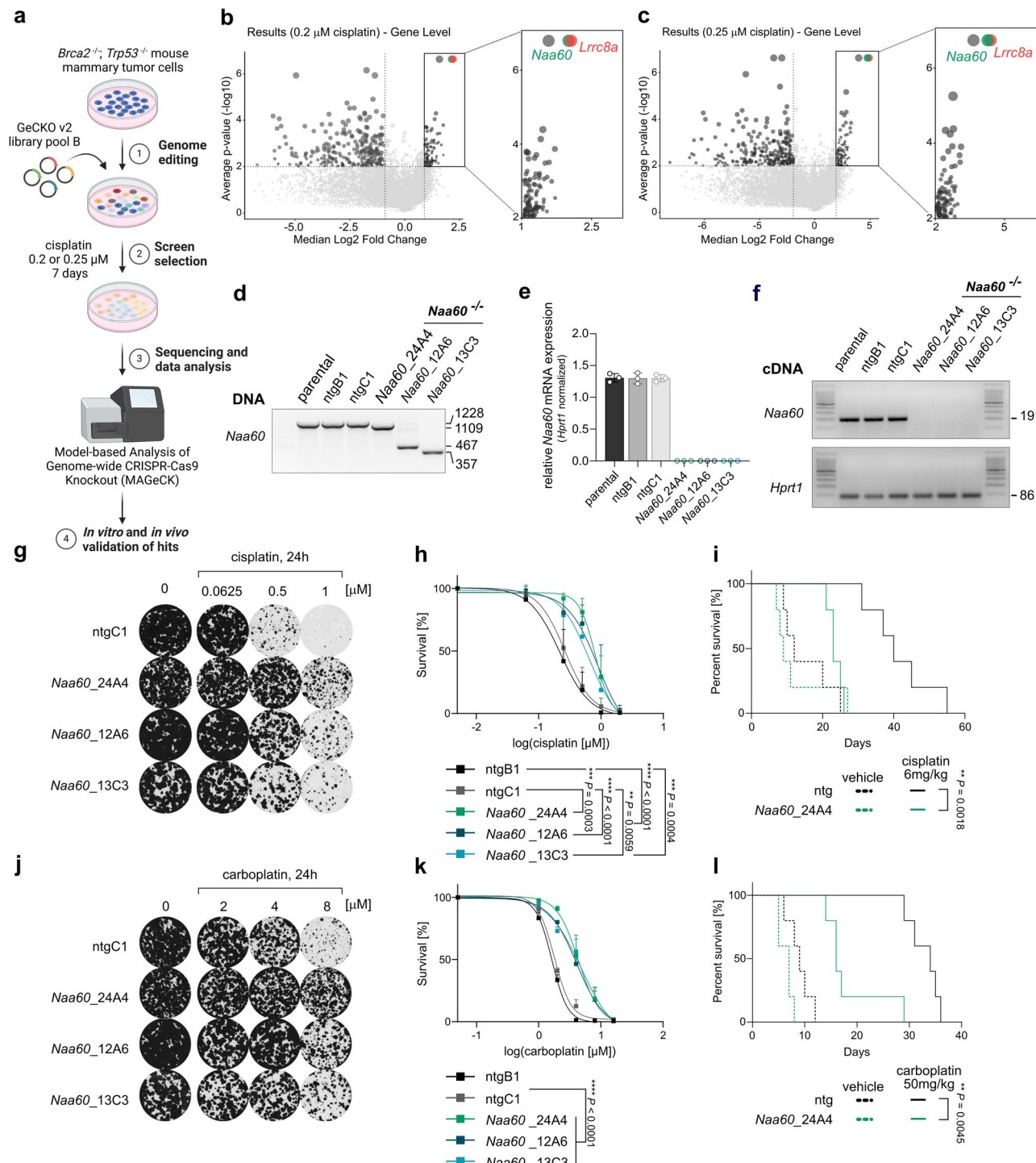

**Fig. 1 | CRISPR/Cas9 screen identifies *Naa60* as mediator of Pt drug sensitivity in BRCA1/2;p53-deficient mouse mammary tumor cells. a** CRISPR/Cas9 screen layout, where KB2P (*K14cre;Brca2*^F/F^;*Trp53*^F/F^) mouse mammary tumor cells were transduced (1) using the GeCKo V2 pool B library and subsequently selected using 0.2 μM or 0.25 μM of cisplatin over the course of 7 days (2). gDNA was harvested, sequenced and subjected to enrichment analysis using MAGeCK (3). The top hits were verified using CRISPR/Cas9 mediated gene knockout (4). Created in BioRender. Widmer (2025) https://BioRender.com/ynvrayk. **b, c** Volcano plots of the results from screens performed with either 0.2 μM or 0.25 μM cisplatin over the course of 7 days. Size of the points was scaled to −log₁₀ of FDR. **d** Representative big gene deletion knockout PCR control of wild-type (ntgB1 and ntgC1) and monoclonal *Naa60* knockout KB1PM5 cell lines (*Naa60*_24A4, *Naa60*_12A6, and *Naa60*_13C3) used in the validation experiments. **e, f** RTqPCR and gel electrophoresis of the resulting product of control and *Naa60* knockout lines.

**g, j** Clonogenic survival assays of KB1PM5 wild-type and NAA60-deficient cell lines treated with cisplatin or carboplatin for 24 h with the indicated drug concentrations. Representative images of selected lines and concentrations are shown.
**h, k** Quantification of clonogenic growth assays in the presence of cisplatin or carboplatin. Data represent mean ± SD of five independent replicates for cisplatin and three independent replicates for carboplatin and were fitted to a non-linear regression dose-response curve (log(inhibitor) vs. normalized response -Variable slope). *p* values are calculated by one-way ANOVA followed by Tukey's multiple comparisons test for the log(IC50) values of the survival curves. Kaplan–Meyer overall survival curves of mice transplanted with NAA60-deficient (*Naa60*_24A4) or wild-type control (ntg) KB1PM5 organoids treated with either vehicle or 6 mg/kg of cisplatin for (**i**) and 50 mg/kg of carboplatin for (**l**). *N* = 5 mice per group. Statistical analysis was performed with the log-rank test (Mantel–Cox).

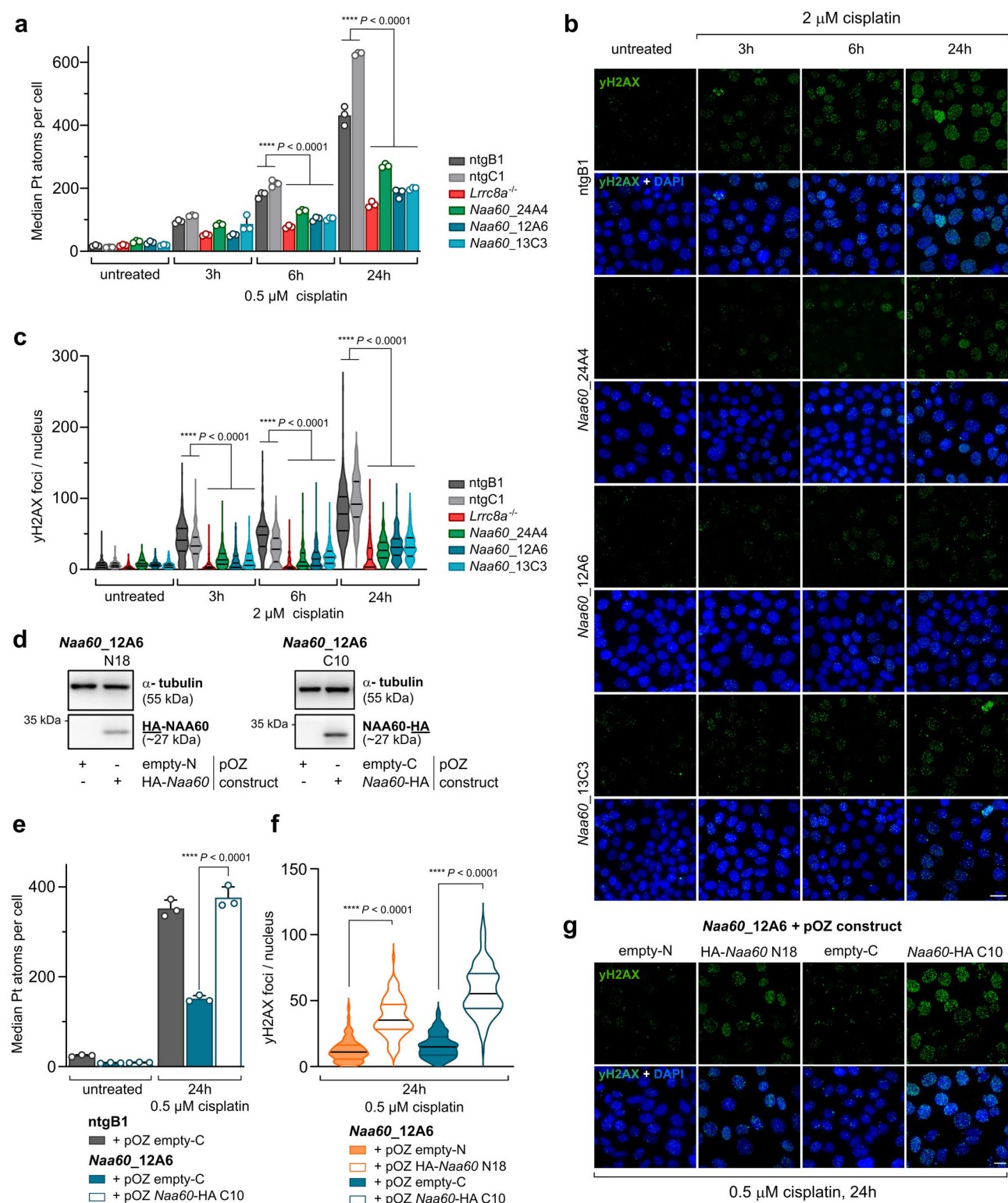

LRRC8D, due to its lower abundance, we excised the gel band corresponding to the molecular weight of LRRC8D in either parental or *Naa60* KO samples and subjected these to mass spectrometry. As for LRRC8A, the Nt acetylation was only identified in the parental group, whereas the unmodified N-terminus was present in the samples of the *Naa60* KO group, (Fig. 3b and Supplementary Data 2). Although the Nt peptide was not identified in all replicates, our data show that NAA60 is indeed important for Nt acetylation of LRRC8A and LRRC8D.

To investigate the co-localization of NAA60 and LRRC8A, we used a proximity ligation assay (PLA), which assesses close proximity of two proteins that are therefore likely to interact with each other. We expressed N-terminally HA-tagged NAA60 and C-terminally Myc-tagged LRRC8A or LRRC8D in KB1PM5 double knockout cell lines *DbKO*_A (*Naa60⁻/⁻*, *Lrrc8a⁻/⁻*) and *DbKO*_D (*Naa60⁻/⁻*; *Lrrc8d⁻/⁻*). Following the PLA reaction, the formation of fluorescent foci confirmed the close proximity of the NAA60 and LRRC8A target proteins. In the case of NAA60 and LRRC8A,

**Fig. 2 | Loss of *Naa60* reduces cisplatin uptake and subsequent accumulation of DNA damage. a** CyTOF-based measurement of Pt drug uptake over time using 0.5 μM cisplatin in wild-type control, *Lrrc8a*-, and *Naa60*-knockout cell lines. The data represents the median ± SD of three independent replicates where approximately 10,000 cells per condition and cell lines were acquired (two-way ANOVA followed by Tukey's multiple comparisons test). **b** Representative images of yH2AX immunofluorescence staining of *Naa60* knockout and control cell lines following 2 μM cisplatin treatment. The scale bar represents 20 μm. **c** Quantification of yH2AX foci in the nucleus of *Lrrc8a* and *Naa60* knockout and control cell lines in response to cisplatin treatment. Per cell line and condition, approximately 150 nuclei were quantified for each replicate. Lines at median and quartiles of three independent replicates are shown (ordinary one-way ANOVA followed by Tukey's multiple comparisons test). **d** Western blot expression confirmation of the pOZ-HA-*Naa60*

or pOZ-*Naa60*-HA constructs in the *Naa60* knockout cell line *Naa60*_12A6. **e** CyTOF-based Pt drug uptake measurement of wild-type, *Naa60* knockout and the *Naa60*-HA C10 rescue cell lines after 24 h of treatment with 0.5 μM of cisplatin. The data represents the median Pt counts of three independent replicates where approximately 50,000 cells per condition and cell lines were acquired (two-way ANOVA followed by Tukey's multiple comparisons test). **f** Quantification of yH2AX foci in the nucleus of *Naa60* knockout and rescue cell lines in response to cisplatin treatment. Per cell line and condition, approximately 150 nuclei were quantified for each replicate. Lines at median and quartiles of three independent replicates are shown (ordinary one-way ANOVA followed by Tukey's multiple comparisons test). **g** Representative images of yH2AX immunofluorescence imaging of empty vector or *Naa60* cDNA transduced *Naa60*_12A6 cell line upon 0.5 μM cisplatin treatment over the course of 24 h. The scale bar equals 10 μm.

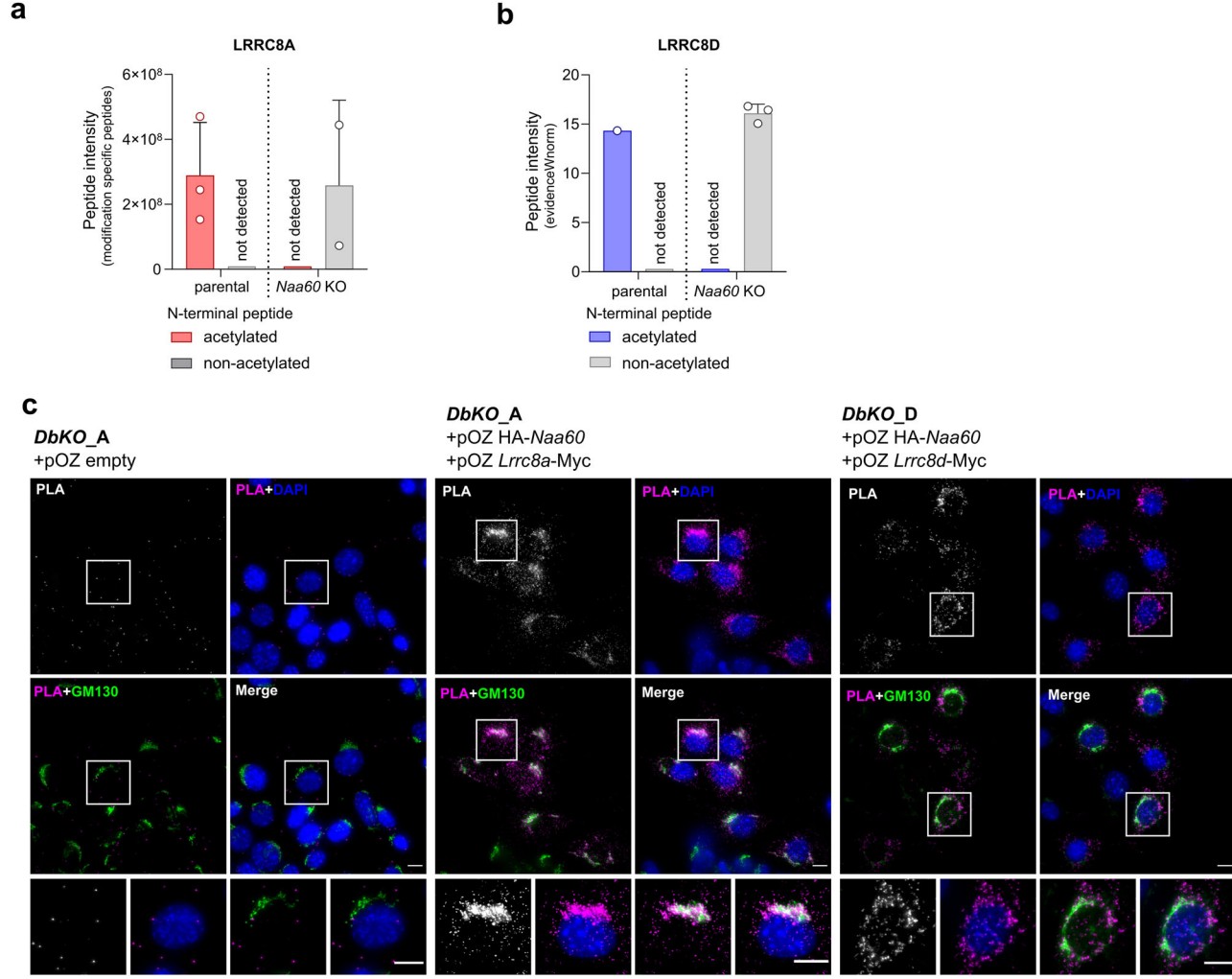

**Fig. 3 | LRRC8A and LRRC8D N-termini are a substrate for NAA60. a** Mass spectrometry analysis of the Nt peptide of LRRC8A after enrichment by immuno-precipitation using anti MYC-magnetic beads. Displayed are the peptide intensity values of three IP replicates for the detection of the acetylated peptide and two for the detection of the non-acetylated peptide. **b** The Nt peptide of LRRC8D was identified and analyzed by molecular weight focused gel band cutting and subsequent proteomics analysis. The results display the measurement of four gel bands for each

group. The Nt peptide could however, not be identified in all analyzed samples. Data represent mean ± SD. **c** Proximity ligation assay (PLA) between the N-terminally HA tagged NAA60 and the C-terminally MYC tagged LRRC8A, shown together with a co-staining for the Golgi Marker 130 (GM130). The scale bars for both full size and cropped images equal 10 μm. The PLA was performed three independent times with similar results. Representative images of one of the replicates are shown.

these foci accumulate at the Golgi apparatus which we assessed by co-staining with a Golgi marker protein (GM130) (Fig. 3c and Supplementary Fig. 6). The formation of PLA foci was also observed for HA-NAA60 and LRRC8D-Myc. However, for LRRC8D a scattered localization pattern was observed (Fig. 3c and Supplementary Fig. 6). The number and localization pattern of the foci may depend on the expression levels of the individual

proteins. The higher dispersion levels seen in some of the images could be an artifact of NAA60 overexpression.

## LRRC8A and NAA60 are epistatic
To test an epistatic interaction of NAA60 with LRRC8A/D, cDNA of either *Naa60* or *Lrrc8a* or both (Fig. 4a, b) were reintroduced into the *DbKO*_A cell

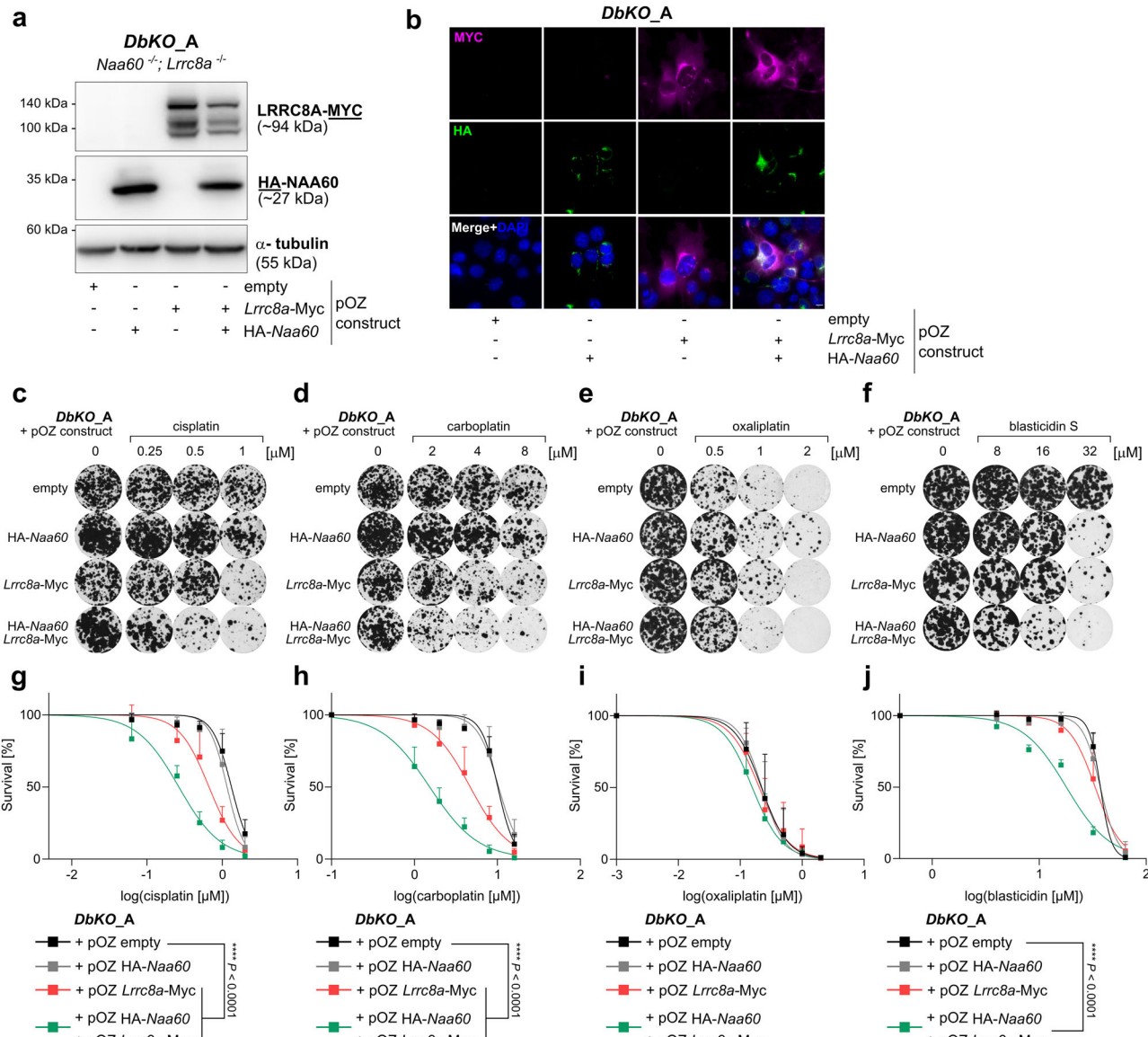

**Fig. 4 | NAA6 and LRRC8A are epistatic. a** Western blot rescue control, where either HA-NAA60 or LRRC8A-MYC or both proteins were reintroduced into a double knockout line *DbKO*_A (*Naa60*[−/−]; *Lrrc8a*[−/−]). **b** Immunofluorescence imaging of the HA or MYC tags on the proteins, which were rescued into the *DbKO*_A line. The scale bar equals 10 μm. **c–f** Clonogenic survival assays of the different *DbKO*_A rescue cell lines treated with cisplatin, carboplatin, oxaliplatin, or blasticidin S for 24 h with the indicated drug concentrations. Representative images of selected concentrations are shown. **g–j** Quantification of clonogenic growth assays using the *DbKO*_A-based rescue cell lines in the presence of cisplatin, carboplatin, oxaliplatin, or blasticidin S. Data represent mean ± SD of four independent replicates and were fitted to a non-linear regression dose-response curve (log(inhibitor) vs. normalized response - Variable slope). *p* values are calculated by one-way ANOVA followed by Tukey's multiple comparisons test for the log(IC50) values of the survival curves.

line and growth assays to detect changes in the Pt-drug or blasticidin S response of each rescue line were performed (Fig. 4c–j). When reconstituting *Naa60* cDNA alone, no clear change in the response to cisplatin, carboplatin or blasticidin S was detected (Fig. 4c–j and Supplementary Data 2). When *Lrrc8a* alone is rescued, partial restoration of the sensitivity is observed. In contrast, the reconstitution of both *Naa60* and *Lrrc8a* cDNA fully restored the sensitivity of the double knockout cell line towards cisplatin, carboplatin, and blasticidin S (Fig. 4c–j). As expected, no difference was observed for oxaliplatin, as its uptake seems to be independent of NAA60 or LRRC8A (Fig. 4e, i). Hence, our genetic complementation assays unambiguously show an epistatic relationship between NAA60 and LRRC8A.

As expected, the reconstitution of *Naa60*, or *Lrrc8d* alone into the *DbKO*_D was sufficient to sensitize the cells to cisplatin, carboplatin, and blasticidin S (Supplementary Fig. 6). Due to the presence of the obligatory

subunit LRRC8A, the channel is still present in the plasma membrane, and the addition of either NAA60 or LRRC8D was able to sensitize the cells. Double rescue of NAA60 and LRRC8D further sensitized the cells to cisplatin, carboplatin, as well as blasticidin S (Supplementary Fig. 7 and Supplementary Data 3).

## LRRC8A is enriched at the plasma membrane of NAA60-deficient cells

Upon loss of NAA60, LRRC8A protein expression levels remained unaffected (Fig. 5a, b and Supplementary Data 2). However, LRRC8D protein levels were increased in the NAA60-deficient cell lines, a phenomenon we also observed in cell lines that had lost LRRC8A[11] (Fig. 5a, b). As changes in the Nt acetylation of a protein might affect its stability or degradation, we performed a cycloheximide chase assay (Fig. 5c, d and Supplementary Data 2). However, no significant difference in the degradation of LRRC8A

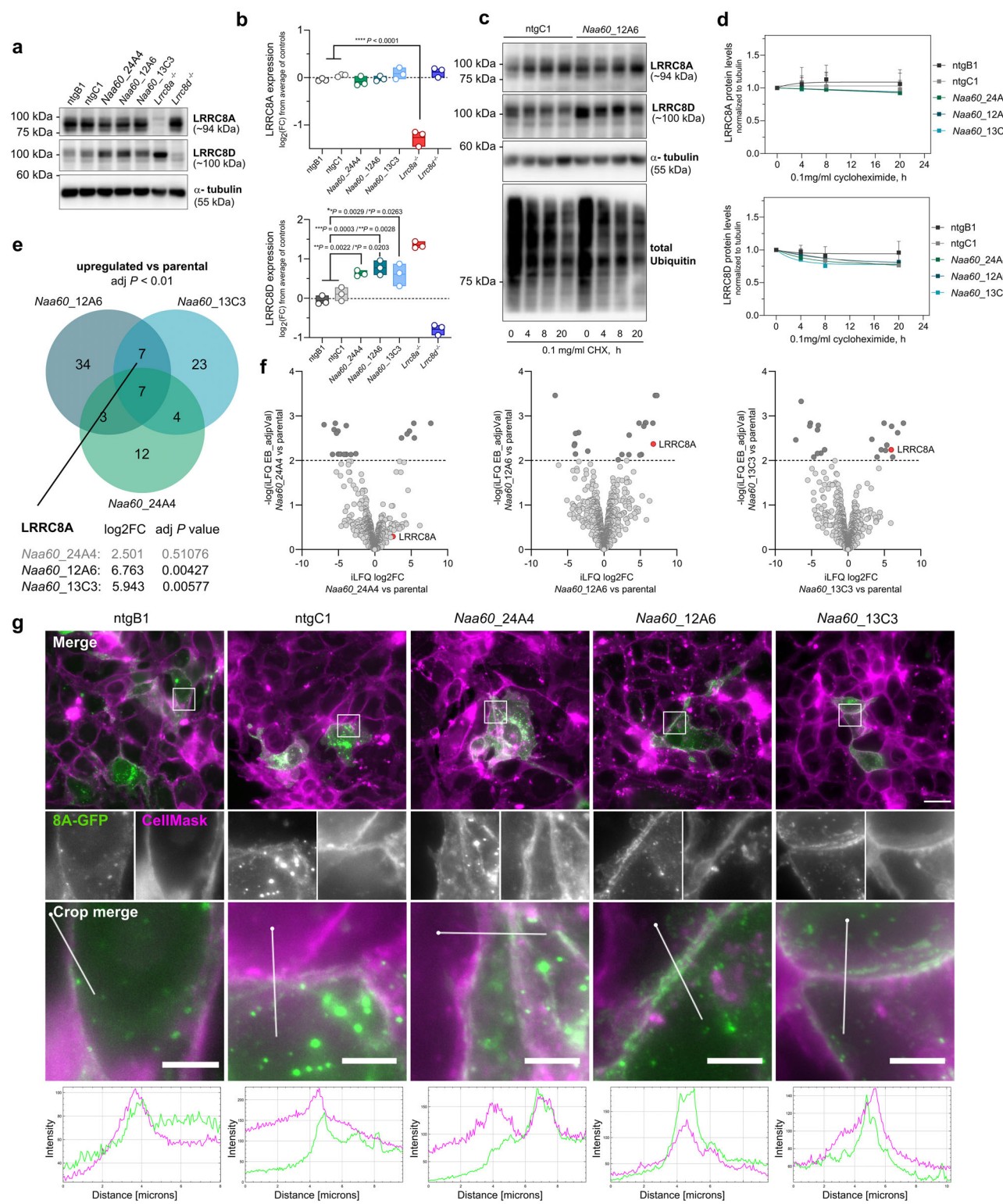

or LRRC8D in the absence of NAA60 was observed compared to the control cell lines (Fig. 5d). As LRRC8A and LRRC8D exhibit their role for the VRAC at the plasma membrane, we considered that the NAA60-mediated acetylation might be important for the trafficking of LRRC8A/D to the plasma membrane. This was recently shown for the phosphate importer SLC20A2: biallelic deletions of *NAA60* were found to cause primary familial brain calcification, because the lack of the NAA60-specific Nt acetylation activity results in decreased SLC20A2 levels at the plasma membrane[43]. We therefore analyzed plasma membrane-enriched protein fractions of wild-type

and *Naa60* knockout cell lines using mass spectrometry. Surprisingly, the analysis revealed that rather than absent, the subunit LRRC8A was instead significantly enriched at the plasma membrane of the two NAA60-deficient cell lines *Naa60*_12A6 and *Naa60*_13C3, when compared to the parental cell line (Fig. 5e, f and Supplementary Data 2). Together with the upregulation of the channel subunit LRRC8D in NAA60-deficient cell lines, this suggests a cellular reaction to compensate a reduction in VRAC function. Regarding endogenous LRRC8D, we did not detect enough peptides during the plasma membrane analysis to draw significant conclusions.

**Fig. 5 | LRRC8A is enriched at the plasma membrane of NAA60-deficient cells.**
**a** Representative Western Blot of the LRRC8A and LRRC8D expression in wild-type and NAA60-deficient cell lines. **b** Quantification of LRRC8A and LRRC8D protein levels in wild-type and NAA60-deficient cell lines. The protein intensities were corrected according to the tubulin signal. Three independent replicates were quantified. The floating bar plot displays the line at mean. *p* values are calculated by ordinary one-way ANOVA followed by Tukey's multiple comparisons test.
**c** Representative Western Blot of the cycloheximide chase assay for LRRC8A, LRRC8D, and tubulin loading control. Results of ntgC1 and *Naa60*_12A6 are shown. The cells were treated with 0.1 mg/ml of cycloheximide for the indicated time periods (0 h, 4 h, 8 h, 20 h) followed by immunoblot analysis. As a positive control for the assay, a total ubiquitin antibody staining was included. **d** Quantification of the cycloheximide chase assay of LRRC8A or LRRC8D in wild-type and NAA60-deficient cell lines. LRRC8A and LRRC8D signals were quantified and corrected according to the tubulin signal intensity. The resulting value at the 0 h timepoint was set as 100%. The data was fit to a one-phase decay curve. Data represent mean ± SD

of five independent replicates for LRRC8A and three independent replicates for LRRC8D. **e** Venn diagram displaying the overlaps of all significantly (adj. *p* < 0.01) upregulated proteins in the NAA60-deficient cell lines compared to the parental KB1PM5 cell line. **f** Differential plasma membrane enriched protein expression analysis of *Naa60* knockout cell lines (*Naa60*_24A4, *Naa60*_12A6, and *Naa60*_13C3) in comparison to the parental KB1PM5 cell line. Data from three replicates of the plasma membrane protein enrichment and analysis are shown. The dotted line indicates the –log(*p*Value) significance cutoff of 2.
**g** Immunofluorescence analysis of LRRC8A-GFP (8A-GFP) expressed in wild-type (ntgB1, ntgC1) and *Naa60* knockout cell lines. Two replicates with three independent transfections of the LRRC8A-GFP construct were performed. Similar results were observed for all. Representative images of the co-localization with CellMask membrane stain are shown. The linear intensity profiles of each channel were generated using the Plot profile plugin of ImageJ. The white dot marks the start of the analysis along the selected linear ROI. The scale bar on the full-size images represents 20 μm and on the cropped images 5 μm.

In addition to the mass spectrometry analysis of the endogenous LRRC8A levels at the plasma membrane, we also observed that GFP-tagged hLRRC8A expressed in the NAA60-deficient KB1PM5 cells still co-localized with the plasma membrane marker CellMask™ at the plasma membrane (Fig. 5g). Hence, trafficking of LRRC8A to the plasma membrane is not abrogated in the absence of NAA60-dependent acetylation. We therefore hypothesized that it is the actual function of LRRC8A and LRRC8D at the plasma membrane that is impaired in the absence of Nt acetylation.

### A positive charge at the N-termini of LRRC8A or LRRC8D reduces Pt drug uptake

Nt acetylation is known to neutralize the positive charge at the N-terminus of connexins, which are transmembrane proteins that are structurally related to LRRC8 proteins and form hexameric hemichannels[44–46]. The otherwise remaining positive charge at the primary amine alters the substrate permeability of these channels under physiological conditions[45,46]. A positive charge within the pore lining not only modulates the substrate specificity of channels from cations to anions, it is also thought to have a general pore-tightening effect[14,45].

To simulate the importance of the neutralizing effect of the acetylation at the N-termini of the LRRC8 subunits, we generated mutated rescue constructs. To this purpose, we exchanged the second amino acid at the N-terminus of both LRRC8A and LRRC8D for the positively charged arginine (LRRC8A p.I2R, LRRC8D p.F2R) and introduced these constructs into clonal *Lrrc8a* or *Lrrc8d* knockout lines, respectively (Fig. 6a–c). The initiator methionine was left unchanged as it is crucial for the N-terminus to fit into the substrate category of NAA60. For the *Lrrc8a* wild-type rescue construct we observed a restoration of sensitivity towards cisplatin, carboplatin, and blasticidin S (Fig. 6d–l and Supplementary Data 2). Interestingly, the mutated *Lrrc8a* construct failed to rescue the sensitivity to cisplatin, carboplatin, and blasticidin S to the same level as the wild-type construct. Also, for the *Lrrc8d* rescue constructs, only the wild-type construct rescued the sensitivity of the knockout lines for cisplatin, carboplatin, and blasticidin S, whereas the mutant construct did not restore the response, despite its higher expression level than that of the wild-type *Lrrc8d* rescue (Fig. 6b). These findings were further corroborated by the immunofluorescence analysis of γH2AX foci formed after treatment with cisplatin, where the reintroduction of the wild-type construct increased γH2AX foci formation to higher levels than the mutated constructs (Fig. 6m–p). Hence, a remaining positive charge at the N-termini of LRRC8A or LRRC8D affects the sensitivity towards drugs that enter cells through VRAC. Therefore, our data suggest that NAA60-mediated acetylation facilitates LRRC8A- and LRRC8D function by neutralizing their N-termini.

### Discussion
In this study, we identified NAA60 as a critical mediator of cis- and carboplatin uptake and our data suggest that this is due to Nt acetylation of the

LRRC8A and LRRC8D subunits of VRAC (Fig. 7). Previous proteomic analyses identified Nt acetylation of LRRC8A, LRRC8C[42], and LRRC8D[14], but could not identify a specific NAT responsible for this modification.

We found NAA60 in a functional genome-scale CRISPR/Cas9 screen using BRCA2;p53-deficient mouse mammary tumor cells in parallel with LRRC8A. Depletion of NAA60 was also significantly enriched in a genome-wide CRISPR/Cas9 screen for cisplatin resistance in the human *BRCA1*-mutant ovarian cancer cell line COV362[38]. Pt drugs are successfully used to treat patients with *BRCA1/2*-mutated tumors, thanks to the defect in homology-directed DNA repair these tumors harbor. The role of NAA60 in modifying LRRC8A or LRRC8D function seems to be independent of BRCA status and is more likely a general mechanism of VRAC regulation: together with *LRRC8D*, *NAA60* has also been identified amongst the top 5 hits in a CRISPR/Cas9 screen for cisplatin resistance using the HR-proficient HAP1 cells (*p* = 5.49−05)[47]. Moreover, *NAA60* was identified as a top hit in a cGAMP uptake/efflux CRISPR/Cas9 screen alongside *LRRC8A*. VRAC was thus determined to be the channel in charge of cGAMP exchange across the plasma membrane[48]. A link to DNA repair may be the initial observation that NAA60 has implications in free histone acetylation (KAT) activity, which facilitates nucleosome assembly[49]. However, recent crystal structures deem KAT activity unlikely due to the GNAT fold structure's inaccessibility for lysine side chains[50,51].

The resistance towards the protein synthesis inhibitor blasticidin S after treatment of NAA60-deficient cells prompted us to further investigate a potential enzyme-substrate interaction between the NAA60 and the VRAC subunits. Close intracellular proximity of NAA60 with the channel subunits was confirmed by PLA, where the foci localized to the Golgi apparatus. The fact that in the absence of NAA60 there is an enrichment of LRRC8A at the plasma membrane, we interpret as a compensation mechanism for impaired channel function. While our mass spectrometry analysis indicated an increased abundance of LRRC8A in plasma membrane-enriched fractions of NAA60-deficient cells, we acknowledge that the biochemical enrichment approach used may co-isolate membrane-bound compartments such as the endoplasmic reticulum. Therefore, some degree of intracellular membrane contamination cannot be excluded. Using CyTOF, we confirmed that the uptake of cis- and carboplatin is indeed reduced by more than 50%. Intracellular cisplatin accumulation was decreased in *Naa60* knockout cell lines in a similar fashion as in cells that lack LRRC8A, which was previously found to entirely abrogate VRAC formation at the plasma membrane[52]. To measure NAA60-mediated Nt-acetylated versus un-Nt-acetylated peptides in our in vitro assay, we used mass spectrometry. As these values are low, further corroboration of our data using radioactively labeled Ac-CoA as a readout may be useful.

Nt acetylation is understood to neutralize the positive charge on the primary amine at the N-terminus of substrate peptides[53]. In channels, certain substrate selectivity filters are based on the presence of charged amino acids within the channel pore[14,15,45,54–56]. Remaining positive charges at the N-termini of channels might change the permeability towards charged

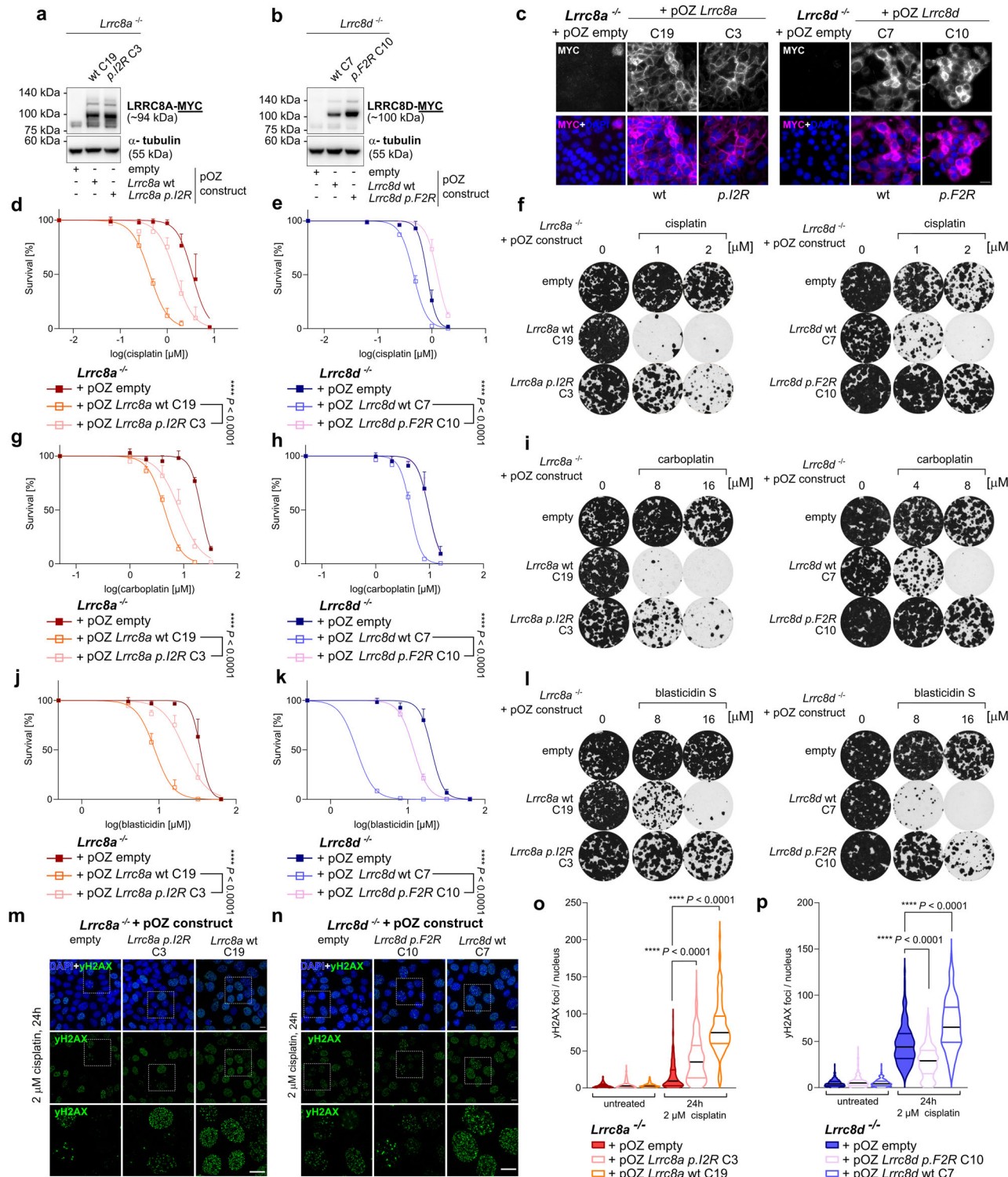

**Fig. 6 | A positive charge at the N-termini of LRRC8A or LRRC8D reduces Pt drug uptake. a, b** Western blot rescue control of the cell lines, where either wild-type (wt) or N-terminally mutated LRRC8A (*p.I2R*) or LRRC8D (*p.F2R*) constructs were reintroduced into LRRC8A or LRRC8D-deficient cells. **c** Immunofluorescence imaging of the clonal rescue cell lines, expressing the C-terminally MYC tagged LRRC8 subunits, the scale bar represents 20 μm. Clonogenic survival assays of the different LRRC8A or LRRC8D rescue cell lines treated with cisplatin (**d–f**), carboplatin (**g–i**), and blasticidin S (**j–l**) for 24 h with the indicated drug concentrations. Representative images of selected lines and concentrations are shown. The data represent mean ± SD of three independent replicates and were fitted to a non-linear regression dose-response curve (log(inhibitor) vs. normalized response -Variable

slope). *p* values are calculated by one-way ANOVA followed by Tukey's multiple comparisons test for the log(IC50) values of the survival curves. **m, n** Representative yH2AX immunofluorescence images of the LRRC8A or LRRC8D-deficient cells expressing either the wt or mutated *Lrrc8a* or *Lrrc8d* rescue constructs treated with 2 μM cisplatin for 24 h. The scale bar represents 10 μm. **o, p** Quantification of yH2AX foci in the nucleus of LRRC8A or LRRC8D-deficient cells expressing either the wt or mutated *Lrrc8a* or *Lrrc8d* rescue constructs treated with 2 μM cisplatin for 24 h. Per cell line and condition, approximately 100 nuclei were quantified. Lines at median and quartiles of three independent replicates are shown (ordinary one-way ANOVA followed by Tukey's multiple comparisons test).

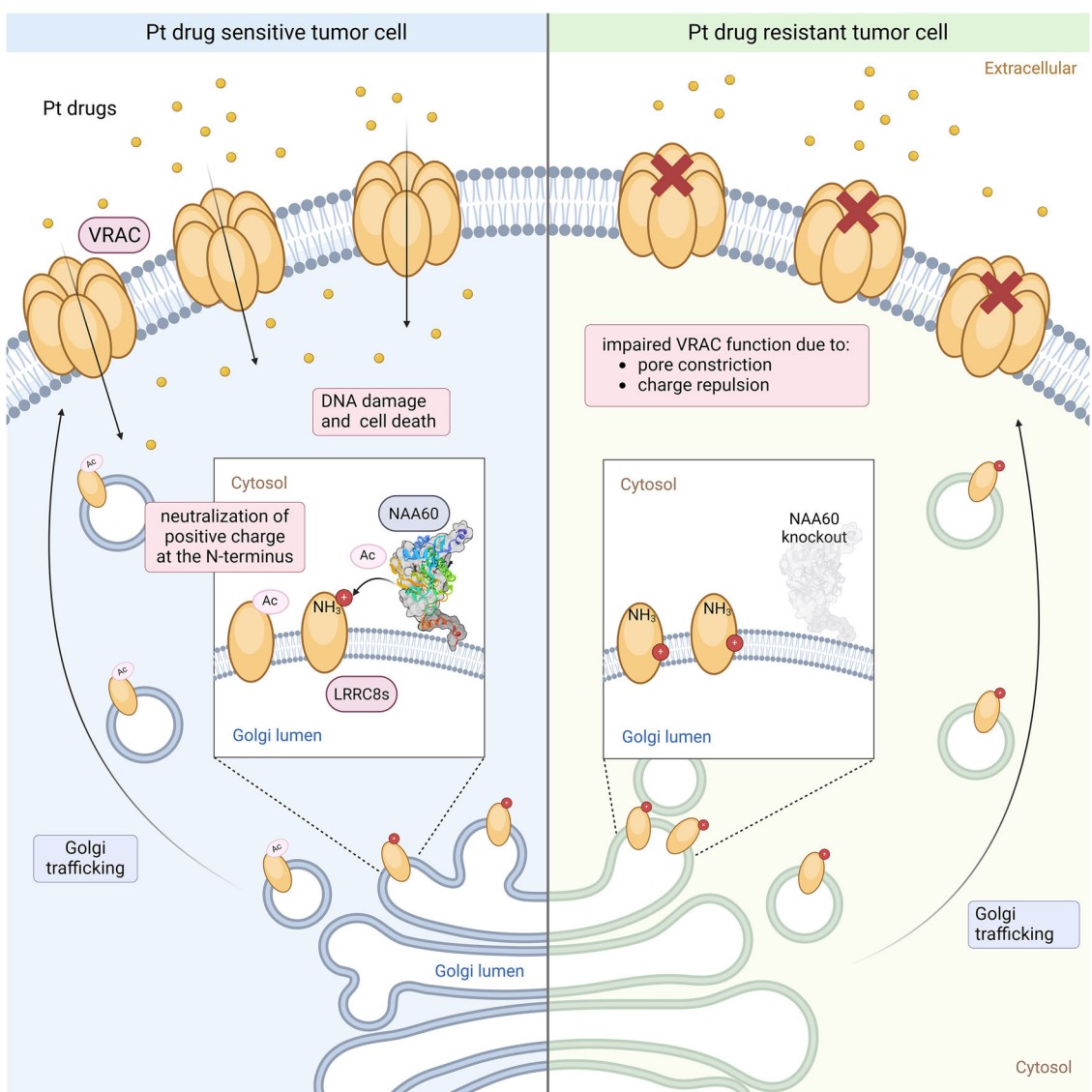

**Fig. 7 | Illustration of how NAA60 facilitates LRRC8A- and LRRC8D-mediated platinum drug uptake.** Upon loss of NAA60, a positive charge at the N-terminus of the LRRC8 subunits remains non-neutralized and potentially leads to channel constriction, which might prevent passing of larger molecules such as cisplatin and carboplatin. This leads to decreased cellular Pt-drug accumulation and lower DNA damage levels, resulting in the resistance phenotype of NAA60-deficient cells. Created in BioRender. Widmer (2025) https://BioRender.com/9t5q7sr.

substrates, potentially acting as a repellent in the case of positively charged substrate molecules. This was previously reported by Kwon et al. for Connexin 26 (Cx26), where the channel remained anion selective in the absence of Nt acetylation[45]. For Cx46 and Cx50 an enhanced cation-to-anion specificity was observed upon Nt acetylation[46].

While the introduction of a positive charge at the N terminus of LRRC8A or LRRC8D does not alter cellular localization, its function appears to be abrogated as cells show decreased Pt drug sensitivity. Based on our data, we infer that the neutralization of the N-termini by acetylation is important for the permeability of hydrolyzed cisplatin and carboplatin, which are often present in aqueous solutions. Hydrolysis of cis- or carboplatin creates the activated form, a positively charged complex[57], which might have altered VRAC permeability. Changed cation to anion properties, when the Nt amino acid is not neutralized, appear to exclude hydrolyzed Pt agents from passing through the channel pore.

Nakamura et al. investigated the crystal structures of hsLRRC8A and hsLRRC8D. They identified constricting residues along the pores of both (i.e. R103 for LRRC8A or E6 and F143 for LRRC8D). Changing the pore lining F143 residue of HsLRRC8D to the positively charged amino acid

arginine resulted in decreased permeability to the negatively charged glutamate and gluconate. These results indicate that the introduction of a positive charge at the constricted site leads to further tightening of the pore. As a result, there is reduced uptake of the negatively charged substrates, rather than an increase in uptake due to electrostatic attraction of the positive and negative charges[14]. Previous reports also suggested that the positive charge and the bulkiness of Arg103 in LRRC8A may be involved in the anion selectivity filter[14,58]. The replacement of Arg103 by Phe143 in LRRC8D may increase the permeability of VRAC to organic compounds, predominantly by an increase in the pore diameter, when the LRRC8D subunit is included in the channel composition[14].

In contrast, Zhou et al. hypothesized that the introduction of positive charges to the pore lining of VRACs might lead to conformational changes of the pore due to electrostatic repulsion of the positively charged segments to one another[33]. We cannot exclude that such a destabilization may also be caused by a positive charge at the initiator methionine of the LRRC8 subunits and this hypothesis should be further investigated.

Moreover, while our use of MR N-termini mutations was intended to ensure the retention of a positive charge at the N-terminus, it is important to

acknowledge that the Nt-acetylation status of these mutated N-termini remains unverified. Future studies should assess whether these MR N-termini are acetylated and by which NAT, as well as explore alternative strategies such as the use of proline at position two to fully elucidate the functional implications of Nt-acetylation in LRRC8A and LRRC8D.

Our data show that the loss of the NAA60-mediated Nt acetylation, and the subsequently remaining positive charge within the channel pore, interfere with the passing of cisplatin and carboplatin (Fig. 7). The decrease in cellular accumulation of these Pt-based agents then leads to decreased levels of DNA damage and reduced survival of mice with NAA60;BRCA1;p53-deficient mouse mammary tumors. Hence, our finding provides novel insights into the regulation of VRAC function and contributes to our understanding of substrate specificity.

## Methods

### Lead contact and material availability
Further information and requests for resources and reagents should be directed to and will be fulfilled by the Lead Contact, Sven Rottenberg (sven.rottenberg@unibe.ch).

### CRISPR/Cas9-based genetic screen
The cisplatin resistance screens were performed in the (K14cre;Brca2$^{F/F}$; Trp53$^{F/F}$) KB2P 1.21 tumor cell line, which was previously established from a KB2P mouse mammary tumor (Evers et al.[35]). The mouse GECKO v2 library, pool B (62,804 gRNAs targeting 20,628 genes (3 gRNAs/gene), including 1000 control non-targeting gRNAs), was stably introduced into the cells by lentiviral transduction at a multiplicity of infection (MOI) of 0.3. For each replicate an independent transduction of the parental cell line was performed. To perform the genetic screen at 100x library coverage, $6 \times 10^6$ mutagenized KB2P 1.21 cells in each replicate were plated onto 15 cm diameter dishes at a density of 300,000 cells per dish. Twenty-four hours after seeding, the medium was replaced with either fresh medium (control) or medium containing 0.2 μM or 0.25 μM cisplatin (accord healthcare, 1 mg/ml, Cat#368668). After 24 h, the medium was replaced with fresh medium and the cells were left to recover, with fresh medium changes every 3 days. On day 12, cells were trypsinized and counted. A total of $6 \times 10^6$ cells per replicate and treatment group were pelleted and stored at −80 °C. Genomic DNA was extracted from the frozen cell pellets (3 replicates of non-treated cells, cisplatin 0.2 uM and 0.25 uM each) using the Gentra Puregene Kit (Qiagen, Cat#158845) and the DNA content was measured with Nanodrop. Subsequently, the gRNA sequences were amplified from genomic DNA by two rounds of PCR amplification as described previously[59,60]. Resulting PCR products were purified using MinElute PCR Purification Kit (Qiagen, Cat#28006) and submitted for Illumina sequencing. Sequence alignment and enrichment analysis were carried out using MAGeCK software[36]. Raw sequencing files are deposited on the European Nucleotide Archive under the study accession number PRJEB75036.

### 2D and 3D cell culture
The K14cre;Brca1$^{F/F}$;Trp53$^{F/F}$;Mdr1a/b$^{−/−}$ (KB1PM5) cell line was previously established and described by Jaspers et al.[39]. Cells were grown in Dulbecco's Modified Eagle Medium/Nutrient Mixture F-12 (DMEM/F12; Gibco, Thermo Scientific Cat#10565018) supplemented with 10% fetal calf serum (FCS, biowest Cat#S1810-500), 50 units/ml penicillin-streptomycin (Gibco, Thermo Scientific Cat#15070063), 5 μg/ml Insulin (Sigma Cat#I0516), 5 ng/ml cholera toxin (Sigma Cat#C8052) and 5 ng/ml murine epidermal growth-factor (EGF, Sigma Cat#E4127). Tissue culture of BRCA-deficient cell lines was carried out under low oxygen conditions (37 °C, 5% CO$_2$; 3% O$_2$). Testing for mycoplasma contamination was performed twice per year.

Organoids were embedded in Culturex Reduced Growth Factor Basement Membrane Extract Type 2 (Trevigen biotechne Cat#3533-010010) 40 μL BME:growth media 1:1 drop in a single well of 24-well plate) and grown in Advanced DMEM/F12 (AdDMEM/F12, Gibco Cat#11550446) supplemented with 1 M HEPES (Sigma Cat#H0887), GlutaMAX (Gibco Cat#35050061), 50U/ml penicillin-streptomycin (Gibco

Cat#A9165), B27 (Gibco Cat#17504044), 125μM N-acetyl-L-cysteine (Sigma Cat#A9165) and 50 ng/ml murine epidermal growth factor (Sigma Cat#E4127). Organoids were cultured under standard conditions (37 °C, 5% CO2) and regularly tested for mycoplasma contamination. Before transplantation into mice, organoids were tested for pathogen contamination by IDEXX BioAnalytics services.

### Genome editing, plasmids, and cloning
Generation of CRISPR/Cas9 plasmids was performed using a modified version of the pX330 backbone (Addgene Cat#42230)[61] into which a puromycin resistance ORF was cloned under the hPGK promoter[62]. The sgRNA sequences (Supplementary Table 1) were cloned in the backbones using custom DNA oligos with the corresponding overhangs (Microsynth), which were melted at 95 °C for 5 min, annealed at RT for 2 h and subsequently ligated with quick-ligase (NEB Cat#M2200S) into BbsI (NEB Cat#R0539) digested pX330 or BsmBI-digested (Fermantas Cat#FD0454) lentiCRISPRv2 backbone. Sanger sequencing verified all construct's sequences.

NAA60–deficient 2D cell lines were generated by transfection with pX330 vectors containing gRNAs targeting the Naa60 coding gene sequence. In brief, KB1PM5 cells were transfected with 2.5 μg of plasmid DNA using the Mirus TransIT-LT1 reagent (Mirus Cat#MIR2300) and the corresponding protocol. Selection was performed using puromycin (Gibco, Thermo Scientific Cat#A1113802) at a concentration of 3 μg/ml for 72 h after transfection. Monoclonal cell lines were isolated by dilution of single cells per well into 96-well plates. Clones bearing big deletion mutations, created by pairing sgRNAs to target the same gene, were identified by gel electrophoresis resolution of PCR amplicons corresponding to edited loci (amplicon primer sequences below). Sanger sequencing furthermore confirmed the gene disruption. Amplicon primers are FW: 5'-ACAGAGCTCCGCTACTTTGC-3' and RV: 5'-GGATGGTCACGTCGGGTATC-3'. The gRNAs were designed to target exon 4 and 5 of Naa60 (amino acids 81-191), which encode for functional domains such as the GNAT fold including the Ac-CoA binding site (amino acids 108-113) of the NAA60 protein: the big deletion caused by gRNAs 1 + 2 removes the Ac-CoA binding site, the deletion caused by gRNAs 1 + 3 additionally includes[63] Furthermore, the deletions caused by gRNA 1 + 3 and 2 + 4 encompass Leu 140, Asn 143, Tyr 165, and Ile 167 which are important for Met1 recognition on the substrate peptide, substrate binding, and peptide anchoring. Mutations of these amino acids have been shown to fully abrogate or greatly reduce protein function[50].

### Reconstitution of Naa60, Lrrc8a, or Lrrc8d cDNA
Naa60 reconstitution was performed using the pOZ-N-FH-IL2Rα, for N-terminal tagging or pOZ-C-FH-IL2Rα, for C-terminal tagging plasmids (kindly provided by Dipanjan Chowdhury, Harvard Medical School). The Naa60 coding sequence was amplified from freshly prepared cDNA of KB1PM5 parental cells using primers that included the corresponding plasmid overlaps suited for in-fusion cloning using the in-fusion HD cloning kit by Takara Bio (Takara Cat#12141). The primer sequences are: N-term HA FW: 5'-GCCGGAGGACTCGAGAC AGAGGTGGTGCCTTCCA-3' and RV: 5'-TCTCGATGCGGCCGC TTACATGGTACGGCTGTACTCTATGCCA-3'. C-term HA FW: 5'-GATCTTCCGCTCGAGATGACAGAGGTGGTGCCTTCCA-3' and RV: 5'-TCCTCCAGCGGCCGCCATGGTACGGCTGTACTCTATGC CA-3'.

Lrrc8a, or Lrrc8d reconstitution was performed using a modified version of the pOZ-N-FH-IL2Rα plasmid. Briefly, the vector backbone was amplified using following primers, which excluded the original FLAG and HA tags from the linearized vector PCR product, FW: 5'-TCGAGA-GATCCGGGAGACACAA-3' and RV: 5'-CTCGAGCGGAAGATC TGGCAGTCT-3'. The Lrrc8a or Lrrc8d coding sequence was amplified from freshly prepared cDNA by primers including the C-terminal Myc sequence and corresponding plasmid overlaps suited for subsequent cloning

using the in-fusion HD cloning kit by Takara Bio (Takara Cat#12141). *Lrrc8a* FW: 5'-GATCTTCCGCTCGAGATGATTCCGGTGACAGAGC TCCGC-3' and RV: 5'-TTGTGTCTCCCGGATCTCTCGATGCGGCCC TACAGATCCTCTTCTGAGATGAGTTTTTGTTCTCCTCCAGCGGC CGCGGCCTGCTCCTTGTCAGCTC-3' *Lrrc8d* FW: 5'-GATCTTCC GCTCGAGATGTTTACCCTTGCGGAAGTTGC-3' and RV: 5'- TTGT GTCTCCCGGATCTCTCGATGCGGCCCTACAGATCCTCTTCTGAG ATGAGTTTTTGTTCTCCTCCAGCGGCCGCAATCCCGTTTGCAAA GGGGACA-3'. The N-terminally mutated *Lrrc8a* and *Lrrc8d* constructs, where the second position amino acid code was changed for the one encoding for the positively charged arginine, were created using the following FW primers: *Lrrc8a p.I2R* FW: 5'-GATCTTCCGCTCG AGATGCGCCCGGTGACAGAGCTCCGC-3' and *Lrrc8d p.F2R* FW: 5'-GATCTTCCGCTCGAGATGCGCACCCTTGCGGAAGTTGC-3' in combination with the RV primers listed above. Correct cDNA insertion was verified by Sanger sequencing of the complete open reading frames.

Fifty percent confluent Phoenix retrovirus producer cells (Gentaur Molecular Products Cat#RVK-1001) were transfected with pOZ-HA-*Naa60*, pOZ-*Naa60*-HA, pOZ-*Lrrc8a*-Myc, pOZ-*Lrrc8a*-I2R-Myc pOZ-*Lrrc8d*-Myc, pOZ-*Lrrc8d*-F2R-Myc or pOZ-empty using Turbofectin transfection reagent (Origene, LabForce, Cat#TF81001). The next day, virus-containing supernatant was collected, filtered through a 0.45 μm filter before application to NAA60, LRRC8A, or LRRC8D-deficient monoclonal target cells. Eight μg/ml Polybrene (Merck Millipore Cat#TR-1003-G) was added to each target cell dish. Virus was harvested and applied to target cells on three consecutive days. IL2Rα-expressing cells were selected using magnetic beads coated with CD25 antibody (Dynabeads CD25, Thermo Scientific Cat#11157D).

## Clonogenic assays

For clonogenic growth assays in 6-well plate (TPP Cat#92406) format, 2000 KB1PM5 cells were seeded per well in DMEM-F12 complete medium. Twenty-four hours after seeding, the cells were treated with the indicated drug doses over the course of 24 h. During the recovery period, the medium was exchanged for fresh growth medium every 4 days. After 8 days of recovery, the wells were fixed with 4% PFA/PBS and stained with 0.1% crystal violet. Quantification of the wells was performed with ImageJ using the ColonyArea plugin (ImageJ version 1.53i)[64]. The data were subsequently fitted to a non-linear regression dose-response curve (log(inhibitor) vs. normalized response -Variable slope) in GraphPad Prism software 9.

## Western blotting

Eighty percent confluent cells were washed and scraped in cold PBS. After pelleting, cells were lysed in RIPA buffer (50 mM Tris-HCl pH 7.4; 1% NP-40; 0.5% Na-deoxycholate; 0.1% SDS; 150 mM NaCl, 2 nM EDTA, 50 mM NaF) containing 1x complete protease inhibitor cocktail (Roche Cat#04693132001) for 60 min on ice, followed by centrifugation for 10 min at 14,000 rpm for supernatant clearance. Protein concentrations of the supernatants were determined using the Pierce BCA assay kit (Thermo Fisher Cat#23225) with a BSA standard curve. Protein lysates were denatured at 70 °C for 10 min in 6x SDS sample buffer and separated by SDS-PAGE on either 10% acrylamide gels for visualizing HA-tagged NAA60 or 7.5% acrylamide gels for LRRC8A/D visualization, or on 8.5 % acrylamide gels when both NAA60 and LRRC8A/D were visualized as for the double rescue experiments in Fig. 4 or Supplementary Fig. 7. Proteins were transferred via overnight wet transfer for 18 h at 15 V to 0.45 μm pore size PVDF membranes (GE Healthcare Cat#10600018). Membranes were blocked in 5% BSA in TBS-T (100 mM Tris, pH 7.5, 0.9% NaCl, 0.05% Tween-20) and subsequently incubated with primary antibodies anti-LRRC8A rabbit polyclonal (Bethyl Laboratories Cat#A304-175A), anti-LRRC8D rabbit polyclonal (Proteintech Cat#11537-1-AP), purified anti-HA.11 Epitope Tag Antibody (Clone 16B12) mouse monoclonal (BioLegend Cat#901501), or anti-Myc-tag (71D10) rabbit polyclonal (Cell signaling Cat#2278) or anti-Myc-Tag (9B11) mouse monoclonal (Cell signaling Cat#2276) diluted 1:1000 or anti-beta actin mouse monoclonal (Sigma Cat#A1978) and anti-

alpha tubulin, mouse monoclonal (Sigma Cat#T5168) diluted 1:2000 in blocking buffer at 4 °C overnight. After washing in TBS-T, Horseradish Peroxidase (HRP)-linked secondary antibodies anti-mouse IgG (Cell Signaling Cat#7076), or anti-rabbit IgG (Cell Signaling Cat#7074) diluted 1:2500 in blocking solution were applied for 2 h at room temperature. Images were acquired using the FUSION FX7 imaging system (Vilber GmbH).

For the expression analysis of LRRC8A or LRRC8D in in *Naa60* knockout cell lines in Fig. 5, 200,000 NAA60-deficient (*Naa60_24A4, Naa60_12A6, Naa60_13C3*) or proficient KB1PM5 cells (ntgB1, ntgC1) per condition were seeded to 60 mm diameter cell culture dishes (SPL Life Sciences Cat#20060). Two days after seeding, the cells were harvested for Western Blotting as described previously under the "Western blotting" section. A total of 60 μg of protein per sample were subjected to electrophoresis on a 7.5% acrylamide gel and subsequently transferred to 0.45 μM pore size PVDF membranes (GE Healthcare Cat#10600018). Membranes were blocked in 5% BSA in TBS-T (100 mM Tris, pH 7.5, 0.9% NaCl, 0.05% Tween-20) and subsequently incubated with primary antibodies anti-LRRC8A rabbit polyclonal (Bethyl Laboratories Cat#A304-175A), anti-LRRC8D rabbit polyclonal (Proteintech Cat#11537-1-AP) diluted 1:1000 in blocking buffer at 4 °C overnight or 1:2000 (anti-alpha tubulin mouse monoclonal (Sigma Cat#T5168) in blocking buffer for 2 h at room temperature. After washing in TBS-T, Horseradish Peroxidase (HRP)-linked secondary antibodies anti-mouse IgG (Cell Signaling Cat#7076) or anti-rabbit IgG (Cell Signaling Cat#7074) at a dilution of 1:2500 were applied for 2 h at room temperature. Images were acquired using the FUSION FX7 imaging system (Vilber GmbH).

## In vivo validation of resistance

All animal experiments were approved by the Animal Ethics Committee (BLV Bern, Switzerland, Application number BE40/18). All experiments were performed in accordance with the Swiss Act on Animal Experimentation (December 2015). CRISPR-Cas9-modified organoid lines were transplanted into 6–9 weeks-old female nude mice (Charles River, Crl:NMRI-Foxn1nu Strain Code 639) for the in vivo validation. No animals were excluded during the experiment and subsequent analysis. Based on previous experiments, we know that the wild-type tumor reaches a volume of 1000 mm³ in around 35 days ± 5 days under treatment with 6 mg/kg cisplatin or 50 mg/kg carboplatin and we expect that the knockout will affect the tumor growth by 10 days. To therefore reach a power of 0.8 and alpha 0.05, we need a minimum of 4 animals per group. One animal extra per group was added in case the tumor did not grow out or an animal had to be excluded from the experiment. The mice for this experiment are kept in the animal facility at the Institute of Animal Pathology (Länggassstrasse 122). Up to 5 mice are held in standard IVC housing in standard condition regarding air conditioning and light. The cages are only opened under a laminar flow hood. All animals are checked daily and the animals with tumors are weighed and checked for tumor size at least 3 times a week. As environmental enrichments, each cage contains a standard house, sterilized wood chips and tissues. Mice were acclimatized to the mouse facility and the personnel for at least 2 weeks before the start of the experiment. Transplantation is performed under general anesthesia with perioperative injection of 5 mg/kg Carprofen (Carprox® 50 mg/ml from Virbac) injected subcutaneously for analgesia. Wild-type (ntg) and NAA60-deficient (*Naa60_24A4*) organoid lines were cultured for 3 days. For transplantation, the organoids were collected, washed with advanced DMEM medium (AdDMEM/F12, Gibco Cat#11550446) supplemented with 1 M HEPES (Sigma Cat#H0887), GlutaMAX (Gibco Cat#35050061) 50 U/ml penicillin-streptomycin (Gibco Cat#A9165), B27 (Gibco Cat#17504044)) to remove the old BME. For the injection, the organoids were resuspended in BME at a density of 100,000 cells/20 μl of BME and continuously stored on ice to prevent polymerization. Skin incision is made and the 4th mammary fat pad is pulled out; a pocket is formed with the pointed forceps to orthotopically transplant 20 μl of the BME organoid suspension; mammary fat pad is put back and the skin is closed with surgical knots using absorbable threads

(Vicryl Plus 5-0, ETHICON). The mice are kept on a heating pad and with oxygen supply for recovery after surgery until they are completely awake. The tumor size and time/days after treatment were the primary outcome measurement. Other measurements included body weight and behavioral changes as an indicator of well-being. Mammary tumor size was measured by caliper measurements and tumor volume was calculated (length × width$^2$/2). Treatment of tumour-bearing mice was initiated when tumors reached a size of ~75 mm$^3$. Animals were randomly assigned to the treatment groups. Animals from the same cage received different treatments to minimize cage effect. The person measuring the tumor size was blinded to the treatment and hypothesis of the treatment groups. A total of 20 animals were used for this experiment ($N = 5$ per group). Cisplatin (Teva Cat#3150036) was administered intravenously at a dose of 6 mg/kg on day 0 and day 14. Similarly, carboplatin (Teva Cat#6985451) was administered at 50 mg/kg intravenously on day 0 and 14. Animals were anesthetized with isoflurane, sacrificed with $CO_2$ followed by tumor and organ harvest when the tumor reached a volume of 1000 mm$^3$. Humane endpoints as exclusion criteria included body weight loss >10% post treatment, tumor size above 1000 mm$^3$ or ulcerated tumor and no tumor outgrowth. A limitation regarding the translation of this study is the mice being immunosuppressed to tolerate the transplantation of Cas9-modified cells.

### Drug uptake measurement (CyTOF)

KB1PM5 *Naa60*, *Lrrc8a/d* wild-type and knockout and rescue cell lines were seeded 2 days prior to drug treatment. It was aimed to have a starting cell number of 300,000 cells per condition at a density of 80% on treatment day. On the day of treatment, cisplatin-containing medium was freshly prepared with the indicated concentration of 0.5 μM and cells were treated for 3, 6, and 24 h for the experiment in Fig. 2a or for 24 h for Fig. 2e. After the treatment, cells were washed 3 times with serum-containing culture medium for 5 min each wash. Subsequently, cells were washed with room temperature PBS and then incubated with 0.25% Trypsin EDTA. Trypsinization was stopped with serum-containing culture medium and the cells were then fully dissociated into a single-cell suspension by gentle pipetting. After dissociation, the cells were counted and 300,000 cells per condition were used for further fixation and barcoding according to the Cell-ID 20-plex Pd Barcoding kit (Fluidigm Cat#201060) protocol. Barcoded samples were then pooled and incubated with the Cell-ID intercalator Ir (Fluidigm Cat#201192 A) at 100 μl/1 Mio cells for 1 h at room temperature and then stored at −80 °C until measurement. For the measurement, samples were thawed, washed, mixed with equilibration beads and acquired on a Helios mass cytometer (Fluidigm). Post-acquisition, data were bead-normalized and debarcoded using the premessa R package released by the Parker Institute for Cancer Immunotherapy (https://github.com/ParkerICI/premessa). Absolute Pt- atom counts were determined by gating for Iridium$^{191}$ + BeadDist and $^{191}$Iridium + $^{193}$Irridium events to exclude cell debris, and $^{191}$Iridium + event length to exclude duplet signals using FlowJo version 10.8.1. Median Pt counts for the isotopes $^{192}$Pt, $^{194}$Pt, $^{195}$Pt, $^{196}$Pt, and $^{198}$Pt were summed to determine the total amount of Pt- atoms per cell. Data of a total of 3 independent replicates, where approximately 10,000 cells were acquired per condition are shown for Fig. 2a. For Fig. 2e three independent replicates, where approximately 50,000 cells were acquired per condition and replicate are shown. Statistical analysis was performed using GraphPad Prism software 9 (two-way ANOVA followed by Tukey's multiple comparisons test).

### In vitro acetylation assay

The C-terminally HA-tagged NAA60 enzyme was isolated from the clonal rescue line KB1PM5 *Naa60*_12A6 + pOZ *Naa60*-HA C10. Briefly, cells were seeded to 15 cm dishes and grown to 80% confluency. The cells from each dish were washed 1x with 5 ml of cold PBS, scraped in 5 ml cold PBS, and transferred to 50 ml tubes. The cells were pelleted at 1500 rpm, 4 °C, for 5 min and lysed in 0.5 ml of RIPA lysis buffer without SDS (50 mM Tris-HCl pH 7.4; 1% NP-40; 0.5% Na-deoxycholate; 150 mM NaCl, 2 nM EDTA, 50 mM NaF) containing 1x complete protease inhibitor cocktail (Roche

Cat#04693132001) per dish. The lysates were incubated on ice for 60 min, homogenized by seven syringe passes through a 25 G needle, and cleared by centrifugation at 14,000 rpm at 4 °C for 10 min. A total of 20 μl anti-HA magnetic bead slurry (Pierce Thermo Scientific Cat#88836) per reaction was washed 1x with TBS-T (0.05%), resuspended with the lysate and incubated for 1 h at room temperature on a rotator. Western Blot samples of the input and flowthrough were taken. The enzyme-loaded beads were then washed 6 x with 5 ml TBS-T (0.05%), for the last three washes, the beads were transferred to a new tube each time. After the final wash, the beads were resuspended in 1x acetylation buffer (50 mM HEPES pH 7.5, 100 mM NaCl, 1 mM EDTA) in the initial bead slurry volume. As a negative control for spontaneously occurring acetylation, beads without enzyme- loading were used. A total of 20 μl bead slurry per reaction were transferred to low protein-binding 2 ml reaction tubes (Thermo Scientific Cat#88379), the tubes were placed on the magnetic stand and the supernatant removed. The beads were subsequently resuspended in acetylation buffer containing a final concentration of 150 μM of Acetyl-CoA (Acetyl-CoA lithium salt, Merck Milipore Cat#A2181) and 100 μM of the individual peptides. The custom 24-mer peptides (Biomatik, 95% purity) are composed of the first seven amino acid residues of either LRRC8A (MIPVTEL), LRRC8D (MFTLAEV), the positive control hnRNP F (MLGPEGG), which previously has been shown to be a substrate of human hNaa50, and the negative control HMGB1 (SESSSKS), which should not be part of the NAA60 substrate cathegory[65]. The next 17 C-terminal carrier residues are derived from the Adrenocorticotropic hormone (RWGRPVGRRRRPVRVYP), where the three Lysines were exchanged for Arginines. Upon receipt, the peptides were dissolved in water to a stock concentration of 2.5 mM, aliquoted and stored at −80 °C until use. The final reaction mixes were then incubated at 37 °C, 600 rpm, for 2 h. Subsequently, the reactions were quenched by the addition of acetic acid to a final concentration of 50 mM. The reaction tubes were placed on a magnet and the supernatants transferred to new low protein-binding tubes and subjected to mass spectrometry analysis. Samples were diluted 1:40 in water containing 1% (v/v) acetonitrile and 0.1% (v/v) tri-fluoroacetic acid and 2 μL were analyzed on an Orbitrap Fusion Lumos instrument as described elsewhere[66] with the following changes. The acetonitrile gradient from 5–40% was developed within 10 min, peptide ions with charge 3–7 were included for ETD fragmentation with supplemental HCD activation, and fragment spectra were acquired in the orbitrap at resolution of 30,000, maximum injection time of 54 ms and AGC of 50,000. Fragment spectra were interpreted with FragPipe version 20.0[67] against the SwissProt human sequence database containing canonical protein sequences and their isoform (release 2023_04) supplemented with common contaminating protein sequences and the custom 24-mer peptide sequences using acetylation of protein N-termini as variable modification and normalization of intensities across runs activated.

### Mass spectrometry analysis of the Nt peptide of LRRC8A and LRRC8D

For LRRC8A, the MYC-tagged protein was enriched by immunoprecipitation using anti-MYC magnetic beads (Pierce, Thermo Fisher Cat#88842). C-terminally MYC-tagged LRRC8A was expressed in the KB1PM5 parental and *Naa60*_12A6 lines. The cells were lysed using RIPA sample buffer without SDS as described previously. The lysate was incubated with the beads for 2 h at RT and subsequently washed 6- times with TBS before subjecting to on-bead digestion and mass spectrometry as described previously[65]. For LRRC8D, the lysates of the KB1PM5 parental or *Naa60*_12A6 cell lines were run on a 7.5% acrylamide gel. Gel bands were visualized using Coomassie blue stain and the bands corresponding to the molecular weight of LRRC8D ( ~ 100 kDa) were excised and analyzed using in-gel digestion and shot-gun mass spectrometry[66]. An aliquot of 5 μL from each digest was analyzed on an instrumental setup consisting of an Ultimate 3000 nano-LC coupled to a LUMOS tribrid orbitrap mass spectrometer. Samples were loaded onto a pre-column (C18 PepMap 100, 5 μm, 100 A, 300 μm i.d. x 5 mm length) at a flow rate of 50 μL/min with solvent C (0.05% TFA in water/acetonitrile 98:2). After loading, peptides were eluted in back

flush mode onto a self-pack C18 CSH Waters column (1.7 µm, 130 Å, 75 µm × 20 cm) using an acetonitrile gradient of 5% to 40% solvent B (0.1% Formic Acid in water/acetonitrile 4,9:95) in 40 min for in-gel digests at a flow rate of 250 nL/min. The column effluent was directly coupled to the Fusion LUMOS mass spectrometer via a nano-spray ESI source. Data acquisition was made in data-dependent mode with precursor ion scans recorded in the orbitrap with resolution of 120,000 (at $m/z = 250$) parallel to top speed fragment spectra of the most intense precursor ions in the linear trap for a cycle time of 3 seconds. Mass spectrometry data were processed with MaxQuant/Andromeda version 1.6.14.0 using default settings for peak detection, a strict trypsin cleavage rule, allowing up to 3 missed cleavages, variable oxidation on methionine, acetylation of protein N-termini, and deamidation of asparagine and glutamine, and fixed carbamidomethylation of cysteines, respectively. Match between runs was used with a retention time window of 0.7 min between neighboring gel fractions. The SwissProt mouse protein sequence database (version 2020_07) enriched by common contaminants was used to interpret fragment spectra with an initial mass tolerance of 10 ppm on precursor and 20 ppm for fragment ions, respectively. Protein identifications were accepted if at least two razor peptides per protein group were identified at a 1% false discovery rate (FDR) cutoff. Differential protein abundance testing was performed as described elsewhere[67]. For the LRRC8D peptide, the intensities detected by MaxQuant were variance normalized to obtain the evidenceWnorm values shown in Fig. 3b.

## Immunofluorescence

For the yH2AX foci detection, the cells were seeded to coverslips, and treated with 2 µM cisplatin for the indicated time points. After treatment, cells were washed with PBS and fixed with 4% PFA/PBS for 20 min at 4 °C. Fixed cells were permeabilized for 20 min in 0.5% Triton X-100/PBS. All subsequent steps were performed in staining buffer (PBS, BSA (2%), glycine (0.15%), and Triton X-100 (0.1%)). Cells were washed 3 times and blocked for 30 min at RT using the described staining buffer. The cells were incubated overnight at 4 °C with the primary antibody anti-phospho-Histone H2A.X (ser139) (clone JBW301, Merck Millipore Cat#05-636) at a dilution of 1:200 in staining buffer, washed 3 times and subsequently incubated with the secondary antibody Goat anti-Mouse IgG (H + L) Cross-Adsorbed Secondary Antibody Alexa Fluor 488 (Thermo Scientific Cat#A11029) for 2 h at RT. The coverslips were washed 5 times after secondary antibody staining, counterstained with DAPI and mounted onto positively charged Superfrost Plus Adhesion Microscope Slides (epredia Cat#J1800AMNZ) using fluorescence mounting medium (Dako Cat#S3023). Analysis was performed on a DeltaVision Elite High Resolution Microscope system (GE Healthcare) consisting of an Olympus IX-70 inverted microscope with a CMOS camera, 100× Olympus Objective, and softworx (Applied Precision, Issaquah, WA, USA) software. Per condition and replicate, approximately 200 cells from different areas of the coverslips were imaged in Z-stacks of 31 slices of 0.2 µm thickness. Images were analyzed using the FIJI image processing package of ImageJ (1.8.0)[68]. Briefly, Z-stacks of the individual channels were projected using the "max intensity" projection setting. All nuclei were detected by the "analyze particles" command using the DAPI channel projection and this ROI selection was then used to determine the number of yH2AX foci of each nucleus by the "finding maxima" command on the FITC channel. ROI touching the edges of the images were excluded. Data were plotted in GraphPad Prism software 9 and significance was calculated using ordinary one-way ANOVA followed by Tukey's multiple comparisons test.

For immunofluorescence analysis of the NAA60 and LRRC8A/D single and double rescue cell lines, 60,000 cells per coverslip were seeded 1 day prior to fixation. The cells were washed with PBS and fixed with 4% PFA/PBS for 20 min at 4 °C. Fixed cells were permeabilized for 20 min in 0.1% Triton X-100/PBS. Post fixation, the cells were washed 3 times with PBS and blocked for 30 min at RT in 5% BSA in PBS. The cells were incubated overnight at 4 °C with the primary antibodies (anti-HA rabbit polyclonal (C29F4), Cell Signaling Cat#3724S; purified anti-HA.11 Epitope Tag Antibody (Clone 16B12), BioLegend Cat#901501; anti-Myc-Tag

(71D10) Rabbit mAb, Cell signaling Cat#2278; and anti-GM130-conj Alexa Fluor 488 (BD Biosciences Cat#560257)) at a dilution of 1:100 in blocking buffer. The coverslips were subsequently washed 3 times with PBS and incubated with the secondary antibodies (Goat anti-Mouse IgG (H + L) Cross-Adsorbed Secondary Antibody Alexa Fluor 488 (Thermo Scientific Cat#A11029) or Goat anti-rabbit IgG (H + L) Cross-Adsorbed Secondary Antibody Texas Red (Thermo Scientific Cat#T-2767)) at a dilution of 1:2000 for 2 h at RT. The coverslips were washed 5 times with PBS after secondary antibody staining, counterstained with DAPI and mounted onto positively charged Superfrost Plus Adhesion Microscope Slides (epredia Cat#-J1800AMNZ) using fluorescence mounting medium (Dako Cat#S3023). Analysis was performed on the DeltaVision Elite High Resolution Microscope system (GE Healthcare) with the 100× Olympus Objective. Samples were imaged in Z-stacks of 31 slices of 0.2 µm thickness. Images were analyzed using the FIJI image processing package of ImageJ (1.8.0)[68].

For the assessment of Golgi fragments in wild-type and *Naa60* knockout cell lines, the cells were seeded at a density of 30,000 cells per coverslip in 24-well plates. Two days after seeding, the cells were fixed using 4% PFA/PBS for 20 min at 4 °C. Cells were washed twice with PBS. Afterwards, the cells were permeabilized using the previously described staining buffer (PBS, BSA (2%), glycine (0.15%), and Triton X-100 (0.1%)) for 10 min and subsequently blocked using 2% BSA in PBS. The cells were incubated with the 1st primary antibody anti-GM130 (BD Transduction Laboratories Cat#610822) for 2 h at RT at a dilution of 1:100 in blocking solution. Coverslips were washed twice with PBS and subsequently incubated with the secondary antibody Goat anti-mouse IgG (H + L) Cross-Adsorbed Secondary Antibody Texas Red (Thermo Scientific Cat#T-862)) at a dilution of 1:2000 for 1 h at room temperature. After secondary incubation, the coverslips were washed 5 times with PBS. Next, the incubation step with the anti E-cadherin FITC-conj. (BD Biosciences Pharmingen Cat#612130) at a dilution of 1:500 in 2% BSA in PBS was performed for 2 h at RT. Subsequently, the coverslips were washed 5x with PBS, counterstained with DAPI and mounted onto positively charged Superfrost Plus Adhesion Microscope Slides (epredia Cat#J1800AMNZ) using fluorescence mounting medium (Dako Cat#S3023). Analysis was performed on the DeltaVision Elite High Resolution Microscope system (GE Healthcare) with the 100× Olympus Objective. Samples were imaged in Z-stacks of 29 slices of 0.2 µm thickness. Images were analyzed using the FIJI image processing package of ImageJ (1.8.0)[68]. For automated quantification of the Golgi fragments a CellProfiler pipeline was generated: First, the channels were split to separate image files. Next, the nuclei and Golgi fragments were identified as primary objects. As a next step, the cells were segmented based on the nuclear segmentation and the E-cadherin staining. Finally, the Golgi objects could be related back to each individual cell and quantified. Due to the chosen cutoffs and thresholds, this pipeline might underestimate the absolute number of Golgi fragments. Furthermore, fragments in line in the Z-axis might have also been projected on top of each other and therefore have been quantified as one unit.

## Proximity ligation assay

The proximity ligation assay was carried out according to the manufacturer's protocol. Briefly, 60,000 empty vector, single or double rescue cells were seeded on coverslips 1 day prior to the assay. Primary antibody incubation was performed as previously described under the immunofluorescence section for anti-HA and anti-Myc staining. The coverslips were washed with wash buffer A (0.01 M Tris-HCl, 0.15 M NaCl, and 0.05% Tween-20, pH 7.4) for 5 min each after overnight incubation with the primary antibodies Purified anti-HA.11 Epitope Tag antibody (clone 16B12) (BioLegend distributed by Luzernachem Cat#901501) and anti-Myc-Tag (71D10) rabbit mAb (Cell Signaling Cat#2278). The Duolink In Situ PLA probes anti-mouse plus (Sigma Cat#DUO92004-100RXN) and anti-rabbit minus (Sigma Cat#DUO92002-100RXN) were diluted 1:5 in the blocking solution (10% goat serum and 0.1% Triton X-100 in PBS), dispensed to slides (30 µl/coverslip) and incubated at 37 °C for 1 h. The coverslips were washed three times with buffer A, 5 min each. The ligation mix was prepared

by diluting the Duolink ligation stock (1:5) and ligase (1:40) in high purity water. A total of 30 μl/coverslip was applied to the samples and incubated at 37 °C for 30 min in a humid chamber. The coverslips were washed with buffer A twice for 2 min each. The amplification mix was prepared by diluting Duolink amplification stock (1:5) and rolling circle polymerase (1:80) in high-purity water. 30 μL/coverslip were applied to the samples and incubated for 100 min at 37 °C in a humid chamber. Subsequently, the coverslips were washed with wash buffer B solution (0.2 M Tris and 0.1 M NaCl) three times for 10 min each. The coverslips were now subjected to either additional staining using anti-GM130 Alexa Fluor 488 conjugated (BD Biosciences Cat#560257) for 2 h at RT at a dilution of 1:1000 in blocking solution to additionally visualize the Golgi apparatus, or secondary antibody staining against the primary antibodies used for the PLA (Goat anti-Mouse IgG (H + L) Cross-Adsorbed Secondary Antibody Alexa Fluor 488 (Thermo Scientific Cat#A11029) or Goat anti-Rabbit IgG (H + L) Highly Cross-Adsorbed Secondary Antibody, Alexa Fluor Plus 555, Thermo Scientific Cat#A32732) for 2 h at RT at a dilution of 1:1000 in blocking solution to visualize double-positive rescue cells. Coverslips were again washed with wash buffer B solution (0.2 M Tris and 0.1 M NaCl) three times for 5 min each and one time in 0.01x diluted wash buffer B solution for 1 min. The coverslips were mounted with ProLong Gold antifade reagent with DAPI (Invitrogen Cat#P36935). For samples with GM130 staining, the PLA reaction was performed using the Duolink detection reagent "red" (Duolink, Sigma Cat#DUO92008-100RXN). For the slides where a secondary antibody staining against the primary antibodies was performed, the Duolink detection reagent in "far red" (Duolink, Sigma Cat#DUO92013-100RXN) was used. The samples were stored at −20 °C until imaging with the DeltaVision Elite High Resolution Microscope system (GE Healthcare) with the 100× Olympus Objective. Cells were imaged on either the filter set 2 (FITC, mCherry, DAPI) for the GM130 double staining or filter set 1 (FITC, TRICT, Cy5, DAPI) for samples with additional secondary antibody staining. Images were acquired in Z-stacks of 26 slices of 0.2 μm thickness from different areas of the coverslips. Image processing steps such as a maximum projection and channel merging were carried out using the FIJI image processing package of ImageJ (1.8.0)[68].

### Cycloheximide chase assay

A total of 100,000 NAA60-deficient (*Naa60_24A4*, *Naa60_12A6*, *Naa60_13C3*) or proficient KB1PM5 cells (ntgB1, ntgC1) per condition were seeded to 60 mm diameter cell culture dishes (SPL Life Sciences Cat#20060). Two days after seeding, the cells were treated with 0.1 mg/ml cycloheximide (Sigma-Aldrich Cat#01810-1 G) for different durations (0 h, 4 h, 8 h, 20 h). Subsequently, the cells were harvested for Western Blotting as described previously under the Western Blotting section. A total of 40 μg of protein per sample were subjected to electrophoresis on a 7.5% acrylamide gel and subsequently transferred to 0.45 μM pore size PVDF membranes (GE Healthcare Cat#10600018). Membranes were blocked in 5% BSA in TBS-T (100 mM Tris, pH 7.5, 0.9% NaCl, 0.05% Tween-20) and subsequently incubated with primary antibodies anti-LRRC8A rabbit polyclonal (Bethyl Laboratories Cat#A304-175A), anti-LRRC8D rabbit polyclonal (Proteintech Cat#11537-1-AP), or anti-ubiquitin rabbit monoclonal (Thermo Scientific Cat#MA5-37950) diluted 1:1000 in blocking buffer at 4 °C overnight or anti-alpha tubulin mouse monoclonal (Sigma Cat#T5168) diluted 1:2000 in blocking buffer for 2 h at room temperature. After washing in TBS-T, Horseradish Peroxidase (HRP)-linked secondary antibodies anti-mouse IgG (Cell Signaling Cat#7076) or anti-rabbit IgG (Cell Signaling Cat#7074) at a dilution of 1:2500 were applied for 2 h at room temperature. Images were acquired using the FUSION FX7 imaging system (Vilber GmbH).

### Plasma membrane enrichment and proteomics analysis

For plasma membrane enrichment, 40 Mio NAA60-deficient (*Naa60_24A4*, *Naa60_12A6*, *Naa60_13C3*) or proficient KB1PM5 cells per sample were lysed with 50 μl per $1 \times 10^6$ cells in a hypotonic buffer (50 mM mannitol and 5 mM HEPES pH 7.4, and protease inhibitor cocktail) and homogenized by sonication (Bioruptor, Diagenode) on the high intensity setting for 20 cycles (15 s ON, 15 s OFF) in a water bath at 4 °C. The lysates were cleared by centrifugation at $600 \times g$ for 10 min at 4 °C. Subsequently, plasma membrane proteins were isolated according to a protocol published by Lin et al.[69] with some modifications. Calcium chloride solution of 1 M was added to the lysates to a final concentration of 10 mM and the suspension was vortexed vigorously for 10 min at RT. After centrifugation for 15 min at $3000 \times g$ and 4 °C, the supernatant was transferred to a new tube and ultra-centrifuged at $48,000 \times g$ and 4 °C for 30 min. The pellet was dissolved in 20 μL SDS-PAGE loading buffer and boiled for 5 min at 95 °C prior to loading on a 12.5% SDS-PAGE gel. The sample was developed into the gel for about 1.5 cm. Proteins were stained with Coomassie blue, and each lane was cut into five bands of equal size. Gel bands were cut into small cubes, which were transferred to 1.5 mL polypropylene reaction vials and wetted with 100 μL 20% ethanol for storage at 4 °C until digestion. For nLC-MS/MS analysis, the five slices of each sample were analyzed together, but samples were randomized in order to control sample processing bias. Proteins were in-gel digested, as described elsewhere[66]. An aliquot of 5 μL from each digest was analyzed on an instrumental setup consisting of an Ultimate 3000 nano-LC coupled to a LUMOS tribrid orbitrap mass spectrometer. Samples were loaded onto a pre-column (C18 PepMap 100, 5 μm, 100 A, 300 μm i.d. x 5 mm length) at a flow rate of 50 μL/min with solvent C (0.05% TFA in water/acetonitrile 98:2). After loading, peptides were eluted in back flush mode onto a self-pack C18 CSH Waters column (1.7 μm, 130 Å, 75 μm × 20 cm) using an acetonitrile gradient of 5% to 40% solvent B (0.1% Formic Acid in water/acetonitrile 4,9:95) in 40 min for in-gel digests at a flow rate of 250 nL/min. The column effluent was directly coupled to the Fusion LUMOS mass spectrometer via a nano-spray ESI source. Data acquisition was made in data-dependent mode with precursor ion scans recorded in the orbitrap with resolution of 120,000 (at $m/z = 250$) parallel to top speed fragment spectra of the most intense precursor ions in the linear trap for a cycle time of 3 s. For each cell line the plasma membrane protein isolation and subsequent mass spectrometry analysis was performed in triplicates. The mass spectrometry proteomics data have been deposited to the ProteomeXchange Consortium via the PRIDE[70] partner repository with the dataset identifier PXD035143.

Mass spectrometry data were processed with MaxQuant/Andromeda version 1.6.14.0 using default settings for peak detection, a strict trypsin cleavage rule, allowing up to 3 missed cleavages, variable oxidation on methionine, acetylation of protein N-termini, and deamidation of asparagine and glutamine, and fixed carbamidomethylation of cysteines, respectively. Match between runs was used with a retention time window of 0.7 min between neighboring gel fractions. The SwissProt mouse protein sequence database (version 2020_07) enriched by common contaminants was used to interpret fragment spectra with an initial mass tolerance of 10 ppm on precursor and 20 ppm for fragment ions, respectively. Protein identifications were accepted if at least two razor peptides per protein group were identified at a 1% false discovery rate (FDR) cutoff. Differential protein abundance testing was performed as described elsewhere[67].

To assess the quality of the membrane enrichment, we compared the representation of known plasma membrane and endoplasmic reticulum (ER) proteins identified in the enriched samples versus whole-cell lysates. While the relative proportion of ER proteins remained comparable between the two preparations, we noted that signal intensities for both plasma membrane and organellar membrane proteins, including those from the ER, were increased in the enriched fractions. This suggests that the protocol globally enriches membrane-associated proteins rather than exclusively plasma membrane components. As such, some level of ER co-enrichment cannot be excluded and should be considered a limitation of the approach.

### Co-localization of hLRRC8A GFP and CellMask™ plasma membrane stain

A total of 30,000 NAA60-proficient or deficient KB1P cells were seeded to coverslips in 24-well plates in triplicates. One day after seeding, the cells were

transfected with hLRRC8A-GFP expression vector (kindly provided by T. Jentsch) using the Turbofectin transfection reagent (Origene, LabForce Cat#TF81001) according to the manufacturer's protocol. Growth medium was refreshed after 24 h. Two days after transfection, the coverslips were incubated in CellMask™ deep red (Invitrogen Cat#C10046) staining solution diluted to 1x concentration in growth medium for 10 min at 37 °C. Afterwards, the cells were washed 1x with PBS and subsequently fixed with cold PFA for 20 min at 4 °C. After fixation, the coverslips were washed twice with PBS and mounted onto positively charged Superfrost Plus Adhesion Microscope Slides (epredia Cat#J1800AMNZ) using fluorescence mounting medium (Dako Cat#S3023). GFP expression was acquired on the FITC channel. The signal of the CellMask™ was acquired on the Cy5 channel of the DeltaVision Microscope using the 100 x Olympus Objective. Samples were imaged in Z-stacks of 20 slices of 0.2 µm thickness. Image processing steps such as a maximum projection and channel merging were carried out using the FIJI image processing package of ImageJ (1.8.0)[68]. The linear intensity profiles for each channel were generated by selecting a line across the plasma membrane as ROI, which was then analyzed by the Plot Profile plugin of ImageJ (2.14.0).

## Statistics and reproducibility
Sample sizes and numbers of replicates for each experiment are detailed in the respective figure legend or "Method" sections. Statistical analyses were conducted using the specified software tool, version, and settings mentioned in the respective method section.

## Reporting summary
Further information on research design is available in the Nature Portfolio Reporting Summary linked to this article.

## Data availability
All data will be shared upon request by the lead contact with no restrictions. Source data are provided with this paper. Raw sequences of the CRISPR/Cas9 screen with cisplatin are available in the European Nucleotide Archive (ENA) under the accession number PRJEB75036. The plasma membrane proteomics data are deposited in the ProteomeXchange Consortium via the PRIDE partner repository with the dataset identifier PXD035143. Uncropped Western Blots are shown as Supplementary Figs. 8 and 9. The remaining data are available within the Article or the Supplementary Information.

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

## Acknowledgements

We wish to thank G. Hayoz and F. Kölliker for their help at the Vetsuisse Faculty mouse facility. We would like to thank D. Howald and S. Franz for their technical support. We would also like to thank D. Chowdhury (Harvard Medical School, USA) for the pOZ-N-FH-IL2Rα and the pOZ-C-FH-IL2Rα

plasmids, and T.J. Jentsch (Leibniz-Forschungsinstitut für Molekulare Pharmakologie (FMP) and Max-Delbrück-Centrum für Molekulare Medizin, D-13125 Berlin, Germany) for providing the GFP-tagged LRRC8A construct. Moreover, we are grateful to D. Stroka, T. Brodie, and J. Iype from the Imaging Mass Cytometry and Mass Cytometry Platform (University of Bern) for their support with the CyTOF measurements. Finally, we would like to thank C. He, L. Lingg, M. Decollogny, M. Roorda, H. Hanzlikova, N. Aruna-salam, and P. Francica for the critical reading of the manuscript. Financial support came from the Swiss National Science Foundation (320030M_219453 to S.R. and J.J.), the European Research Council (ERC-2019-AdG-883877 to S.R.), the Swiss Cancer Research Foundation (KFS-5519-02-2022 to S.R.), the ISREC foundation, and the Office of the Assistant Secretary of Defense for Health Affairs through the Ovarian Cancer Research Program under Award No. (W81XWH-22-1-0557 to S.R.).

## Author contributions

C.A.W., J.J., and S.R. conceptualized the study and developed the methodology. C.A.W., A.M., I.K., and M.D. performed the experiments. M.G.F. and E.G. performed the analysis of the CRISPR/Cas9 screen. C.A.W., I.K., and M.S. performed the in vivo experiments. N.B., S.B.L., A.C.U., and M.H. performed the mass spectrometry measurements and analysis. C.A.W. and S.R. wrote the manuscript. All authors contributed to the final version of the manuscript.

## Competing interests

The authors declare no competing interests.
