## [Transparent Peer Review file · Communications Biology]

NAA60 facilitates LRRC8A- and LRRC8D-mediated platinum drug uptake

Corresponding Author: Professor Sven Rottenberg

Version 0:

Reviewer comments:

Reviewer #1

(Remarks to the Author)

The manuscript was evaluated by one PI with three early-career researchers as co-reviewers.

Brief summary of the manuscript / What are the major claims of the paper?

In this manuscript, Widmer et al used a CRISPR screen for platinum drug insensitivity where lack of Naa60 was identified to contribute to drug resistance. This screen finding was validated with 3 KO cell lines and mouse-implanted organoids. CyTOF measurements showed decreased drug uptake and accumulation of DNA damage in the absence of Naa60 in the cancer mouse cell model used. In vitro Nt-acetylation assay, substrate IP from WT vs Naa60 KO analysed by MS, as well as PLA was used to claim that Lrrc8a and Lrrc8d are substrates of Naa60-mediated Nt-acetylation. Since combined Naa60 and Lrrc8a re-expression was needed for the restoration of drug sensitivity in a Naa60/Lrrc8a double KO cell line, the authors concluded that Naa60 and Lrrc8a are epistatic. WB and proteomics were used to show that Lrrc8a overall protein level is unchanged, but that the protein is enriched at the plasma membrane. Introduction of a positive charge at the Lrrc8a and Lrrc8d N-termini was used to mimic loss of Nt-acetylation and these mutated plasmids could not rescue the drug sensitivity to the same degree as the WT plasmids.

Overall impression of the work

These findings are novel and will be of interest to researchers in the PTM community as well as to cancer drug (resistance) fields. There are not many papers on NAA60 and this work adds a highly interesting finding on the functionality of channel proteins in the absence of NAA60-mediated Nt-acetylation. The work also contributes to our understanding of how cancers may acquire drug insensitivity, and may thus have implications for future cancer treatment. The work is very neat and tidy, the story is very well told, and most of the work is convincing.

Specific comments, with recommendations for addressing each comment

General/overall

- 1) A graphical abstract or a schematic figure to summarize and contextualize the findings could have been nice.
- 2) Please make sure to include the most recent literature on NAA60.
- 3) We believe that correct nomenclature (gene/protein/species) has mostly been used throughout. NAA60 human gene, NAA60 human protein, Naa60 mouse gene, Naa60 mouse protein. But perhaps the authors can check this and make the distinction clear to the reader upon introduction? For example, please check Results chapter 3 and accompanying figure.

Abstract:

- 4) Line 22 "a N-terminal" -> "an N-terminal..."

Introduction

- 5) Line 49 (as well as overall): Change "N-acetyltransferase" to N-terminal acetyltransferase" (since N-acetyltransferase may also refer to other enzymes than NATs).
- 6) Line 58 (and overall): The commonly used abbreviations in the field are N-terminal acetyltransferase (NAT); N-terminal acetyltransferase B (NatB), N-alpha-acetyltransferase XX (NaaXX). Thus, it is suggested to change "NATB" and "NATF" to

“NatB” and “NatF”.

Results

7) Overall:

a) It is necessary to have a bit more methods information in the figure legends.
b) It is very nice that the authors show the internal clonal names as this would facilitate sharing with the scientific community. However, although the internal labels for clone names is a neat system, it challenges reader-friendliness and could therefore be considered simplified (perhaps in combination with a supplemental table listing the internal reference labels?). There could be one mistake in Ext. data fig 1b (left) where one of the Naa60 KO cell lines is labelled as “N16A6”, which perhaps should be “12A6”, and also without “N” for the two other clones in the figure to cohere with the labelling system overall.

8) General for WBs:

a) Replica blots are not shown in source data file, to be provided if possible.
b) kDA -> kDa
c) WBs have no mol weight marker, only approx. number, which is perhaps ok when there is only one band, but a bit problematic when more than one band (f.ex. Fig. 4).
d) Overall protein stain is preferable as loading control.
e) Was quantification (Fig. 5b+c) performed on blots exposed as in a? This is quite strong signal which in our experience can make quantification challenging.

9) General for figures:

a) The colours used for the graphs sometimes makes it challenging to tell the different lines apart.
b) It should be considered to make the microscopy images colour blind-friendly.

Fig 1

10) In the main text (line 73), the term “biological replicates” is used for a replica experiment. We believe that three samples from one cell line should be referred to as “three technical replicates” or “three independent experimental runs”. The latter would also convey information that it is an independent sample from cell seeding to DNA analysis. To clarify/exemplify our perception of the terms biological vs technical replicate: three different KO cell lines used are true biological replicates. This is a strength. In any case, this type of info could be reserved for the figure legends.

11) Could consider having a methods schematic figure to increase accessibility.

12) CRISPR:

a) It could be mentioned how the gRNA was selected/where it targets (in methods).
b) Are the sequencing results from verification of the KOs made available? Information on how the gene is affected (other than reduced size) is missing.
c) The KOs are relatively robustly verified. Ideally lack of protein by WB could be shown, but this may however be hindered by antibody availability for mouse Naa60. Alternatively, it could be possible to confirm that the disruption found (currently not specified) would affect Naa60 protein function, for example by using a prediction tool like for example mutation taster.

13) Fig 1 h+k could be a bit more clearly introduced as it is the only in vivo mouse experiment. Perhaps it could also improve accessibility if the authors were to add a sentence here that helps the reader understand the interpretation, such as “indicating that loss of Naa60 results in reduced efficacy of the Pt drugs and therefore faster progression of the tumor/poor survival of the mice”.

14) Please include an explanation/justification for fitting to a 4PL sigmoidal curve (Fig 1g+j).

Ext data fig 1

15) Phenotypic verification of Naa60 KOs by Golgi fragmentation seems valid and robustly assessed with an automated image analysis. The effect however seems a bit modest and the actual number of Golgi fragments per cell seem a bit underestimated (for both WT and KO). Therefore, one can wonder whether the analysis was a bit crude/superficial. This is however more than enough for a phenotypic confirmation, but perhaps the authors could add a caution sentence as to the absolute numbers of Golgi fragments per cell.

Ext data fig 2

16) In g, why is the ntgC1 not shown here?

Ext data fig 3

17) In Ext data fig 3a, the overview could be a bit clearer. Also, the time-point of imaging could be included.

Fig 3

18) In Fig 3a, the authors perform an in vitro N-terminal acetylation assay using IPed Naa60 in an acetylation mix with different peptides to be tested. Appropriate positive and negative peptide controls were used, which adds strength to the experiment. Traditionally, in the literature, this assay has been performed using radioactive labelled Ac-CoA. However, here MS was opted to detect Nt-acetylated vs un-Nt-acetylated peptides. We are unsure whether this readout works. It is a bit challenging to assess this experiment as a reviewer without access to raw/source MS data. We presume that the authors have performed LFQ MS, which makes accurate quantification Nt-Ac peptide vs un-Nt-Ac peptide problematic. Also, the bar chart/dot plots show the relative abundance, with a very modest increase for the NtAc-peptide in the presence of Naa60. Are these data statistically significant?

19) In Fig. 3c, the sequence of the peptides identified is not mentioned. It would be good if for reviewers could be able to access raw data.

20) In Fig. 3d, how was this normalized? The normalization strategy should be explained and justified.

- 21) In Fig. 3d, the Nt-acetylated peptide was only detected in one of three samples. That is a weakness, if the interpretation of a positive substrate relies on lacking detection of acetylated peptide in the Naa60 KO samples.
- 22) In Fig 3c vs 3d, two different approaches were used for isolating LRRC8A vs LRRC8D. The reason for this should be mentioned.
- 23) Fig 3e and Ext data fig. 5, the PLA experiments:
- Could preferably be quantified, perhaps with a similar strategy as used in Ext data fig 1, or with accompanying flow cytometry.
 - For the negative control with empty plasmid, it is important that both primary antibodies were used and full PLA protocol was performed. Is this the case? Furthermore, ideally, single transfections pOZ HA-Naa60 and pOZ Lrrc8a-Myc in two separate samples, each treated with both anti-HA and -Myc primaries and full PLA protocol would have been ideal. Can the authors please clarify that this is what is shown in Ext data fig. 5?
 - A co-transfected negative control protein could have been considered. However, it is difficult to know for sure what may be a suitable negative control with a similar subcellular localization, since binding partners of NAA60 has not been investigated. Nevertheless, if the interaction is assumed to occur because of Nt-acetylation reaction, it should then be dependent on the N-terminal sequence of Lrrc8a. The ideal control (in order to conclude a substrate-enzyme based interaction) would therefore be to mutate the Lrrc8a-Myc, to an N-terminus non-suitable for Nt-acetylation (f. ex. I2P) to use as a negative control.
 - To be able to make comparisons, it should ideally have been shown cells with similar expression levels and cell size/confluency.
 - It is stated that "The scale bars equal 20µm.", but two different sets of scalebar sizes are shown in the images.
 - It is not stated what cells were used in Ext. data fig. 5.
 - Lines 163-166 the authors state: "In the case of NAA60 and LRRC8A, these foci accumulate at the Golgi apparatus which we assessed by co-staining with a Golgi marker protein (GM130) (Fig.3e and Extended Data Fig.5). The formation of PLA foci was also observed for HA-NAA60 and LRRC8D-Myc. However, for LRRC8D a scattered localization pattern could be observed (Fig.3e and Extended Data Fig.5)." The accumulation of PLA signal at the Golgi seems to be true, but is however not robustly investigated here. Comparing Fig 3e and Ext. data fig. 5, to our impression, there is some variation in the degree of Golgi-closeness vs cytosol dispersion of the PLA signals of both LRRC8A and LRRC8D.

Fig 4

- 24) In Fig. 4 c-f, it would have been possible to better interpret the degree of rescue if treated WT cells had been included. Perhaps it is possible to point this out based on already existing data.
- 25) Line 176 in the main text: seems like the wrong figure reference has been inserted.

Fig 5

- 26) In Fig. 5c, a representative blot could be shown.
- 27) In Fig. 5d and e, data availability will be particularly important.
- 28) In Fig. 5d and e, was the success/quality of the PM isolation assessed? This should preferentially be shown. Perhaps the proteomics will demonstrate enrichment of PM proteins?
- 29) Line 197: In this plasma membrane proteomics, endogenous LRRC8A was detected in WT and Naa60 KO cells. Was the Nt-Ac checked here? If so, this could be a very nice confirmation of Fig. 3c.
- 30) In Fig. 5f, a rather simplified co-localization experiment is done. We are unsure whether it is possible to conclude with PM localization based on this. Very dense cell cultures are used, and it is impossible to judge whether the seemingly overlapping signal in the zoom-in frames indeed represents PM localization. Although CellMask is considered a PM stain, it relatively rapidly gets internalized by cells.

Fig 6

- 31) In Fig. 6 the authors functionally test LRRC8A and LRRC8D with a positively charged N-terminus to mimic the loss of Nt-acetylation. We question whether the mutations I2R and F2R in LRRC8A and LRRC8D, is indeed a good mimic of lacking N-terminal acetylation. The most frequently used approach to this strategy is to mutate to an N-terminus that is not compatible with N-terminal acetylation (X2P), to get a proline-starting N-terminus. The authors assume that the MR N-termini are Nt-acetylated by NAA60 as the WT proteins, however there is no grounds for believing that. MR N-termini has been understudied in the field due to protease digestion, and as such not many MR-starting proteins have been assessed for their Nt-acetylation status and no MR-starting proteins has been identified as NAA60 substrates. Thus, it is unknown whether the mutated N-termini will be Nt-acetylated (and by which NAT). Thus these N-termini may have a double positive charge (both from the arginine as well as from the alpha amino NH₃⁺). Either the Nt-acetylation status needs to be assessed, or the more conventional proline strategy used, or else it will be necessary to rephrase the claims concerning Fig. 6.
- 32) The plasmids with the mutated N-termini seem to partially provide the increased drug sensitivity, is the difference statistically significant?, could comment on this.
- 33) Fig. 6 m and n are not convincing.

Methods

- 34) Please add more information on how the MS data were analysed and normalized etc.

Reviewer #2

(Remarks to the Author)

In this paper Widmer and colleagues use in vitro and in vivo models to search for cellular determinants of sensitivity to platinum-based drugs. Earlier work of the lab has identified LRRC8A and LRRC8D, two volume-regulated anion channel subunits to mediate platinum uptake and hence cellular sensitivity. Here, a genome-wide CRISPR screen is used to reveal

NAA60, a N-terminal acetyltransferase to modulate LRRC8A and LRRC8D function. Convincing results are shown to demonstrate the regulation of LRRC8A/D function through N-terminal acetylation mediated by NAA60.

1. I am not sure if the full dataset showing the results of the CRISPR screen is released in the paper. I think it should be.
2. Is Pt uptake different in LRRC8A KO and NAA60 KO cells? If yes, what is the explanation? (Fig 2a)
3. What is the explanation of the staining pattern of LRRC8A/D?
4. What is the clinical relevance of these findings? Can the role of NAA60 in response to treatment or survival be ascertained? In the light of the answer to this question, is NAA60 a therapeutic target?
5. Can the role of N-terminal acetylation be also verified in whole-cell patch-clamp experiments, as performed by the authors in their 2015 EMBO Journal paper?

In summary, this is an excellent study, reporting novel findings on the functional association of NAA60 and LRRC8A/D. The paper is of interest to the wider community. Analysis of patient data would help to evaluate the clinical relevance of the results.

Reviewer #3

(Remarks to the Author)

Reviewer #4

(Remarks to the Author)

Reviewer #5

(Remarks to the Author)

The manuscript was evaluated by one PI with three early-career researchers as co-reviewers.

Brief summary of the manuscript / What are the major claims of the paper?

In this manuscript, Widmer et al used a CRISPR screen for platinum drug insensitivity where lack of Naa60 was identified to contribute to drug resistance. This screen finding was validated with 3 KO cell lines and mouse-implanted organoids. CyTOF measurements showed decreased drug uptake and accumulation of DNA damage in the absence of Naa60 in the cancer mouse cell model used. In vitro Nt-acetylation assay, substrate IP from WT vs Naa60 KO analysed by MS, as well as PLA was used to claim that Lrrc8a and Lrrc8d are substrates of Naa60-mediated Nt-acetylation. Since combined Naa60 and Lrrc8a re-expression was needed for the restoration of drug sensitivity in a Naa60/Lrrc8a double KO cell line, the authors concluded that Naa60 and Lrrc8a are epistatic. WB and proteomics were used to show that Lrrc8a overall protein level is unchanged, but that the protein is enriched at the plasma membrane. Introduction of a positive charge at the Lrrc8a and Lrrc8d N-termini was used to mimic loss of Nt-acetylation and these mutated plasmids could not rescue the drug sensitivity to the same degree as the WT plasmids.

Overall impression of the work

These findings are novel and will be of interest to researchers in the PTM community as well as to cancer drug (resistance) fields. There are not many papers on NAA60 and this work adds a highly interesting finding on the functionality of channel proteins in the absence of NAA60-mediated Nt-acetylation. The work also contributes to our understanding of how cancers may acquire drug insensitivity, and may thus have implications for future cancer treatment. The work is very neat and tidy, the story is very well told, and most of the work is convincing.

Specific comments, with recommendations for addressing each comment

General/overall

- 1) A graphical abstract or a schematic figure to summarize and contextualize the findings could have been nice.
- 2) Please make sure to include the most recent literature on NAA60.
- 3) We believe that correct nomenclature (gene/protein/species) has mostly been used throughout. NAA60 human gene, NAA60 human protein, Naa60 mouse gene, Naa60 mouse protein. But perhaps the authors can check this and make the distinction clear to the reader upon introduction? For example, please check Results chapter 3 and accompanying figure.

Abstract:

- 4) Line 22 "a N-terminal" -> "an N-terminal..."

Introduction

- 5) Line 49 (as well as overall): Change "N-acetyltransferase" to "N-terminal acetyltransferase" (since N-acetyltransferase may also refer to other enzymes than NATs).
- 6) Line 58 (and overall): The commonly used abbreviations in the field are N-terminal acetyltransferase (NAT); N-terminal acetyltransferase B (NatB), N-alpha-acetyltransferase XX (NaaXX). Thus, it is suggested to change "NATB" and "NATF" to "NatB" and "NatF".

Results

7) Overall:

a) It is necessary to have a bit more methods information in the figure legends.
b) It is very nice that the authors show the internal clonal names as this would facilitate sharing with the scientific community. However, although the internal labels for clone names is a neat system, it challenges reader-friendliness and could therefore be considered simplified (perhaps in combination with a supplemental table listing the internal reference labels?). There could be one mistake in Ext. data fig 1b (left) where one of the Naa60 KO cell lines is labelled as "N16A6", which perhaps should be "12A6", and also without "N" for the two other clones in the figure to cohere with the labelling system overall.

8) General for WBs:

a) Replica blots are not shown in source data file, to be provided if possible.
b) kDa → kDa
c) WBs have no mol weight marker, only approx. number, which is perhaps ok when there is only one band, but a bit problematic when more than one band (f.ex. Fig. 4).
d) Overall protein stain is preferable as loading control.
e) Was quantification (Fig. 5b+c) performed on blots exposed as in a? This is quite strong signal which in our experience can make quantification challenging.

9) General for figures:

a) The colours used for the graphs sometimes makes it challenging to tell the different lines apart.
b) It should be considered to make the microscopy images colour blind-friendly.

Fig 1

10) In the main text (line 73), the term "biological replicates" is used for a replica experiment. We believe that three samples from one cell line should be referred to as "three technical replicates" or "three independent experimental runs". The latter would also convey information that it is an independent sample from cell seeding to DNA analysis. To clarify/exemplify our perception of the terms biological vs technical replicate: three different KO cell lines used are true biological replicates. This is a strength. In any case, this type of info could be reserved for the figure legends.

11) Could consider having a methods schematic figure to increase accessibility.

12) CRISPR:

a) It could be mentioned how the gRNA was selected/where it targets (in methods).
b) Are the sequencing results from verification of the KOs made available? Information on how the gene is affected (other than reduced size) is missing.
c) The KOs are relatively robustly verified. Ideally lack of protein by WB could be shown, but this may however be hindered by antibody availability for mouse Naa60. Alternatively, it could be possible to confirm that the disruption found (currently not specified) would affect Naa60 protein function, for example by using a prediction tool like for example mutation taster.

13) Fig 1 h+k could be a bit more clearly introduced as it is the only in vivo mouse experiment. Perhaps it could also improve accessibility if the authors were to add a sentence here that helps the reader understand the interpretation, such as "indicating that loss of Naa60 results in reduced efficacy of the Pt drugs and therefore faster progression of the tumor/poor survival of the mice".

14) Please include an explanation/justification for fitting to a 4PL sigmoidal curve (Fig 1g+j).

Ext data fig 1

15) Phenotypic verification of Naa60 KOs by Golgi fragmentation seems valid and robustly assessed with an automated image analysis. The effect however seems a bit modest and the actual number of Golgi fragments per cell seem a bit underestimated (for both WT and KO). Therefore, one can wonder whether the analysis was a bit crude/superficial. This is however more than enough for a phenotypic confirmation, but perhaps the authors could add a caution sentence as to the absolute numbers of Golgi fragments per cell.

Ext data fig 2

16) In g, why is the ntgC1 not shown here?

Ext data fig 3

17) In Ext data fig 3a, the overview could be a bit clearer. Also, the time-point of imaging could be included.

Fig 3

18) In Fig 3a, the authors perform an in vitro N-terminal acetylation assay using IPed Naa60 in an acetylation mix with different peptides to be tested. Appropriate positive and negative peptide controls were used, which adds strength to the experiment. Traditionally, in the literature, this assay has been performed using radioactive labelled Ac-CoA. However, here MS was opted to detect Nt-acetylated vs un-Nt-acetylated peptides. We are unsure whether this readout works. It is a bit challenging to assess this experiment as a reviewer without access to raw/source MS data. We presume that the authors have performed LFQ MS, which makes accurate quantification Nt-Ac peptide vs un-Nt-Ac peptide problematic. Also, the bar chart/dot plots show the relative abundance, with a very modest increase for the NtAc-peptide in the presence of Naa60. Are these data statistically significant?

19) In Fig. 3c, the sequence of the peptides identified is not mentioned. It would be good if for reviewers could be able to access raw data.

20) In Fig. 3d, how was this normalized? The normalization strategy should be explained and justified.

21) In Fig. 3d, the Nt-acetylated peptide was only detected in one of three samples. That is a weakness, if the interpretation of a positive substrate relies on lacking detection of acetylated peptide in the Naa60 KO samples.

22) In Fig 3c vs 3d, two different approaches were used for isolating LRRC8A vs LRRC8D. The reason for this should be mentioned.

23) Fig 3e and Ext data fig. 5, the PLA experiments:

- a) Could preferably be quantified, perhaps with a similar strategy as used in Ext data fig 1, or with accompanying flow cytometry.
- b) For the negative control with empty plasmid, it is important that both primary antibodies were used and full PLA protocol was performed. Is this the case? Furthermore, ideally, single transfections pOZ HA-Naa60 and pOZ Lrrc8a-Myc in two separate samples, each treated with both anti-HA and -Myc primaries and full PLA protocol would have been ideal. Can the authors please clarify that this is what is shown in Ext data fig. 5?
- c) A co-transfected negative control protein could have been considered. However, it is difficult to know for sure what may be a suitable negative control with a similar subcellular localization, since binding partners of NAA60 has not been investigated. Nevertheless, if the interaction is assumed to occur because of Nt-acetylation reaction, it should then be dependent on the N-terminal sequence of Lrrc8a. The ideal control (in order to conclude a substrate-enzyme based interaction) would therefore be to mutate the Lrrc8a-Myc, to an N-terminus non-suitable for Nt-acetylation (f. ex. I2P) to use as a negative control.
- d) To be able to make comparisons, it should ideally have been shown cells with similar expression levels and cell size/confluency.
- e) It is stated that "The scale bars equal 20µm.", but two different sets of scalebar sizes are shown in the images.
- f) It is not stated what cells were used in Ext. data fig. 5.
- g) Lines 163-166 the authors state: "In the case of NAA60 and LRRC8A, these foci accumulate at the Golgi apparatus which we assessed by co-staining with a Golgi marker protein (GM130) (Fig.3e and Extended Data Fig.5). The formation of PLA foci was also observed for HA-NAA60 and LRRC8D-Myc. However, for LRRC8D a scattered localization pattern could be observed (Fig.3e and Extended Data Fig.5)." The accumulation of PLA signal at the Golgi seems to be true, but is however not robustly investigated here. Comparing Fig 3e and Ext. data fig. 5, to our impression, there is some variation in the degree of Golgi-closeness vs cytosol dispersion of the PLA signals of both LRRC8A and LRRC8D.

Fig 4

- 24) In Fig. 4 c-f, it would have been possible to better interpret the degree of rescue if treated WT cells had been included. Perhaps it is possible to point this out based on already existing data.
- 25) Line 176 in the main text: seems like the wrong figure reference has been inserted.

Fig 5

- 26) In Fig. 5c, a representative blot could be shown.
- 27) In Fig. 5d and e, data availability will be particularly important.
- 28) In Fig. 5d and e, was the success/quality of the PM isolation assessed? This should preferentially be shown. Perhaps the proteomics will demonstrate enrichment of PM proteins?
- 29) Line 197: In this plasma membrane proteomics, endogenous LRRC8A was detected in WT and Naa60 KO cells. Was the Nt-Ac checked here? If so, this could be a very nice confirmation of Fig. 3c.
- 30) In Fig. 5f, a rather simplified co-localization experiment is done. We are unsure whether it is possible to conclude with PM localization based on this. Very dense cell cultures are used, and it is impossible to judge whether the seemingly overlapping signal in the zoom-in frames indeed represents PM localization. Although CellMask is considered a PM stain, it relatively rapidly gets internalized by cells.

Fig 6

- 31) In Fig. 6 the authors functionally test LRRC8A and LRRC8D with a positively charged N-terminus to mimic the loss of Nt-acetylation. We question whether the mutations I2R and F2R in LRRC8A and LRRC8D, is indeed a good mimic of lacking N-terminal acetylation. The most frequently used approach to this strategy is to mutate to an N-terminus that is not compatible with N-terminal acetylation (X2P), to get a proline-starting N-terminus. The authors assume that the MR N-termini are Nt-acetylated by NAA60 as the WT proteins, however there is no grounds for believing that. MR N-termini has been understudied in the field due to protease digestion, and as such not many MR-starting proteins have been assessed for their Nt-acetylation status and no MR-starting proteins has been identified as NAA60 substrates. Thus, it is unknown whether the mutated N-termini will be Nt-acetylated (and by which NAT). Thus these N-termini may have a double positive charge (both from the arginine as well as from the alpha amino NH₃⁺). Either the Nt-acetylation status needs to be assessed, or the more conventional proline strategy used, or else it will be necessary to rephrase the claims concerning Fig. 6.
- 32) The plasmids with the mutated N-termini seem to partially provide the increased drug sensitivity, is the difference statistically significant?, could comment on this.
- 33) Fig. 6 m and n are not convincing.

Methods

- 34) Please add more information on how the MS data were analysed and normalized etc.

Version 1:

Reviewer comments:

Reviewer #1

(Remarks to the Author)

Overall, the authors have come back with several good answers to our concerns. And our suggestions for minor

improvements have been followed. However, some of the more severe comments on data quality was not met. While the authors present solid evidence that NAA60 is necessary for effective platinum drug uptake, we still (as no additional experiments were performed) have concerns about the data used to claim that the effect is mediated via NAA60's function as an Nt-acetyl transferase. Although it may appear obvious that lack of Nt-acetylation must indeed be at play, it is important to acknowledge that for many of the NATs moonlighting functions are being revealed. Therefore, weak evidence is not enough to claim that the Pt drug effect is dependent on NAA60's Nt-acetylation. Specifically:

Comment 18 on Fig. 3a:

The values are VERY low. What are the raw intensity data here? Raw data was not shared as requested. We would like to see the numbers before the relative normalization. Data in 3a is not significant and hence does not show increased Nt-acetylation in the presence of Naa60. It is not possible to make any claims/statements/conclusions about Nt-acetylation status based on these experiments.

Comment 19 on Fig. 3c:

We had concerns and asked to see raw data. This could be an excel file of the identified peptides and their intensities.

Comments 20-21 on Fig. 3d:

Raw data available? The y axis label should be explained. Why were not raw intensities used like in c? While more methodological details for the MS was added to the manuscript, info about data normalization was not added. It should be transparent how that data were normalized/transformed from raw values.

Comment 23 on Fig 3e and Ext data fig. 5, the PLA experiments:

The text should convey a bit more caution when interpreting the results and be open about possible variations due to expression levels and so on with transfection.

Comment 30 concerning Fig. 5f:

While it sounds like the authors have used the CellMask stain correct, our comment concerning the difficulties in interpreting these results was not responded to. We have a hard time accepting this as evidence for PM localization and it is definitely not possible to conclude with increased PM localization in the absence of Naa60. We understand that microscopy of PM localization is difficult, however the current data does not demonstrate what the authors claim it to demonstrate. We see that a 100 x objective was used and that samples were imaged in Z-stacks of 20 slices of 0.2 μ m thickness. This should allow for a 3D analysis in order to measure plasma membrane localization rather than simply showing the images as maximum projection. The images shown do not support the claim.

Comment 31, concerning Fig. 6:

While the authors have added a precautionary sentence to the manuscript, their claims made in figure headings, abstract, title of the manuscript etc are unchanged and as explained we do not find the evidence concerning Nt-acetylation to be sufficient. Taken together, the data on Nt-acetylation (Figure 3 and Figure 6) all have flaws/weaknesses, making it overall insufficient to conclude that the Pt drug uptake effect observed in the absence of NAA60 is explained by loss of Nt-acetylation.

Reviewer #2

(Remarks to the Author)

The authors did an excellent job in addressing the extensive comments of the reviewers. Although my comments and suggestions did not result in any changes in the revised manuscript, I find the manuscript ready to be published. Publication of the reviewer comments and the author rebuttal letters in a supplementary file is strongly recommended.

Version 2:

Reviewer comments:

Reviewer #1

(Remarks to the Author)

Submitted as PDF file.

Referee expertise:

Referee #1: N-terminal acetylation, molecular biology, cell biology

Referee #2: drug resistance

Reviewers' comments:

Reviewer #1 (Remarks to the Author):

The manuscript was evaluated by one PI with three early-career researchers as co-reviewers.

Brief summary of the manuscript / What are the major claims of the paper? In this manuscript, Widmer et al used a CRISPR screen for platinum drug insensitivity where lack of Naa60 was identified to contribute to drug resistance. This screen finding was validated with 3 KO cell lines and mouse-implanted organoids. CyTOF measurements showed decreased drug uptake and accumulation of DNA damage in the absence of Naa60 in the cancer mouse cell model used. In vitro Nt-acetylation assay, substrate IP from WT vs Naa60 KO analysed by MS, as well as PLA was used to claim that Lrrc8a and Lrrc8d are substrates of Naa60-mediated Nt-acetylation. Since combined Naa60 and Lrrc8a re-expression was needed for the restoration of drug sensitivity in a Naa60/Lrrc8a double KO cell line, the authors concluded that Naa60 and Lrrc8a are epistatic. WB and proteomics were used to show that Lrrc8a overall protein level is unchanged, but that the protein is enriched at the plasma membrane. Introduction of a positive charge at the Lrrc8a and Lrrc8d N-termini was used to mimic loss of Nt-acetylation and these mutated plasmids could not rescue the drug sensitivity to the same degree as the WT plasmids.

Overall impression of the work

These findings are novel and will be of interest to researchers in the PTM community as well as to cancer drug (resistance) fields. There are not many papers on NAA60 and this work adds a highly interesting finding on the functionality of channel proteins in the absence of NAA60-mediated Nt-acetylation. The work also contributes to our understanding of how cancers may acquire drug insensitivity, and may thus have implications for future cancer treatment. The work is very neat and tidy, the story is very well told, and most of the work is convincing.

Answer: We thank the reviewer (actually all 4 reviewers!) for their thorough and thoughtful evaluation of our manuscript. We appreciate the positive feedback on the novelty and significance of our findings, as well as the constructive comments provided. In particular, we greatly appreciate the reviewer's positive assessment of the manuscript's impact and the quality of the work. We have strived to present a clear and compelling narrative, supported by rigorous experimentation. The recognition of the

manuscript's potential relevance to both the PTM and cancer research communities is particularly gratifying.

Specific comments, with recommendations for addressing each comment

General/overall

1) A graphical abstract or a schematic figure to summarize and contextualize the findings could have been nice.

Answer: We thank the reviewer for this suggestion. A new figure 7 summarizing the findings has been added. This figure could also be used as graphical abstract.

2) Please make sure to include the most recent literature on NAA60.

Answer: We thank the reviewer for this reminder. We are not sure whether the reviewer had a specific publication in mind, but we think a very interesting and recent publication is the one by Chelban *et al.* (new reference 43)¹, which discusses the enzyme/substrate relationship of NAA60 and the phosphate importer SLC20A2. We have added the citation of this publication in *lines 202-205*:

"As LRRC8A and LRRC8D exhibit their role for the VRAC at the plasma membrane, we considered that the NAA60-mediated acetylation might be important for the trafficking of LRRC8A/D to the plasma membrane. This was recently shown for the phosphate importer SLC20A2: biallelic deletions of NAA60 were found to cause primary familial brain calcification, because the lack of the NAA60-specific Nt acetylation activity results in decreased SLC20A2 levels at the plasma membrane⁴³."

3) We believe that correct nomenclature (gene/protein/species) has mostly been used throughout. *NAA60* human gene, NAA60 human protein, *Naa60* mouse gene, Naa60 mouse protein. But perhaps the authors can check this and make the distinction clear to the reader upon introduction? For example, please check Results chapter 3 and accompanying figure.

Answer: Thanks for checking. We have used a slightly different nomenclature. According to the MGI (<https://www.informatics.jax.org/mgihome/nomen/gene.shtml>) mouse proteins should be all upper case, therefore NAA60 would be the correct name for both the human and mouse protein. In case the editor thinks that we should change our nomenclature, we are happy to adapt.

Abstract:

4) Line 22 "a N-terminal" -> "an N-terminal..."

Answer: Corrected.

Introduction

5) Line 49 (as well as overall): Change "N-acetyltransferase" to "N-terminal acetyltransferase" (since N-acetyltransferase may also refer to other enzymes than NATs).

Answer: Corrected.

6) Line 58 (and overall): The commonly used abbreviations in the field are N-terminal acetyltransferase

(NAT); N-terminal acetyltransferase B (NatB), N-alpha-acetyltransferase XX (NaaXX). Thus, it is suggested to change "NATB" and "NATF" to "NatB" and "NatF".

Answer: Corrected.

Results

7) Overall:

a) It is necessary to have a bit more methods information in the figure legends.

Answer: We appreciate the reviewer's suggestion to include more methodological details in the figure legends. Our intention was to provide a balance between offering sufficient information to understand the figures and avoiding overloading the legends with extensive details that are covered in the methods section. We aimed to keep the figure legends concise while ensuring that relevant methodological information was conveyed clearly. To address this, we have added additional information as detailed in our responses below (e.g. adding a new Fig. 1a to provide more details on our methods) and we now indicate drug treatment durations in the figure legends, where applicable. In case additional clarifications are needed, we are open to making more adaptations to the figure legends as needed.

b) It is very nice that the authors show the internal clonal names as this would facilitate sharing with the scientific community. However, although the internal labels for clone names is a neat system, it challenges reader-friendliness and could therefore be considered simplified (perhaps in combination with a supplemental table listing the internal reference labels?). There could be one mistake in Ext. data fig 1b (left) where one of the Naa60 KO cell lines is labelled as "N16A6", which perhaps should be "12A6", and also without "N" for the two other clones in the figure to cohere with the labelling system overall.

Answer: Corrected.

8) General for WBs:

a) Replica blots are not shown in source data file, to be provided if possible.

Answer: Replicate blots for the quantification in Figure 5 a and b are now included in the source data file.

b) kDA -> kDa

Answer: Corrected, thanks!

c) WBs have no mol weight marker, only approx. number, which is perhaps ok when there is only one band, but a bit problematic when more than one band (f.ex. Fig. 4).

Answer: The images including the molecular weight marker overlap are shown in the source data for the Western Blots, but the reviewers are correct that the marker band sizes were not labelled. In the revised manuscript, we have now added the corresponding marker band sizes in the source data files.

Regarding Figure 4a, we have observed that the rescue product of C-terminally Myc tagged LRRC8A often produces three bands. The vector was checked for the correct sequence and whether there was a mix of constructs present, which was not the case. For improved interpretation of the results, we have added the position of the molecular weight marker bands to all the Western Blot images.

d) Overall protein stain is preferable as loading control.

Answer: Indeed, a Ponceau S staining was performed of each membrane prior to blocking and subsequent antibody incubation as additional verification of equal protein loading, migration pattern and absence of any bubbles or problems regarding the transfer. In our revised manuscript, we have now included the Ponceau images into the source data files of the uncropped Western Blots.

e) Was quantification (Fig. 5b+c) performed on blots exposed as in a? This is quite strong signal which in our experience can make quantification challenging.

Answer: Yes. We used the gel casting units of 1.5 mm thickness, therefore the amount of protein transferred per area is quite high, resulting in thicker rounder bands. For the tubulin staining the lowest exposure possible was used. The machine did not indicate oversaturation of the detector during imaging.

9) General for figures:

a) The colours used for the graphs sometimes makes it challenging to tell the different lines apart.

Answer: We thank the reviewer for pointing out the potential issue with color differentiation in the graphs. We aimed to use colors that would be both visually appealing and distinct; however, we understand that in some cases, the color choices may make it difficult to distinguish between different lines or data points. Please see our response to b) for the specific changes that we added.

b) It should be considered to make the microscopy images colour blind-friendly.

Answer: For improved contrast and visibility, the red color on a black background was exchanged to magenta. Generally, the red/green color combination was avoided. We hope these changes improve the accessibility of our study.

Fig 1

10) In the main text (line 73), the term “biological replicates” is used for a replica experiment. We believe that three samples from one cell line should be referred to as “three technical replicates” or “three independent experimental runs”. The latter would also convey information that it is an independent sample from cell seeding to DNA analysis. To clarify/exemplify our perception of the terms biological vs technical replicate: three different KO cell lines used are true biological replicates. This is a strength. In any case, this type of info could be reserved for the figure legends.

Answer: Thank you very much for pointing this out. We agree with the reviewer that replicate experiments using the same knockout cell line should be generally considered as “technical replicates”. In the context of the screen results that we present, however, three independent transductions of the parental cell line were performed to yield the three replicate results, which we therefore termed to be “biological”. To clarify, the following sentence was added to the method section describing the CRISPR screen: “For each replicate an independent transduction of the parental cell line was performed.” –
Line 331-332

11) Could consider having a methods schematic figure to increase accessibility.

Answer: A schematic overview of the screen procedure was added to Figure 1 (new Figure 1a). The labelling of the remaining graphs therefore was shifted by one letter, which was adjusted in the text body and figure legend as well.

12) CRISPR:

a) It could be mentioned how the gRNA was selected/where it targets (in methods).

Answer: We apologize for omitting this information in the methods section. In our revised manuscript, we have added the following information to the “Genome editing, plasmids, and cloning” section:

“The gRNAs were designed to target exon 4 and 5 of *Naa60* (amino acids 81-191), which encode for functional domains such as the GNAT fold including the Ac-CoA binding site (amino acids 108-113) of the NAA60 protein: the big deletion caused by gRNAs 1+2 removes the Ac-CoA binding site, the deletion caused by gRNAs 1+3 additionally includes the amino acid His 138, which was shown to be important for hNAA60 function.² Furthermore, the deletions caused by gRNA 1+3 and 2+4 encompass Leu 140, Asn 143, Tyr 165, and Ile 167 which are important for Met1 recognition on the substrate peptide, substrate binding, and peptide anchoring. Mutations of these amino acids have been shown to fully abrogate or greatly reduce protein function³.” – *Line 387-395*

b) Are the sequencing results from verification of the KOs made available? Information on how the gene is affected (other than reduced size) is missing.

Answer: The homozygous deletion has been validated by Sanger sequencing of the genomic locus for the three knockout cell lines. Figure 1 below shows how the deleted nucleotides affect the translation in the three different knockout cell lines:

The amino acid sequences are affected in the following way: (Intron between exon 4 and 5: 674)

Naa60_24A4, deletion of coding nucleotides (no intron in between gRNA cut sites): 119 nucleotides, deletion of amino acids 129-167 + 2 nucleotides, likely leading to frameshift throughout the remaining gene sequence.

Naa60_12A6, deletion: 761, deletion of coding nucleotides: 87, deletion of amino acids 99-127, *p.A128P* mutation

Naa60_13C3, deletion: 871, deletion of coding nucleotides: 197, deletion of amino acids 99-163 + two nucleotides deleted leading to a frameshift mutation for the rest of the gene from amino acid position 99.

In our revised manuscript, we have added the following sentence referring to Figure 1 in the results section: “The monoclonal *Naa60* knockout cell lines harbor big deletions in exons 4 and 5 (*Naa60_12A6*, 87 coding nucleotides deletion), which encode for functional domains of the NAA60 protein. The deletions of *Naa60_24A4* (119 coding nucleotides deleted) and *Naa60_13C3* (197 coding nucleotides deleted) additionally lead to a frame shift.” – *Line 82-85*

Furthermore, the PCR product band sizes of Fig.1d were updated to the actual PCR product sizes according to the sequencing results, rather than the hypothetical band sizes expected from the gRNA cut sites.

c) The KO lines are relatively robustly verified. Ideally lack of protein by WB could be shown, but this may however be hindered by antibody availability for mouse Naa60. Alternatively, it could be possible to confirm that the disruption found (currently not specified) would affect Naa60 protein function, for example by using a prediction tool like for example mutation taster.

Answer: Indeed, commercially available antibodies do not detect mouse NAA60, and our attempts to raise new antibodies were unsuccessful thus far. We also tried to enrich for Golgi organelles utilizing a subcellular fractionation approach, but unfortunately, no specific band could be identified (testing wild-type, knockout and rescue cell lines) at the corresponding kDa weight. However, as the reviewer wrote, we think that our KO lines are robustly verified. The mutations are confirmed at the DNA level, and due to the nature of the deletions it is extremely unlikely that there is any functional protein product for *Naa60_24A4*, *Naa60_13C3* or *Naa60_12A6* (see Figure 1 for the reviewers). Moreover, we have the functional validation of the knockout with the Golgi fragmentations and the functional restoration of our phenotypes when we add back the *Naa60* cDNA.

```

Page 1 Naa60-211 (ENSMUST00000186375) [3690239-3722634]_Naa60_24A4 translation [3712239-3719...

      1                               82
NP_083366... MTEVVPSSALSSEVSLRLLCHDDIDTVKHLGCGDFPIEYDPSWYRDITSNKKFFSLAATYRGAIVGMIVAEIKNRTKHKEDG
Naa60_24A... MTEVVPSSALSSEVSLRLLCHDDIDTVKHLGCGDFPIEYDPSWYRDITSNKKFFSLAATYRGAIVGMIVAEIKNRTKHKEDG
Naa60_12A... MTEVVPSSALSSEVSLRLLCHDDIDTVKHLGCGDFPIEYDPSWYRDITSNKKFFSLAATYRGAIVGMIVAEIKNRTKHKEDG
Naa60_13C... MTEVVPSSALSSEVSLRLLCHDDIDTVKHLGCGDFPIEYDPSWYRDITSNKKFFSLAATYRGAIVGMIVAEIKNRTKHKEDG
.....

      83                               164
NP_083366... DILASSFSVDTQVAYILSLGVVKEFRKHGIGSLLLESKDHISTTAQDHCKAIYLHVLTTNN-TAINFYENRDFRQHHYLPY
Naa60_24A... DILASSFSVDTQVAYILSLGVVKEFRKHGIGSLLLESKDHISTTARGPRWLHLCPLHQRPPSLD-----HFRLHPA---
Naa60_12A... DILASSFSVDTQVAYI-----PDHCKAIYLHVLTTNN-TAINFYENRDFRQHHYLPY
Naa60_13C... DILASSFSVDTQVAYILL-----HSRGPQRWLHLCPLHQRPPSLD-----HFRLHPA---
.....

      165                               246
NP_083366... YYSIRGVLKDGFTYVL-----YINGGHPPTILDYIQHLGSALANLSPCSIPHRIYRQAHSLLCSFLPWSSISTKGGIEYSR
Naa60_24A... ----PGLSTSQPEPILHPTQDLPPGPPALQLSAMVQHPHQRWHRVQPYHV-----
Naa60_12A... YYSIRGVLKDGFTYVL-----YINGGHPPTILDYIQHLGSALANLSPCSIPHRIYRQAHSLLCSFLPWSSISTKGGIEYSR
Naa60_13C... ----PGLSTSQPEPILHPTQDLPPGPPALQLSAMVQHPHQRWHRVQPYHV-----
.....

      247
NP_083366... TM
Naa60_24A... --
Naa60_12A... TM
Naa60_13C... --

```

Figure 1 for the reviewers. Amino acid sequence alignment of the Naa60 knockout lines used in this study. The amino acid sequences were translated from sequences obtained by Sanger sequencing using benchling.com and aligned to the wild-type NAA60 protein sequence (NP_083366.1. N-alpha-acetyltransferase 60 isoform 1 [Mus musculus]). In *Naa60_24A4* and *Naa60_13C3*, due to the nature of the deletion, a frameshift occurs, which greatly alters the translated amino acid sequence. In *Naa60_12A6* a deletion of 29 amino acids and an amino acid

13) Fig 1 h+k could be a bit more clearly introduced as it is the only in vivo mouse experiment. Perhaps it could also improve accessibility if the authors were to add a sentence here that helps the reader understand the interpretation, such as “indicating that loss of Naa60 results in reduced efficacy of the Pt drugs and therefore faster progression of the tumor/poor survival of the mice”.

Answer: Thank you for pointing this out. We now introduce the *in vivo* experiment by “To study the lack of *Naa60* expression in the *in vivo* context, ...”- *Line 95*

For improved interpretation of the results, we have added the sentence “..., indicating that loss of NAA60 results in reduced efficacy of the Pt-based agents cisplatin and carboplatin and therefore faster progression of the tumor followed by poor survival of the mice.” – *Line 100-102*

14) Please include an explanation/justification for fitting to a 4PL sigmoidal curve (Fig 1g+j).

Answer: Thanks for this feedback. We realized that the curve fitted is a non-linear regression dose-response curve log (inhibitor) vs. **normalized** response – Variable slope and not the log (inhibitor) vs. response – variable slope (four parameters) one. We have now corrected this mistake in all the figure legends accordingly.

In the revised manuscript, we have added the following sentence to the methods section for the Clonogenic assays: “The data were subsequently fitted to a non-linear regression dose-response curve (log(inhibitor) vs. normalized response -Variable slope) in GraphPad Prism 9.”-*Line 442-443*

Ext data fig 1

15) Phenotypic verification of *Naa60* KOs by Golgi fragmentation seems valid and robustly assessed with an automated image analysis. The effect however seems a bit modest and the actual number of Golgi fragments per cell seem a bit underestimated (for both WT and KO). Therefore, one can wonder whether the analysis was a bit crude/superficial. This is however more than enough for a phenotypic confirmation, but perhaps the authors could add a caution sentence as to the absolute numbers of Golgi fragments per cell.

Answer: In our revised manuscript, we have added the following sentence regarding the limitation of our quantification in the methods section for the Golgi fragments analysis: “Due to the chosen cutoffs and thresholds, this pipeline might underestimate the absolute number of Golgi fragments. Furthermore, fragments in line in the Z-axis might have also been projected on top of each other and therefore have been quantified as one unit.” – *Line 680-681*

Additionally, we realized that the methods section for the assessment of the Golgi fragments has been missing from the main method section. We have added this information (*Line – 657-683*)

Ext data fig 2

16) In g, why is the ntgC1 not shown here?

Answer: The ntgC1+ pOZ empty dataset was left out of the graphs of Ext data fig 2c,d,e,g, because the number of datasets within these graphs was already quite high and we felt that the readability might have been compromised by adding more datasets. For the revised version we have added the ntgC1 + pOZ empty dataset back into the graphs.

Ext data fig 3

17) In Ext data fig 3a, the overview could be a bit clearer. Also, the time-point of imaging could be included.

Answer: Thanks for pointing this out. The timepoint of imaging was at day 14 right before the CellTiter-Blue assay. We have added the information to the timeline as well as the figure legend. Furthermore, the timeline was exchanged for a more detailed version.

The following sentence was added to the figure legend: “Outline of the organoids experiment which included two cycles of drug treatment on day 1 and day 7 for 24 hours each, afterwards the medium was changed to fresh growth medium, and the organoids were left to recover. Imaging and CellTiter Blue viability assay were performed on day 14.” – Line 998-1000

Fig 3

18) In Fig 3a, the authors perform an *in vitro* N-terminal acetylation assay using IPed Naa60 in an acetylation mix with different peptides to be tested. Appropriate positive and negative peptide controls were used, which adds strength to the experiment. Traditionally, in the literature, this assay has been performed using radioactive labelled Ac-CoA. However, here MS was opted to detect Nt-acetylated vs un-Nt-acetylated peptides. We are unsure whether this readout works. It is a bit challenging to assess this experiment as a reviewer without access to raw/source MS data. We presume that the authors have performed LFQ MS, which makes accurate quantification Nt-Ac peptide vs un-Nt-Ac peptide problematic. Also, the bar chart/dot plots show the relative abundance, with a very modest increase for the NtAc-peptide in the presence of Naa60. Are these data statistically significant?

Answer: We appreciate the reviewer’s detailed feedback on our *in vitro* N-terminal acetylation assay. While we opted for mass spectrometry (MS) as an alternative to the use of radioactive labelled Ac-CoA, we acknowledge the challenges this method presents, particularly with respect to quantifying Nt-acetylated peptides. Indeed, the increase for the NtAc-peptide in the presence of NAA60 is modest in our current assay, and we agree that further optimization could improve this process. The relative abundance of acetylated versus unacetylated peptides can indeed depend heavily on factors such as input levels and enzyme conversion rates, which we aimed to optimize by using a high substrate input to ensure robust enzyme-substrate interactions. Although modest, we do find an increase for the NtAc-peptide in the presence of NAA60 for the LRRC8A, LRRC8D, and we think that the presentation of these qualitative data are biologically relevant.

19) In Fig. 3c, the sequence of the peptides identified is not mentioned. It would be good if for reviewers could be able to access raw data.

Answer: The N-terminal peptide fragment which got identified was: MIPVTELR for both the acetylated and non-acetylated form.

20) In Fig. 3d, how was this normalized? The normalization strategy should be explained and justified.

Answer: We have made extensive additions to the methods sections for the mass spectrometry analysis of the Nt peptide of LRRC8A and LRRC8D, where sample preparation, data acquisition, and analysis are explained in depth (Lines 588-609). Specifically, to the normalization of the data in 3d: Peptide intensities were calculated by summing the ion intensities for each peptide form. All peptide intensities across all samples were then normalized by variance stabilization in R.

21) In Fig. 3d, the Nt-acetylated peptide was only detected in one of three samples. That is a weakness, if the interpretation of a positive substrate relies on lacking detection of acetylated peptide in the Naa60 KO samples.

Answer: We agree with the reviewer that detecting the Nt-acetylated peptide in only one of the three samples represents a limitation and weakens the interpretation that the absence of the acetylated form in the Naa60 KO samples reflects a lack of acetylation. We recognize that interpreting the lack of detection is inherently challenging due to technical variability in mass spectrometry analyses. However, we would like to emphasize that for LRRC8D, we consistently detected the non-acetylated form exclusively in the Naa60 knockout samples, while the acetylated form was only detected in the wild-type samples. This differential detection supports the conclusion that NAA60 is responsible for Nt-acetylation of LRRC8D, even though the variability in detecting the LRRC8A peptide introduces some uncertainty. We acknowledge this as a limitation in the interpretation of our data, but we believe the findings for LRRC8D still provide important evidence of NAA60's role in Nt-acetylation of this protein.

22) In Fig 3c vs 3d, two different approaches were used for isolating LRRC8A vs LRRC8D. The reason for this should be mentioned.

Answer: Due to the low abundance, the N-terminal peptides were generally very difficult to detect. For LRRC8A we could improve the detection of the N-terminal peptide by using the immune precipitation approach, and for LRRC8D it was only possible to identify the N-terminal peptide using the molecular weight band isolation approach in all groups.

In line 158, we added: "For the analysis of LRRC8D, due to its lower abundance, we excised the gel band corresponding to the molecular weight of LRRC8D..."

23) Fig 3e and Ext data fig. 5, the PLA experiments:

a) Could preferably be quantified, perhaps with a similar strategy as used in Ext data fig 1, or with accompanying flow cytometry.

Answer: The PLA experiment was performed to qualitatively assess the formation and localization of foci, rather than quantification. For these experiments a polyclonal rescue cell line was used, meaning that not all of the cells expressed both constructs (as seen in Figure S5). A quantification of the data shown in Figure 3e would be heavily influenced by the number of positive and negative cells per image. The number of foci would furthermore depend on the expression levels within each cell. Therefore, we refrained from quantifying the absolute number of foci per cell and we prefer to stick to the qualitative assessment.

b) For the negative control with empty plasmid, it is important that both primary antibodies were used and full PLA protocol was performed. Is this the case? Furthermore, ideally, single transfections pOZ HA-Naa60 and pOZ Lrrc8a-Myc in two separate samples, each treated with both anti-HA and -Myc

primaries and full PLA protocol would have been ideal. Can the authors please clarify that this is what is shown in Ext data fig. 5?

Answer: Yes, the full staining protocol including both primary antibodies and PLA reaction was used also for the empty plasmid transduced controls as well as for the single transduction controls (all the samples were treated the same). The secondary antibody staining against the primary antibodies used in the PLA to identify positive cells shown in extended data fig. 5 was performed only after the PLA reaction was concluded.

c) A co-transfected negative control protein could have been considered. However, it is difficult to know for sure what may be a suitable negative control with a similar subcellular localization, since binding partners of NAA60 has not been investigated. Nevertheless, if the interaction is assumed to occur because of Nt-acetylation reaction, it should then be dependent on the N-terminal sequence of *Lrrc8a*. The ideal control (in order to conclude a substrate-enzyme based interaction) would therefore be to mutate the *Lrrc8a*-Myc, to an N-terminus non-suitable for Nt-acetylation (f. ex. I2P) to use as a negative control.

Answer: We appreciate the reviewer's suggestion and agree that including a control transduction with a mutated N-terminus, such as an I2P mutation, would have provided additional valuable information on the substrate-enzyme interaction. However, the primary goal of this assay was to demonstrate the close proximity between NAA60 and LRRC8A, which is an important first step in understanding their potential interaction. While introducing a mutated *Lrrc8a* N-terminus could have given more insights into the substrate-enzyme relationship, it is challenging to determine a suitable negative control for NAA60, as we cannot be certain whether the selected N-terminus is indeed a substrate for NAA60 or not. Additionally, the number of foci observed in such assays may be influenced by differences in the expression levels of each rescue protein, complicating the interpretation of substrate-enzyme interactions. We would have preferred to use endogenous expression for these assays, but as mentioned earlier, we lacked reliable antibodies to detect mouse NAA60 and LRRC8A in an immunofluorescence setting. This technical limitation prevented us from pursuing this approach.

d) To be able to make comparisons, it should ideally have been shown cells with similar expression levels and cell size/confluency.

Answer: We agree that for the displayed empty plasmid control image, a region is shown where the cell density is much higher than for the rescue cell lines. For a better comparison a more similar control image was chosen. In our revised manuscript, we have exchanged the image of the empty plasmid control in Figure 3e for another one out of the same replicate, which displays a region with similar cell density. For improved visibility, the color of the single PLA channel images was exchanged from red to grayscale.

e) It is stated that "The scale bars equal 20µm.", but two different sets of scalebar sizes are shown in the images.

Answer: In Figure 3e the scale bars for both the full-size images and the cropped images equal 10 µm. This information was indeed missing and we have now added it to the figure legend of Figure 3e.

f) It is not stated what cells were used in Ext. data fig. 5.

Answer: For the LRRC8A-related PLA we used the *DbKO_A (Naa60 -/-; Lrrc8a -/-)* line and for the LRRC8D-related PLA the *DbKO_D (Naa60 -/-; Lrrc8d -/-)* line. We realized that we had not included the

empty plasmid and single HA-NAA60 control transduction images for the results of the PLA with the *DbKO_D* line. We have now included those as well as the corresponding labelling of the cell lines into the figure. Thank you for pointing this out.

g) Lines 163-166 the authors state: “In the case of NAA60 and LRRC8A, these foci accumulate at the Golgi apparatus which we assessed by co-staining with a Golgi marker protein (GM130) (Fig.3e and Extended Data Fig.5). The formation of PLA foci was also observed for HA-NAA60 and LRRC8D-Myc. However, for LRRC8D a scattered localization pattern could be observed (Fig.3e and Extended Data Fig.5).” The accumulation of PLA signal at the Golgi seems to be true, but is however not robustly investigated here. Comparing Fig 3e and Ext. data fig. 5, to our impression, there is some variation in the degree of Golgi-closeness vs cytosol dispersion of the PLA signals of both LRRC8A and LRRC8D.

Answer: The dispersion level might be dependent on the expression level of each of the constructs. The higher the NAA60 expression, the more the enzyme is dispersed also into Golgi vesicles. As mentioned earlier using the endogenous expression would have been the favorable approach to be able to truly quantify the interactions between NAA60 and its substrates.

Fig 4

24) In Fig. 4 c-f, it would have been possible to better interpret the degree of rescue if treated WT cells had been included. Perhaps it is possible to point this out based on already existing data.

Answer: The *DbKO_A* line grows slower and generally is less viable than the ntg, and single knockout cell lines. This is why we refrained from comparing it to the ntg or any of the other knockout cell lines and rather chose the approach to rescue each individual gene back in.

25) Line 176 in the main text: seems like the wrong figure reference has been inserted.

Answer: Thanks for spotting this mistake. It is corrected in the revised manuscript.

Fig 5

26) In Fig. 5c, a representative blot could be shown.

Answer: A representative blot of the cell lines ntgC1 and Naa60_12A6 was included as Figure 5c. As there was barely any degradation visible for LRRC8A and LRRC8D over the treatment time course (not uncommon for membrane proteins), we also included an anti-ubiquitin staining as a positive control for the cycloheximide treatment, where a clear decrease in total ubiquitin levels is apparent. The uncropped versions of these Western Blots were also included in the main figure source data file.

27) In Fig. 5d and e, data availability will be particularly important.

Answer: The mass spectrometry proteomics data have been deposited to the ProteomeXchange Consortium via the PRIDE partner repository with the dataset identifier PXD035143. The data can be accessed, using the following credentials:

Username: reviewer_pxd035143@ebi.ac.uk

Password: ERUCD4EB

28) In Fig. 5d and e, was the success/quality of the PM isolation assessed? This should preferentially be shown. Perhaps the proteomics will demonstrate enrichment of PM proteins?

Answer: We appreciate the reviewer's attention to the quality assessment of our plasma membrane (PM) isolation. Ensuring the success of the PM enrichment is indeed crucial for the validity of our proteomic analyses. We performed mass spectrometry analyses on both whole cell lysates and plasma membrane-enriched fractions from wild-type and NAA60-deficient cell lines. To confirm that the PM isolation procedure effectively enriched plasma membrane proteins we identified several known plasma membrane proteins that were detected in both the whole cell lysates and the PM-enriched fractions. We observed a significant increase in the abundance of these proteins in the PM-enriched fractions compared to the whole cell lysates, indicating successful enrichment of plasma membrane components (Figure 2 for the reviewers). This figure confirms the effectiveness of our PM isolation. If the editor thinks that we should add this figure as additional supplementary figure, we are happy to include it.

Figure 2 for the reviewers shows the enrichment for selected plasma membrane proteins using the plasma membrane enrichment protocol in comparison to when whole cell lysate was used for the analysis.

29) Line 197: In this plasma membrane proteomics, endogenous LRCC8A was detected in WT and Naa60 KO cells. Was the Nt-Ac checked here? If so, this could be a very nice confirmation of Fig. 3c. Answer: Unfortunately, the N-terminal peptide of LRRC8A was not identified in the plasma membrane enriched fraction, therefore also no statement on the Nt-Ac status could be made.

30) In Fig. 5f, a rather simplified co-localization experiment is done. We are unsure whether it is possible to conclude with PM localization based on this. Very dense cell cultures are used, and it is

impossible to judge whether the seemingly overlapping signal in the zoom-in frames indeed represents PM localization. Although CellMask is considered a PM stain, it relatively rapidly gets internalized by cells.

Answer: We prefer the CellMask staining as no membrane permeabilization is necessary for this staining step and therefore no disruption of the membrane occurs. We have found that most permeabilization steps lead to blotchiness of membrane protein-related antibody staining.

The provider's (Invitrogen) protocol suggests incubation times for up to 60-90 minutes without detectable internalization of the CellMask plasma membrane stain. We have not tested the incubation limits until internalization occurs in our cell line, however with the very short incubation time of only 10 minutes at 37°C which we used, the likelihood of internalization should be very low.

Fig 6

31 In Fig. 6 the authors functionally test LRRC8A and LRRC8D with a positively charged N-terminus to mimic the loss of Nt-acetylation. We question whether the mutations I2R and F2R in LRRC8A and LRRC8D, is indeed a good mimic of lacking N-terminal acetylation. The most frequently used approach to this strategy is to mutate to an N-terminus that is not compatible with N-terminal acetylation (X2P), to get a proline-starting N-terminus. The authors assume that the MR N-termini are Nt-acetylated by NAA60 as the WT proteins, however there is no grounds for believing that. MR N-termini has been understudied in the field due to protease digestion, and as such not many MR-starting proteins have been assessed for their Nt-acetylation status and no MR-starting proteins has been identified as NAA60 substrates. Thus, it is unknown whether the mutated N-termini will be Nt-acetylated (and by which NAT). Thus these N-termini may have a double positive charge (both from the arginine as well as from the alpha amino NH₃⁺). Either the Nt-acetylation status needs to be assessed, or the more conventional proline strategy used, or else it will be necessary to rephrase the claims concerning Fig. 6.

Answer: We thank the reviewer for raising this important point. Our rationale for using the MR mutations (I2R and F2R) was to ensure that the N-termini retained a positive charge, even in the event that the mutated N-terminus might fall into the substrate category of another NAT, which could potentially acetylate these residues. We were concerned that insertion of a proline at the second position could destabilize the protein, given that proline residues are known to introduce kinks in polypeptide chains and could potentially impair proper protein folding. Additionally, previous studies have shown that cleavage of the LRRC8A N-terminus results in the abrogation of current, which could compromise the functional integrity of the protein.

However, we acknowledge the reviewer's concern that the MR N-termini might undergo protease digestion, which could indeed impact protein function similarly to a Met-Pro sequence. This is a potential limitation of our approach, and we agree that the Nt-acetylation status of these MR N-termini should be experimentally assessed to clarify whether they are acetylated and by which NAT. We also understand that the MR-starting N-termini have been less studied in the context of Nt-acetylation, and we appreciate the reviewer bringing this to our attention.

To address this limitation, we have included the following sentence in the discussion in *lines 310-315*:

"Moreover, while our use of MR N-termini mutations was intended to ensure the retention of a positive charge at the N-terminus, it is important to acknowledge that the Nt-acetylation status of these mutated N-termini remains unverified. Future studies should assess whether these MR N-termini are acetylated and by which NAT, as well as explore alternative strategies such as the use of proline at position two to fully elucidate the functional implications of Nt-acetylation in LRRC8A and LRRC8D."

32) The plasmids with the mutated N-termini seem to partially provide the increased drug sensitivity, is the difference statistically significant?, could comment on this.

Answer: Yes, although there is a clear difference to the wild-type, the mutated rescue constructs do provide statistically significant drug sensitization in certain cases. For the *Lrrc8a* rescue, the I2R mutation significantly increased sensitivity to all tested drugs (cisplatin, carboplatin, and blasticidin). However, for the *Lrrc8d* rescue, the F2R mutation significantly increased sensitivity only to blasticidin. In contrast, when treated with cisplatin and carboplatin, the mutated *Lrrc8d* construct actually resulted in increased drug resistance, as opposed to sensitization. This difference likely reflects the distinct properties of the drugs. Our approach to simulate a positive charge at the N-terminus using the I2R and F2R mutations seems effective in the case of blasticidin, a drug that does not undergo hydrolysis and does not acquire additional charges, unlike the platinum-based agents. Cisplatin and carboplatin, on the other hand, hydrolyze in aqueous environments, gaining a positive charge that might affect how they permeate through the channel. This could explain why the mutation leads to increased resistance in the case of platinum-based agents, but not for blasticidin. Additionally, the rescue of the mutated constructs may lead to overexpression of the channel. In the case of the *Lrrc8d* F2R mutation, this overexpression may cause more of the mutated LRRC8D subunits to be incorporated into the VRAC complex, resulting in a higher proportion of VRACs containing mutant subunits, which could affect drug permeability and sensitivity. Furthermore, the level of resistance observed upon loss of Naa60 is not as pronounced as that seen with complete loss of the VRAC. This suggests that even with a positive charge at the N-terminus, the channel remains at least partially permeable to platinum-based agents, though the permeability may be reduced compared to the wild-type channel.

33) Fig. 6 m and n are not convincing.

Answer: We appreciate the reviewer's feedback regarding Figures 6m and 6n. We understand the importance of clearly demonstrating the effects of the rescue constructs on γ H2AX foci formation, which reflects DNA damage following cisplatin treatment. In the *Lrrc8a* knockout cells, the baseline levels of γ H2AX foci are indeed low, with fewer cells displaying high positivity after 24 hours of treatment with 2 μ M cisplatin. This is consistent with the reduced sensitivity of these knockout cells to cisplatin. Upon reintroduction of the rescue constructs, we observed an increase in the number of highly γ H2AX-positive cells, particularly with the wild-type construct. The wild-type *Lrrc8a* construct significantly sensitizes the cells to cisplatin, as evidenced by the widespread increase in γ H2AX-positive cells, indicating enhanced DNA damage. The I2R mutated construct also increases the number of γ H2AX-positive cells, but to a lesser extent than the wild-type construct, which aligns with the partial restoration of cisplatin sensitivity seen in the growth assays. For the *Lrrc8d* rescue, the cisplatin concentration and treatment duration were optimized to induce a detectable level of DNA damage in the knockout cells. As expected, a high level of γ H2AX foci is observed in these cells following cisplatin treatment. Upon reintroduction of the F2R mutated construct, we observed a decrease in the number of γ H2AX foci and fewer cells being highly positive. This suggests that the F2R mutation impairs the function of LRRC8D, leading to increased resistance to cisplatin, as reflected by the decreased DNA damage. In contrast, the wild-type *Lrrc8d* construct restores cisplatin sensitivity, resulting in a higher number of γ H2AX-positive cells, which is consistent with the growth assay results. We believe that these results demonstrate the differential effects of the wild-type and mutated constructs on DNA damage response and cisplatin sensitivity.

Methods

34) Please add more information on how the MS data were analysed and normalized etc.
Answer: We have made extensive additions to the method sections for the mass spectrometry analysis of the Nt peptide of LRRC8A and LRRC8D (*lines 588-609*) as well as the plasma membrane enrichment analysis (*lines 753-771 and lines 779-789*).

Reviewer #2 (Remarks to the Author):

In this paper Widmer and colleagues use in vitro and in vivo models to search for cellular determinants of sensitivity to platinum-based drugs. Earlier work of the lab has identified LRRC8A and LRRC8D, two volume-regulated anion channel subunits to mediate platinum uptake and hence cellular sensitivity. Here, a genome-wide CRISPR screen is used to reveal NAA60, a N-terminal acetyltransferase to modulate LRRC8A and LRRC8D function. Convincing results are shown to demonstrate the regulation of LRRC8A/D function through N-terminal acetylation mediated by NAA60.

Answer: We sincerely thank the reviewer for their positive assessment of our manuscript and for recognizing the significance of our findings. We are pleased that the reviewer found our results convincing in demonstrating the regulation of LRRC8A/D function through N-terminal acetylation mediated by NAA60. We believe these insights contribute to a better understanding of the mechanisms underlying cellular sensitivity to platinum-based drugs and are grateful for the reviewer's supportive comments.

1) I am not sure if the full dataset showing the results of the CRISPR screen is released in the paper. I think it should be.

Answer: We have included the results of the CRISPR screen as Supplementary Table 1 in the main manuscript file instead of the source data file for Figure 1. Raw sequencing data is released and made accessible in the European Nucleotide Archive under the study accession number PRJEB75036 (<https://www.ebi.ac.uk/ena/browser/view/PRJEB75036>).

2) Is Pt uptake different in LRRC8A KO and NAA60 KO cells? If yes, what is the explanation? (Fig 2a)

Answer: We thank the reviewer for this observation. In previous studies it was shown that LRRC8A is the obligatory subunit for channel formation and function⁴. In the case of *Naa60* knockout, the channel is still present at the plasma membrane, however, the positive charges at the N-termini of the LRRC8 subunits might restrict the permeability to substrates such as the Pt agents cisplatin and carboplatin. This is also what we observe in the Helios Pt drug uptake experiments: *Lrrc8a* knockout cell lines accumulate less platinum than *Naa60* knockout cells (Figure 2a).

3) What is the explanation of the staining pattern of LRRC8D?

Answer: The Western Blot staining pattern of LRRC8D might display multiple bands due to phosphorylation or cleaved products.

4) What is the clinical relevance of these findings? Can the role of NAA60 in response to treatment or survival be ascertained? In the light of the answer to this question, is NAA60 a therapeutic target?

Answer: We appreciate the reviewer's inquiry into the clinical relevance of our findings and the potential therapeutic implications of NAA60. Gene defects and loss of function mutations in NAA60 do

indeed exist, as recently demonstrated by Chelban *et al.* (Nat Comm 2024, our new reference 43), who linked biallelic mutations in NAA60 to primary familial brain calcification¹. While these findings highlight the physiological importance of NAA60, our study specifically addresses its role in modulating cellular sensitivity to platinum-based chemotherapeutic agents through the regulation of LRRC8A/D function. Regarding clinical relevance, our data suggest that NAA60 is a key determinant of platinum drug uptake and subsequent DNA damage in cancer cells. This implicates NAA60 as a potential biomarker for predicting the efficacy of platinum-based therapies, particularly in tumors where NAA60 expression or function is compromised. Furthermore, since NAA60-mediated Nt-acetylation is crucial for the proper function of LRRC8A/D in facilitating drug uptake, targeting NAA60 or its acetylation activity could be explored as a novel therapeutic strategy to enhance the sensitivity of cancer cells to platinum-based treatments. However, further studies are needed to ascertain the direct impact of NAA60 expression on patient outcomes and treatment response in clinical settings. Investigations into the expression levels of NAA60 in various cancers, as well as studies correlating NAA60 status with patient survival and response to platinum-based therapies, would be crucial steps toward validating NAA60 as a therapeutic target. However, this would also require working antibodies that reliably detect NAA60 in tissues, and, unfortunately, all of our attempts to detect NAA60 with available antibodies or new antibodies that we raised failed. Therefore, we think that such analyses are beyond the scope of our current study. In our article, we prefer to refrain from claims regarding the clinical relevance, and with the NAA60-mediated acetylation of LRRC8A/D focus on adding an important mechanism to the uptake of platinum drugs.

5. Can the role of N-terminal acetylation be also verified in whole-cell patch-clamp experiments, as performed by the authors in their 2015 EMBO Journal paper?

Answer: We appreciate the reviewer's suggestion to verify the role of N-terminal acetylation using whole-cell patch-clamp experiments. While patch-clamp assays would indeed provide direct measurements of ion currents under hypotonic/hypertonic challenge, we did not perform these experiments due to the unavailability of the necessary equipment in our laboratory. Instead, we employed alternative approaches to assess the functional consequences of *Naa60* and *Lrrc8a* deficiency under hypotonic conditions. Specifically, we conducted clonogenic assays where wild-type, *Lrrc8a*, and *Naa60* knockout cells were challenged with hypotonic medium for 24 hours. Specifically, the growth medium, which per manufacturer's information has an osmolality of ~330 mOsm/kg was diluted 1:1 or 1:2 with distilled water. Following this treatment, the cells were allowed to recover for 5 days, and survival was assessed using crystal violet staining (Figure 3a and 3b for the reviewer). Our results indicate that both *Lrrc8a* and *Naa60* knockout cell lines display decreased survival compared to control cell lines, suggesting that NAA60 may play a role in long-term volume regulation and cellular resilience under hypotonic stress. Additionally, we assessed the short-term volume response by measuring changes in calcium fluorescence intensity upon hypotonic challenge in control (ntgB1, ntgC1), *Lrrc8a*, and *Naa60* knockout cell lines (Figure 3c for the reviewer). Our data show that the *Lrrc8a* knockout cells exhibit a pronounced increase in volume response and reduced recovery capabilities, while the *Naa60* knockout cells do not show the same acute deficiency in coping with hypotonic stress (Figure 3d and 3e). These results suggest that while the loss of *Naa60* may affect the cell's ability to manage long-term hypotonic stress, it does not cause an acute deficiency in volume regulation as observed with *Lrrc8a* knockout cells. We believe these findings provide useful insights into the role of N-terminal acetylation in volume regulation and cellular response to osmotic stress, albeit through an alternative experimental approach. We show the data for the reviewers only, since we do not think that this is crucial for the main message of our manuscript. If the editor thinks that

these data should be included as supplementary data in the manuscript, we have happy to include them.

Figure 3 for the reviewers. Comparison of wild type, Lrrc8a, Lrrc8d, and Naa60 knockout cell lines towards hypotonic challenges **a** representative images of a growth assay where cell lines were challenged with normal and hypotonic medium conditions for 24h, afterwards they were left to recover for 5 days and a crystal violet staining was performed **b** Quantification of the growth assay of wild type and Naa60 knockout cell lines challenged with hypotonic conditions for 24h, data of 6 replicates is shown. Images were quantified using the ColonyArea plugin of image J. P values were calculated by GraphPad Prism software 9 using 2way ANOVA multiple comparisons test. **c** calcein fluorescence increase upon hypotonic challenge of wild type, Lrrc8a, and Naa60 knockout cell lines. The cells were loaded with 5 μ M calcein solution. For the fluorescence acquisition, the medium in the wells was exchanged to PBS (+) and the hypotonic challenge consisted of the addition of distilled water at the ratio of 1:2 **d** percent of fluorescence response to hypotonic challenge **e** percent of deviation from baseline at t=2500s of the assay, P values were calculated by GraphPad Prism software 9 using ordinary one-way ANOVA multiple comparisons test.

In summary, this is an excellent study, reporting novel findings on the functional association of NAA60 and LRR8A/D. The paper is of interest to the wider community. Analysis of patient data would help to evaluate the clinical relevance of the results.

References

1. Chelban, V. *et al.* Biallelic NAA60 variants with impaired n-terminal acetylation capacity cause autosomal recessive primary familial brain calcifications. *Nat. Commun.* **15**, 2269 (2024).
2. Chen, J. Y. *et al.* Structure and function of human Naa60 (NatF), a Golgi-localized bi-functional acetyltransferase. *Sci. Reports 2016 61* **6**, 1–12 (2016).
3. Støve, S. I. I. *et al.* Crystal Structure of the Golgi-Associated Human N α -Acetyltransferase 60 Reveals the Molecular Determinants for Substrate-Specific Acetylation. *Structure* **24**, 1044–1056 (2016).
4. Planells-Cases, R. *et al.* Subunit composition of VRAC channels determines substrate specificity and cellular resistance to Pt-based anti-cancer drugs. *EMBO J.* **34**, 2993–3008 (2015).

Reviewers' comments:

Reviewer #1 (Remarks to the Author):

Overall, the authors have come back with several good answers to our concerns. And our suggestions for minor improvements have been followed. However, some of the more severe comments on data quality was not met. While the authors present solid evidence that NAA60 is necessary for effective platinum drug uptake, we still (as no additional experiments were performed) have concerns about the data used to claim that the effect is mediated via NAA60's function as an Nt-acetyl transferase. Although it may appear obvious that lack of Nt-acetylation must indeed be at play, it is important to acknowledge that for many of the NATs moonlighting functions are being revealed. Therefore, weak evidence is not enough to claim that the Pt drug effect is dependent on NAA60's Nt-acetylation.

Answer: We sincerely thank the reviewer for their detailed and constructive feedback. We acknowledge the importance of carefully addressing the mechanistic role of NAA60 in platinum drug uptake and its attribution to NAA60's function as an N-terminal acetyltransferase. While we provided indirect but strong evidence that NAA60 mediates its effects via Nt acetylation of LRRC8A and LRRC8D subunits, we agree that additional experiments to further substantiate this claim would strengthen the conclusions. Unfortunately, our attempts to establish a more sensitive assay to directly measure Nt acetylation activity or improve the sensitivity of our current assays were not successful. Thus, we have taken steps to adjust the manuscript to ensure our claims reflect the available data more accurately and address your concerns. In particular, we have changed the following:

- Title: We have removed the phrase "by N-terminal acetylation" from the title to avoid overstatement of this specific mechanism.
- Abstract: In the abstract, we have replaced "We demonstrate" with "Our data strongly suggest that" in reference to the role of NAA60's Nt acetylation in platinum drug uptake. This better reflects the level of evidence presented in our study and acknowledges that alternative mechanisms, including potential moonlighting functions, cannot be excluded at this stage.
- Results: In line 135, we have changed the subtitle "LRRC8A and LRRC8D N-termini are a substrate for NAA60" to "Investigation of LRRC8A and LRRC8D N-termini as a substrate for NAA60".
- Discussion: In line 251/252 we have replaced "... we show that this is due to Nt acetylation of the LRRC8A and LRRC8D subunits of VRAC" by "... our data strongly suggest that this is due to Nt acetylation of the LRRC8A and LRRC8D subunits of VRAC. " Moreover, in the revised discussion, we explicitly acknowledge the limitations of our current assay: In lines 281-283, we added: "To measure NAA60-mediated Nt-acetylated versus un-Nt-acetylated peptides in our *in vitro* assay, we used mass spectrometry. As these values are low, further corroboration of our data using radioactively labelled Ac-CoA as a readout may be useful."

We hope that the reviewer agrees that these modifications sufficiently balance our findings with their limitations.

Specifically:

Comment 18 on Fig. 3a:

The values are VERY low. What are the raw intensity data here? Raw data was not shared as requested. We would like to see the numbers before the relative normalization. Data in 3a is not significant and hence does not show increased Nt-acetylation in the presence of Naa60. It is not possible to make any claims/statements/conclusions about Nt-acetylation status based on these experiments.

Answer:

We appreciate the reviewer's request for the raw intensity data and acknowledge the need for transparency in data presentation. In response, we have now included two Excel files (Fig3a_NAA60_in_vitro_acetylation_T7 and Fig3a_NAA60_in_vitro_acetylation_R7) containing the raw spectral counts and intensity data from the T7 and R7 experiments:

- T7 Experiment: A trial experiment conducted to ensure that the experimental setup was functional before proceeding with full measurements.
- R7 Experiment: Three biological replicates (R7.1-3) per peptide and enzyme incubation condition to validate the findings.

To further address the reviewer's concerns, Figure 1 in Revision 2 now displays both raw intensity values (top row) and spectral counts (bottom row) per peptide. These graphs can be included as a supplementary figure, if deemed necessary, to allow for a more thorough evaluation of the results.

Regarding potential confounding factors, we observed additional modifications at the initiator methionine site, including:

- Methionine removal
- Methylation (+15.9949 Da)
- Suspected bead crosslinking reagent component (+113.9953 Da), potentially introduced during NAA60 enzyme isolation via HA-tagged magnetic beads.

These modifications may have competed with the N-terminal acetylation reaction, reducing the availability of non-acetylated substrate peptides. To account for this:

- In T7 samples, we excluded these modifications from relative abundance calculations.
- In R7 samples, methylated peptides were initially included in the relative abundance calculations but have now been excluded to ensure consistency in the updated Figure 3a and source data file.

Finally, we have now added the full raw data to the source data file for Figure 3, as requested. We hope that these clarifications, along with the newly provided data, will sufficiently address the reviewer's concerns.

Figure 1 for Review 2 Raw peptide intensity values (a) and spectral counts (b) for each peptide.

Comment 19 on Fig. 3c:

We had concerns and asked to see raw data. This could be an excel file of the identified peptides and their intensities.

Answer: A excel file with the raw intensity data of this experiment is provided with the revised manuscript.

Fig3c_20220222_IP_LRRC8A

Fig3c_20220704_IP_LRRC8A

Comments 20-21 on Fig. 3d:

Raw data available? The y axis label should be explained. Why were not raw intensities used like in c? While more methodological details for the MS was added to the manuscript, info about data normalization was not added. It should be transparent how that data were normalized/transformed from raw values.

Answer:

We appreciate the reviewer's request. In response, we have made the following clarifications and revisions:

Availability of raw and normalized data:

- The raw MaxQuant intensities (pre-normalization) are now provided in the file Fig3d_20210511_MQ_evidence.
- The variance-normalized data using the evidenceWnorm approach are provided in Fig3d_20210511_evidenceWnorm.

- To facilitate interpretation, peptides included in the analysis are highlighted in yellow and the table has been sorted accordingly.

Explanation of data normalization:

- The evidenceWnorm values were obtained using variance normalization based on intensities detected by MaxQuant.
- This normalization method adjusts for intensity-dependent variance to make comparisons more robust and interpretable.
- The PM1 label represents samples from the parental line, while PM2 corresponds to samples from the Naa60_12A6 line.

Clarification of the Y-axis label in Fig. 3d:

We have revised the Y-axis label for clarity and will ensure that the description in the figure legend explicitly states what is being represented.

Methods section update:

To enhance transparency, we have added the following sentence to the methods section (Lines 619-621):

“For the LRRC8D peptide, the intensities detected by MaxQuant were variance normalized to obtain the evidenceWnorm values shown in Figure 3d.”

We hope these revisions provide the necessary clarity and transparency regarding the raw data, normalization approach, and figure labeling. Thank you for your valuable feedback.

Comment 23 on Fig 3e and Ext data fig. 5, the PLA experiments: The text should convey a bit more caution when interpreting the results and be open about possible variations due to expression levels and so on with transfection.

Answer: The following cautionary sentences were added to the main text of the manuscript:

“The number and localization pattern of the foci may depend on the expression levels of the individual proteins. The higher dispersion levels seen in some of the images could be an artefact of NAA60 overexpression.” – Line 173-174

Comment 30 concerning Fig. 5f:

While it sounds like the authors have used the CellMask stain correct, our comment concerning the difficulties in interpreting these results was not responded to. We have a hard time accepting this as evidence for PM localization and it is definitely not possible to conclude with increased PM localization in the absence of Naa60. We understand that microscopy of PM localization is difficult, however the current data does not demonstrate what the authors claim it to demonstrate. We see that a 100 x objective was used and that samples were imaged in Z-stacks of 20 slices of 0.2µm thickness. This

should allow for a 3D analysis in order to measure plasma membrane localization rather than simply showing the images as maximum projection. The images shown do not support the claim.

Answer:

We appreciate the reviewer's concerns regarding the interpretation of our PM localization data and the challenges in quantifying membrane colocalization using microscopy. We would like to clarify that the aim of this experiment was not to claim an increase in LRRC8A PM localization in the absence of NAA60, but rather to demonstrate that LRRC8A is still present at the plasma membrane despite the loss of NAA60.

We acknowledge that a more quantitative approach would require a proper negative control, such as a solely cytosolic GFP construct, for direct comparison. Unfortunately, we did not acquire such a control in the original experiment, which is why we have refrained from performing a full quantification of PM localization. To enhance the interpretability of the existing data, we have now:

- Used the "Plot Profile" function in ImageJ to generate linear intensity profiles for the GFP and CellMask channels across the plasma membrane.
- Added these intensity profiles to Figure 5g to better illustrate the observed colocalization pattern.
- Updated the figure legend for Figure 5g to describe the new intensity profile analysis.
- Expanded the methods section to detail the approach used for this visualization.

While we acknowledge the reviewer's suggestion for a 3D analysis using Z-stacks, the lack of a cytosolic control limits our ability to perform quantitative measurements of PM enrichment. Nonetheless, we believe that the intensity profile visualization now provides additional context for interpreting the colocalization pattern and further supports our observation that LRRC8A is not completely absent from the plasma membrane in the absence of NAA60.

We hope this clarification and the additional analysis address the reviewer's concerns. Thank you for your valuable feedback.

Comment 31, concerning Fig. 6:

While the authors have added a precautionary sentence to the manuscript, their claims made in figure headings, abstract, title of the manuscript etc are unchanged and as explained we do not find the evidence concerning Nt-acetylation to be sufficient. Taken together, the data on Nt-acetylation (Figure 3 and Figure 6) all have flaws/weaknesses, making it overall insufficient to conclude that the Pt drug uptake effect observed in the absence of NAA60 is explained by loss of Nt-acetylation.

Answer:

We sincerely appreciate the reviewer's careful evaluation of our work and their emphasis on ensuring that our conclusions are well supported by the available data. We understand and acknowledge the reviewer's concerns regarding the strength of the evidence linking NAA60's role in platinum drug uptake specifically to its N-terminal acetylation activity. In response to this concern, as specified above, we have revised multiple key sections of the manuscript to avoid overstatements and ensure that our claims more accurately reflect the limitations of our data.

Reviewer #2 (Remarks to the Author):

The authors did an excellent job in addressing the extensive comments of the reviewers. Although my comments and suggestions did not result in any changes in the revised manuscript, I find the manuscript ready to be published. Publication of the reviewer comments and the author rebuttal letters in a supplementary file is strongly recommended.

Answer: Thank you very much.

Our answers to the latest comments of R1 (in red) are written in green

Reviewers' comments:

Reviewer #1 (Remarks to the Author):

Overall, the authors have come back with several good answers to our concerns. And our suggestions for minor improvements have been followed. However, some of the more severe comments on data quality was not met. While the authors present solid evidence that NAA60 is necessary for effective platinum drug uptake, we still (as no additional experiments were performed) have concerns about the data used to claim that the effect is mediated via NAA60's function as an Nt-acetyl transferase. Although it may appear obvious that lack of Nt-acetylation must indeed be at play, it is important to acknowledge that for many of the NATs moonlighting functions are being revealed. Therefore, weak evidence is not enough to claim that the Pt drug effect is dependent on NAA60's Nt-acetylation.

Answer: We sincerely thank the reviewer for their detailed and constructive feedback. We acknowledge the importance of carefully addressing the mechanistic role of NAA60 in platinum drug uptake and its attribution to NAA60's function as an N-terminal acetyltransferase. While we provided indirect but strong evidence that NAA60 mediates its effects via Nt acetylation of LRRC8A and LRRC8D subunits, we agree that additional experiments to further substantiate this claim would strengthen the conclusions. Unfortunately, our attempts to establish a more sensitive assay to directly measure Nt acetylation activity or improve the sensitivity of our current assays were not successful. Thus, we have taken steps to adjust the manuscript to ensure our claims reflect the available data more accurately and address your concerns. In particular, we have changed the following:

- Title: We have removed the phrase "by N-terminal acetylation" from the title to avoid overstatement of this specific mechanism.
- R1: OK.
- Abstract: In the abstract, we have replaced "We demonstrate" with "Our data strongly suggest that" in reference to the role of NAA60's Nt acetylation in platinum drug uptake. This better reflects the level of evidence presented in our study and acknowledges that alternative mechanisms, including potential moonlighting functions, cannot be excluded at this stage.
- R1: "We demonstrate" exchanged to "Our data strongly suggest" is a very minor change. The sum of the conclusions/statements still tells an overall story that is stretched somewhat too far beyond what is backed by data. For example, the word "modulate" used in the Abstract is an overstatement since no active regulation by NAA60 was shown, simply an effect of its absence/knockout.

Answer 2: We thank the reviewer for their continued critical and thoughtful feedback. We acknowledge the concern that our phrasing may still overstate the mechanistic interpretation of our data. In response, we have revised the abstract as follows:

- Replaced "modulate" with "affect", to better reflect the absence of direct evidence for active regulation.
- Removed the word "strongly" from "Our data strongly suggest", to further tone down the interpretation of the mechanistic findings.

- Results: In line 135, we have changed the subtitle “LRR8A and LRR8D N-termini are a substrate for NAA60” to “Investigation of LRR8A and LRR8D N-termini as a substrate for NAA60”.
- R1: OK.
- Discussion: In line 251/252 we have replaced “... we show that this is due to Nt acetylation of the LRR8A and LRR8D subunits of VRAC” by “... our data strongly suggest that this is due to Nt acetylation of the LRR8A and LRR8D subunits of VRAC. “
- R1: Very minor change, that does not really meet the concern.

Answer 2: Also here the word «strongly» was removed to further alleviate the reviewer’s concern.

- Moreover, in the revised discussion, we explicitly acknowledge the limitations of our current assay: In lines 281-283, we added: “To measure NAA60-mediated Nt-acetylated versus un-Nt-acetylated peptides in our *in vitro* assay, we used mass spectrometry. As these values are low, further corroboration of our data using radioactively labelled Ac-CoA as a readout may be useful.”
- R1: As explained below we do not find this assay to be able to report anything of value. We acknowledge that Nt-acetylation assays may be difficult to set up and that the authors made a robust effort, albeit it was unfortunately not successful. If the authors would like to include Nt-acetylation assay in their paper we suggest that they use one of the published assays. Although not ideal, Fig 3c and d does however provide evidence of LRR8A and LRR8D IPed from NAA60 KO cells to be less Nt-acetylated compared to IPs from WT cells. Which could justify balanced suggestive statements about the direct involvement of Nt-acetylation.

Answer 2: We thank the reviewer for their constructive feedback regarding the Nt-acetylation assay. We acknowledge the concern that our *in vitro* assay (previously Fig. 3a and b) may not provide definitive evidence, and we agree that published protocols would be preferable. However, despite extensive efforts, the published assays were not useful in our hands.

To reflect the limitations and minimize emphasis on these data, we have moved the corresponding panels to Supplementary Fig. 5. As suggested, we now focus primarily on the mass spectrometry-based analysis of immunoprecipitated LRR8A and LRR8D (now Fig. 3a and b), which more directly supports the conclusion that Nt-acetylation is reduced in the absence of NAA60. We think that this repositioning ensures a more balanced presentation of our data and aligns with the reviewer’s recommendations for making suggestive—rather than conclusive—mechanistic statements.

We hope that the reviewer agrees that these modifications sufficiently balance our findings with their limitations.

Specifically:

Comment 18 on Fig. 3a:

The values are VERY low. What are the raw intensity data here? Raw data was not shared as requested. We would like to see the numbers before the relative normalization. Data in 3a is not significant and hence does not show increased Nt-acetylation in the presence of Naa60. It is not possible to make any claims/statements/conclusions about Nt-acetylation status based on these experiments.

Answer:

We appreciate the reviewer's request for the raw intensity data and acknowledge the need for transparency in data presentation. In response, we have now included two Excel files (Fig3a_NAA60_in_vitro_acetylation_T7 and Fig3a_NAA60_in_vitro_acetylation_R7) containing the raw spectral counts and intensity data from the T7 and R7 experiments:

- T7 Experiment: A trial experiment conducted to ensure that the experimental setup was functional before proceeding with full measurements.
- R7 Experiment: Three biological replicates (R7.1-3) per peptide and enzyme incubation condition to validate the findings.
- **R1: These changes makes sense.**

To further address the reviewer's concerns, Figure 1 in Revision 2 now displays both raw intensity values (top row) and spectral counts (bottom row) per peptide. These graphs can be included as a supplementary figure, if deemed necessary, to allow for a more thorough evaluation of the results.

Regarding potential confounding factors, we observed additional modifications at the initiator methionine site, including:

- Methionine removal
- Methylation (+15.9949 Da)
- Suspected bead crosslinking reagent component (+113.9953 Da), potentially introduced during NAA60 enzyme isolation via HA-tagged magnetic beads.

These modifications may have competed with the N-terminal acetylation reaction, reducing the availability of non-acetylated substrate peptides. To account for this:

- In T7 samples, we excluded these modifications from relative abundance calculations.
- In R7 samples, methylated peptides were initially included in the relative abundance calculations but have now been excluded to ensure consistency in the updated Figure 3a and source data file.

Finally, we have now added the full raw data to the source data file for Figure 3, as requested. We hope that these clarifications, along with the newly provided data, will sufficiently address the reviewer's concerns.

R1: We thank the authors for the additional data material provided. In files Fig3a_NAA60_in_vitro_acetylation_T7 and Fig3a_NAA60_in_vitro_acetylation_R7 (identified as files named: 24920_2_data_set_865470_srb3pb and 24920_2_data_set_865471_srm3pb), we were however unable to identify sample identities (+ vs – NAA60). The Figure 1 in the rebuttal letter is however relatively informative. We now understand a bit more about how these data were processed. However, the authors did not explain the rationale for their normalization strategy, and it seems like the values in Fig 1 in the rebuttal does not add up to what is shown in Fig 3a in the revised manuscript. Thus, it is not transparent how that data were normalized/transformed from raw values. There seems to be too much of the Non-Nt-Ac peptides and these numbers do not decrease when adding NAA60. Another concern, as revealed by Fig 1b there is quite the response in Nt-acetylation by addition of

NAA60 enzyme to the negative control peptide SESSSKS. Moreover, no statistical testing was performed for these experiments, and it is likely that the potential modest differences observed are not statistically significant, hence the concern stated in the previous revision stands, as it is not possible to make any claims/statements/conclusions about Nt-acetylation status based on these experiments”.

Figure 1 for Review 2 Raw peptide intensity values (a) and spectral counts (b) for each peptide.

Answer 2: We sincerely thank the reviewer for their detailed and thoughtful evaluation of the *in vitro* N-terminal acetylation assay. In light of the concerns raised, we have now moved the corresponding panel (previously Fig. 3a) to Supplementary Fig. 5. This repositioning ensures that these data are accessible for transparency, but no longer form a central part of the mechanistic argument.

We agree that the results of this assay are limited and cannot support definitive conclusions. Nonetheless, we believe that showing the data, along with an open discussion of their limitations, is valuable for the field, particularly for others attempting similar assays. If the editor prefers to leave these data completely out, we are also fine with it. To address the reviewer’s comments in more detail:

- Sample identities and annotations have been clarified in the revised source data file.
- We describe our normalization strategy in the Methods section.
- We acknowledge in the text that no statistical significance could be established from the changes observed and we have carefully avoided drawing any strong conclusions from these results. In the results section in lines 151-152 we wrote: “However, the differences observed using this assay did not reach statistical significance.”

In summary, we have taken care to reduce the emphasis on these data, and ensure our interpretations remain appropriately cautious.

Comment 19 on Fig. 3c:

We had concerns and asked to see raw data. This could be an excel file of the identified peptides and their intensities.

Answer: A excel file with the raw intensity data of this experiment is provided with the revised manuscript.

Fig3c_20220222_IP_LRRC8A

Fig3c_20220704_IP_LRRC8A

R1: Although this setup could have preferentially been optimized, the experiments and reported results are acceptable.

Comments 20-21 on Fig. 3d:

Raw data available? The y axis label should be explained. Why were not raw intensities used like in c? While more methodological details for the MS was added to the manuscript, info about data normalization was not added. It should be transparent how that data were normalized/transformed from raw values.

Answer:

We appreciate the reviewer's request. In response, we have made the following clarifications and revisions:

Availability of raw and normalized data:

- The raw MaxQuant intensities (pre-normalization) are now provided in the file Fig3d_20210511_MQ_evidence.
- The variance-normalized data using the evidenceWnorm approach are provided in Fig3d_20210511_evidenceWnorm.
- To facilitate interpretation, peptides included in the analysis are highlighted in yellow and the table has been sorted accordingly.

Explanation of data normalization:

- The evidenceWnorm values were obtained using variance normalization based on intensities detected by MaxQuant.
- This normalization method adjusts for intensity-dependent variance to make comparisons more robust and interpretable.
- The PM1 label represents samples from the parental line, while PM2 corresponds to samples from the Naa60_12A6 line.

Clarification of the Y-axis label in Fig. 3d:

We have revised the Y-axis label for clarity and will ensure that the description in the figure legend explicitly states what is being represented.

Methods section update:

To enhance transparency, we have added the following sentence to the methods section (Lines 619-621):

“For the LRRC8D peptide, the intensities detected by MaxQuant were variance normalized to obtain the evidenceWnorm values shown in Figure 3d.”

We hope these revisions provide the necessary clarity and transparency regarding the raw data, normalization approach, and figure labeling. Thank you for your valuable feedback.

R1: We thank the authors for the additional data and information about normalization. We have no remaining issues to address concerning this. Although this setup could have preferentially been optimized, the experiments and reported results are acceptable.

Answer 2: We thank the reviewer for their feedback.

Comment 23 on Fig 3e and Ext data fig. 5, the PLA experiments: The text should convey a bit more caution when interpreting the results and be open about possible variations due to expression levels and so on with transfection.

Answer: The following cautionary sentences were added to the main text of the manuscript:

“The number and localization pattern of the foci may depend on the expression levels of the individual proteins. The higher dispersion levels seen in some of the images could be an artefact of NAA60 overexpression.” –Line 173-174

R1: OK.

Comment 30 concerning Fig. 5f:

While it sounds like the authors have used the CellMask stain correct, our comment concerning the difficulties in interpreting these results was not responded to. We have a hard time accepting this as evidence for PM localization and it is definitely not possible to conclude with increased PM localization in the absence of Naa60. We understand that microscopy of PM localization is difficult, however the current data does not demonstrate what the authors claim it to demonstrate. We see that a 100 x objective was used and that samples were imaged in Z-stacks of 20 slices of 0.2µm thickness. This should allow for a 3D analysis in order to measure plasma membrane localization rather than simply showing the images as maximum projection. The images shown do not support the claim.

Answer:

We appreciate the reviewer’s concerns regarding the interpretation of our PM localization data and the challenges in quantifying membrane colocalization using microscopy. We would like to clarify that the aim of this experiment was not to claim an increase in LRRC8A PM localization in the absence of NAA60, but rather to demonstrate that LRRC8A is still present at the plasma membrane despite the loss of NAA60.

We acknowledge that a more quantitative approach would require a proper negative control, such as a solely cytosolic GFP construct, for direct comparison. Unfortunately, we did not acquire such a

control in the original experiment, which is why we have refrained from performing a full quantification of PM localization. To enhance the interpretability of the existing data, we have now:

- Used the “Plot Profile” function in ImageJ to generate linear intensity profiles for the GFP and CellMask channels across the plasma membrane.
- Added these intensity profiles to Figure 5g to better illustrate the observed colocalization pattern.
- Updated the figure legend for Figure 5g to describe the new intensity profile analysis.
- Expanded the methods section to detail the approach used for this visualization.

While we acknowledge the reviewer’s suggestion for a 3D analysis using Z-stacks, the lack of a cytosolic control limits our ability to perform quantitative measurements of PM enrichment. Nonetheless, we believe that the intensity profile visualization now provides additional context for interpreting the colocalization pattern and further supports our observation that LRRC8A is not completely absent from the plasma membrane in the absence of NAA60.

We hope this clarification and the additional analysis address the reviewer’s concerns. Thank you for your valuable feedback.

R1: The microscopy has improved somewhat by the additional graph visualization examples, and Fig. 5g is OK to demonstrate that LRRC8A is still present at the plasma membrane despite the loss of NAA60. Increase in LRRC8A PM localization in the absence of NAA60, is still one of the conclusions in the text. Then this is based on the MS data alone. From the Results text: “We therefore analyzed plasma membrane-enriched protein fractions of wild-type and Naa60 knockout cell lines using mass spectrometry. Surprisingly, the analysis revealed that rather than absent, the subunit LRRC8A was instead significantly enriched at the plasma membrane of the two NAA60-deficient cell lines Naa60_12A6 and Naa60_13C3, when compared to the parental cell line (Fig.5e+f).” This means that the quality assessment of the PM isolate is crucial. We therefore revisited the previous rebuttal letter where the authors provided additional supportive data suggested for the supplemental:

Figure 2 for the reviewers shows the enrichment for selected plasma membrane proteins using the plasma membrane enrichment protocol in comparison to when whole cell lysate was used for the analysis.

R1: We noticed that this graph says TM proteins, not PM proteins, so this should be clarified. In order for successful isolation of PM to be validated, it will be necessary to demonstrate that PM proteins are in general enriched, and importantly that the contamination from ER is minor. This could be possible to assess better in the existing dataset and present in supplemental item(s). If the isolate is enriched also in ER proteins, then it will not be possible to claim that loss of NAA60 cause increase in PM localization.

Answer 2: The graph encompasses the analysis of several selected membrane proteins using the whole cell isolation or the plasma membrane enrichment protocols. The assessed proteins are the following: CACNA1B, CACNA1A, CACNA1H, ABCC1, MTND5, TMEM245, SLC7A1, PIGN, SLC6A14, NNT, MTND4, MTCO1, SLC9A1, STT3A, SLC6A9, SLC6A6, SLC12A4, ABCA7.

We thank the reviewers for raising the concern of the contamination with ER proteins. Using a STRING-based classification, we identified ER-associated transmembrane proteins across both datasets and found the following:

- In the PM-enriched samples, 220 out of 912 transmembrane proteins (24%) were ER-associated.
- In the WCL samples, 163 out of 624 transmembrane proteins (26%) were ER-associated.

This indicates that the relative number of ER proteins was not disproportionately increased by the PM enrichment protocol.

However, we acknowledge that signal intensity values (iLFQ) of ER proteins, as well as other organellar membrane proteins, were also increased in the PM-enriched samples, likely reflecting non-specific co-enrichment of membrane-bound compartments (see the Figure 3 for the reviewer below). We now state this as a limitation of the enrichment method in the revised Methods section in lines (lines 751-758):

“To assess the quality of the membrane enrichment, we compared the representation of known plasma membrane and endoplasmic reticulum (ER) proteins identified in the enriched samples versus whole-cell lysates. While the relative proportion of ER proteins remained comparable between the two preparations, we noted that signal intensities for both plasma membrane and organellar membrane proteins, including those from the ER, were increased in the enriched fractions. This suggests that the protocol globally enriches membrane-associated proteins rather than exclusively plasma membrane components. As such, some level of ER co-enrichment cannot be excluded and should be considered a limitation of the approach.”

Figure 3 for the reviewers shows the comparison of iLFQ values for ER and PM proteins depending on the protein isolation procedure.

Regarding LRRC8A, the increase in its abundance in the PM-enriched fractions of NAA60-deficient cells, as identified by MS, could, in principle, reflect accumulation at intracellular membranes, such as the ER. While our microscopy data support the presence (not increase) of LRRC8A at the plasma

membrane, we argue that the overall upregulation of LRRC8A protein in these fractions may still reflect a compensatory mechanism in response to impaired VRAC function, regardless of its exact subcellular localization. This is consistent with previous observations that loss of channel activity can trigger compensatory overexpression of channel subunits. To acknowledge the potential limitation of ER protein enrichment, we added the following sentence in the discussion in lines 275-280:

“While our mass spectrometry analysis indicated an increased abundance of LRRC8A in plasma membrane-enriched fractions of NAA60-deficient cells, we acknowledge that the biochemical enrichment approach used may co-isolate membrane-bound compartments such as the endoplasmic reticulum. Therefore, some degree of intracellular membrane contamination cannot be excluded.”

Comment 31, concerning Fig. 6:

While the authors have added a precautionary sentence to the manuscript, their claims made in figure headings, abstract, title of the manuscript etc are unchanged and as explained we do not find the evidence concerning Nt-acetylation to be sufficient. Taken together, the data on Nt-acetylation (Figure 3 and Figure 6) all have flaws/weaknesses, making it overall insufficient to conclude that the Pt drug uptake effect observed in the absence of NAA60 is explained by loss of Nt-acetylation.

Answer:

We sincerely appreciate the reviewer’s careful evaluation of our work and their emphasis on ensuring that our conclusions are well supported by the available data. We understand and acknowledge the reviewer’s concerns regarding the strength of the evidence linking NAA60’s role in platinum drug uptake specifically to its N-terminal acetylation activity. In response to this concern, as specified above, we have revised multiple key sections of the manuscript to avoid overstatements and ensure that our claims more accurately reflect the limitations of our data.

R1: Concerning Fig. 6 there is no change in data and only very minor changes in the text, meaning that overall there is not much improvement to ensure that the conclusions are well supported by the available data. Concerning Fig. 3, the in vitro Nt-acetylation assay (3a) does not provide any conclusive data, however the IP-MS and IP-gel cutting-MS may be used as evidence indicating that LRRC8A and LRRC8D are less Nt-acetylated in NAA60 KO cells.

Answer 2: We thank the reviewer for their continued critical and constructive feedback. We fully acknowledge the importance of ensuring that the conclusions in the manuscript are firmly supported by the data. Regarding Figure 6, we appreciate that the data itself remains unchanged. However, in response to the reviewer’s concerns, we have carefully revised the title, abstract, results, figure legends, and discussion to ensure that all claims concerning the role of NAA60’s N-terminal acetylation activity are phrased more cautiously and reflect the suggestive nature of the evidence. Together, these revisions reflect our careful consideration of the reviewer’s comments.

Reviewers' response to the author's 2nd rebuttal letter:

Reviewer #1 (Remarks to the Author):

Overall, the authors have come back with several good answers to our concerns. And our suggestions for minor improvements have been followed. However, some of the more severe comments on data quality was not met. While the authors present solid evidence that NAA60 is necessary for effective platinum drug uptake, we still (as no additional experiments were performed) have concerns about the data used to claim that the effect is mediated via NAA60's function as an Nt-acetyl transferase. Although it may appear obvious that lack of Nt-acetylation must indeed be at play, it is important to acknowledge that for many of the NATs moonlighting functions are being revealed. Therefore, weak evidence is not enough to claim that the Pt drug effect is dependent on NAA60's Nt-acetylation.

Answer: We sincerely thank the reviewer for their detailed and constructive feedback. We acknowledge the importance of carefully addressing the mechanistic role of NAA60 in platinum drug uptake and its attribution to NAA60's function as an N-terminal acetyltransferase. While we provided indirect but strong evidence that NAA60 mediates its effects via Nt acetylation of LRRC8A and LRRC8D subunits, we agree that additional experiments to further substantiate this claim would strengthen the conclusions. Unfortunately, our attempts to establish a more sensitive assay to directly measure Nt acetylation activity or improve the sensitivity of our current assays were not successful. Thus, we have taken steps to adjust the manuscript to ensure our claims reflect the available data more accurately and address your concerns. In particular, we have changed the following:

- Title: We have removed the phrase "by N-terminal acetylation" from the title to avoid overstatement of this specific mechanism.
- **R1: OK.**
- Abstract: In the abstract, we have replaced "We demonstrate" with "Our data strongly suggest that" in reference to the role of NAA60's Nt acetylation in platinum drug uptake. This better reflects the level of evidence presented in our study and acknowledges that alternative mechanisms, including potential moonlighting functions, cannot be excluded at this stage.
- **R1: "We demonstrate" exchanged to "Our data strongly suggest" is a very minor change. The sum of the conclusions/statements still tells an overall story that is stretched somewhat too far beyond what is backed by data. For example, the word "modulate" used in the Abstract is an overstatement since no active regulation by NAA60 was shown, simply an effect of its absence/knockout.**
- Results: In line 135, we have changed the subtitle "LRRC8A and LRRC8D N-termini are a substrate for NAA60" to "Investigation of LRRC8A and LRRC8D N-termini as a substrate for NAA60".
- **R1: OK.**
- Discussion: In line 251/252 we have replaced "... we show that this is due to Nt acetylation of the LRRC8A and LRRC8D subunits of VRAC" by "... our data strongly suggest that this is due to Nt acetylation of the LRRC8A and LRRC8D subunits of VRAC. "
- **R1: Very minor change, that does not really meet the concern.**
- Moreover, in the revised discussion, we explicitly acknowledge the limitations of our current assay: In lines 281-283, we added: "To measure NAA60-mediated Nt-acetylated versus un-Nt-

acetylated peptides in our *in vitro* assay, we used mass spectrometry. As these values are low, further corroboration of our data using radioactively labelled Ac-CoA as a readout may be useful.”

- R1: As explained below we do not find this assay to be able to report anything of value. We acknowledge that Nt-acetylation assays may be difficult to set up and that the authors made a robust effort, albeit it was unfortunately not successful. If the authors would like to include Nt-acetylation assay in their paper we suggest that they use one of the published assays. Although not ideal, Fig 3c and d does however provide evidence of LRRC8A and LRRC8D IPed from NAA60 KO cells to be less Nt-acetylated compared to IPs from WT cells. Which could justify balanced suggestive statements about the direct involvement of Nt-acetylation

We hope that the reviewer agrees that these modifications sufficiently balance our findings with their limitations.

Specifically:

Comment 18 on Fig. 3a:

The values are VERY low. What are the raw intensity data here? Raw data was not shared as requested. We would like to see the numbers before the relative normalization. Data in 3a is not significant and hence does not show increased Nt-acetylation in the presence of Naa60. It is not possible to make any claims/statements/conclusions about Nt-acetylation status based on these experiments.

Answer:

We appreciate the reviewer’s request for the raw intensity data and acknowledge the need for transparency in data presentation. In response, we have now included two Excel files (Fig3a_NAA60_in_vitro_acetylation_T7 and Fig3a_NAA60_in_vitro_acetylation_R7) containing the raw spectral counts and intensity data from the T7 and R7 experiments:

- T7 Experiment: A trial experiment conducted to ensure that the experimental setup was functional before proceeding with full measurements.
- R7 Experiment: Three biological replicates (R7.1-3) per peptide and enzyme incubation condition to validate the findings.
- R1: These changes makes sense.

To further address the reviewer’s concerns, Figure 1 in Revision 2 now displays both raw intensity values (top row) and spectral counts (bottom row) per peptide. These graphs can be included as a supplementary figure, if deemed necessary, to allow for a more thorough evaluation of the results.

Regarding potential confounding factors, we observed additional modifications at the initiator methionine site, including:

- Methionine removal

- Methylation (+15.9949 Da)
- Suspected bead crosslinking reagent component (+113.9953 Da), potentially introduced during NAA60 enzyme isolation via HA-tagged magnetic beads.

These modifications may have competed with the N-terminal acetylation reaction, reducing the availability of non-acetylated substrate peptides. To account for this:

- In T7 samples, we excluded these modifications from relative abundance calculations.
- In R7 samples, methylated peptides were initially included in the relative abundance calculations but have now been excluded to ensure consistency in the updated Figure 3a and source data file.

Finally, we have now added the full raw data to the source data file for Figure 3, as requested. We hope that these clarifications, along with the newly provided data, will sufficiently address the reviewer's concerns.

R1: We thank the authors for the additional data material provided. In files Fig3a_NAA60_in_vitro_acetylation_T7 and Fig3a_NAA60_in_vitro_acetylation_R7 (identified as files named: 24920_2_data_set_865470_srb3pb and 24920_2_data_set_865471_srm3pb), we were however unable to identify sample identities (+ vs – NAA60). The Figure 1 in the rebuttal letter is however relatively informative. We now understand a bit more about how these data were processed. However, the authors did not explain the rationale for their normalization strategy, and it seems like the values in Fig 1 in the rebuttal does not add up to what is shown in Fig 3a in the revised manuscript. Thus, it is not transparent how that data were normalized/transformed from raw values. There seems to be too much of the Non-Nt-Ac peptides and these numbers do not decrease when adding NAA60. Another concern, as revealed by Fig 1b there is quite the response in Nt-acetylation by addition of NAA60 enzyme to the negative control peptide SESSKS. Moreover, no statistical testing was performed for these experiments, and it is likely that the potential modest differences observed are not statistically significant, hence the concern stated in the previous revision stands, as it is not possible to make any claims/statements/conclusions about Nt-acetylation status based on these experiments”.

Figure 1 for Review 2 Raw peptide intensity values (a) and spectral counts (b) for each peptide.

Comment 19 on Fig. 3c:

We had concerns and asked to see raw data. This could be an excel file of the identified peptides and their intensities.

Answer: A excel file with the raw intensity data of this experiment is provided with the revised manuscript.

Fig3c_20220222_IP_LRRC8A

Fig3c_20220704_IP_LRRC8A

R1: Although this setup could have preferentially been optimized, the experiments and reported results are acceptable.

Comments 20-21 on Fig. 3d:

Raw data available? The y axis label should be explained. Why were not raw intensities used like in c? While more methodological details for the MS was added to the manuscript, info about data normalization was not added. It should be transparent how that data were normalized/transformed from raw values.

Answer:

We appreciate the reviewer's request. In response, we have made the following clarifications and revisions:

Availability of raw and normalized data:

- The raw MaxQuant intensities (pre-normalization) are now provided in the file Fig3d_20210511_MQ_evidence.

- The variance-normalized data using the evidenceWnorm approach are provided in Fig3d_20210511_evidenceWnorm.
- To facilitate interpretation, peptides included in the analysis are highlighted in yellow and the table has been sorted accordingly.

Explanation of data normalization:

- The evidenceWnorm values were obtained using variance normalization based on intensities detected by MaxQuant.
- This normalization method adjusts for intensity-dependent variance to make comparisons more robust and interpretable.
- The PM1 label represents samples from the parental line, while PM2 corresponds to samples from the Naa60_12A6 line.

Clarification of the Y-axis label in Fig. 3d:

We have revised the Y-axis label for clarity and will ensure that the description in the figure legend explicitly states what is being represented.

Methods section update:

To enhance transparency, we have added the following sentence to the methods section (Lines 619-621):

“For the LRRC8D peptide, the intensities detected by MaxQuant were variance normalized to obtain the evidenceWnorm values shown in Figure 3d.”

We hope these revisions provide the necessary clarity and transparency regarding the raw data, normalization approach, and figure labeling. Thank you for your valuable feedback.

R1: We thank the authors for the additional data and information about normalization. We have no remaining issues to address concerning this. Although this setup could have preferentially been optimized, the experiments and reported results are acceptable.

Comment 23 on Fig 3e and Ext data fig. 5, the PLA experiments: The text should convey a bit more caution when interpreting the results and be open about possible variations due to expression levels and so on with transfection.

Answer: The following cautionary sentences were added to the main text of the manuscript:

“The number and localization pattern of the foci may depend on the expression levels of the individual proteins. The higher dispersion levels seen in some of the images could be an artefact of NAA60 overexpression.” – Line 173-174

R1: OK.

Comment 30 concerning Fig. 5f:

While it sounds like the authors have used the CellMask stain correct, our comment concerning the difficulties in interpreting these results was not responded to. We have a hard time accepting this as

evidence for PM localization and it is definitely not possible to conclude with increased PM localization in the absence of Naa60. We understand that microscopy of PM localization is difficult, however the current data does not demonstrate what the authors claim it to demonstrate. We see that a 100 x objective was used and that samples were imaged in Z-stacks of 20 slices of 0.2 μ m thickness. This should allow for a 3D analysis in order to measure plasma membrane localization rather than simply showing the images as maximum projection. The images shown do not support the claim.

Answer:

We appreciate the reviewer's concerns regarding the interpretation of our PM localization data and the challenges in quantifying membrane colocalization using microscopy. We would like to clarify that the aim of this experiment was not to claim an increase in LRRC8A PM localization in the absence of NAA60, but rather to demonstrate that LRRC8A is still present at the plasma membrane despite the loss of NAA60.

We acknowledge that a more quantitative approach would require a proper negative control, such as a solely cytosolic GFP construct, for direct comparison. Unfortunately, we did not acquire such a control in the original experiment, which is why we have refrained from performing a full quantification of PM localization. To enhance the interpretability of the existing data, we have now:

- Used the "Plot Profile" function in ImageJ to generate linear intensity profiles for the GFP and CellMask channels across the plasma membrane.
- Added these intensity profiles to Figure 5g to better illustrate the observed colocalization pattern.
- Updated the figure legend for Figure 5g to describe the new intensity profile analysis.
- Expanded the methods section to detail the approach used for this visualization.

While we acknowledge the reviewer's suggestion for a 3D analysis using Z-stacks, the lack of a cytosolic control limits our ability to perform quantitative measurements of PM enrichment. Nonetheless, we believe that the intensity profile visualization now provides additional context for interpreting the colocalization pattern and further supports our observation that LRRC8A is not completely absent from the plasma membrane in the absence of NAA60.

We hope this clarification and the additional analysis address the reviewer's concerns. Thank you for your valuable feedback.

R1: The microscopy has improved somewhat by the additional graph visualization examples, and Fig. 5g is OK to demonstrate that LRRC8A is still present at the plasma membrane despite the loss of NAA60. Increase in LRRC8A PM localization in the absence of NAA60, is still one of the conclusions in the text. Then this is based on the MS data alone. From the Results text: "We therefore analyzed plasma membrane-enriched protein fractions of wild-type and *Naa60* knockout cell lines using mass spectrometry. Surprisingly, the analysis revealed that rather than absent, the subunit LRRC8A was instead significantly enriched at the plasma membrane of the two NAA60-deficient cell lines *Naa60_12A6* and *Naa60_13C3*, when compared to the parental cell line (Fig.5e+f)." This means that the quality assessment of the PM isolate is crucial. We therefore revisited the previous rebuttal letter where the authors provided additional supportive data suggested for the supplemental:

Figure 2 for the reviewers shows the enrichment for selected plasma membrane proteins using the plasma membrane enrichment protocol in comparison to when whole cell lysate was used for the analysis.

R1: We noticed that this graph says TM proteins, not PM proteins, so this should be clarified. In order for successful isolation of PM to be validated, it will be necessary to demonstrate that PM proteins are in general enriched, and importantly that the contamination from ER is minor. This could be possible to assess better in the existing dataset and present in supplemental item(s). If the isolate is enriched also in ER proteins, then it will not be possible to claim that loss of NAA60 cause increase in PM localization.

Comment 31, concerning Fig. 6:

While the authors have added a precautionary sentence to the manuscript, their claims made in figure headings, abstract, title of the manuscript etc are unchanged and as explained we do not find the evidence concerning Nt-acetylation to be sufficient. Taken together, the data on Nt-acetylation (Figure 3 and Figure 6) all have flaws/weaknesses, making it overall insufficient to conclude that the Pt drug uptake effect observed in the absence of NAA60 is explained by loss of Nt-acetylation.

Answer:

We sincerely appreciate the reviewer's careful evaluation of our work and their emphasis on ensuring that our conclusions are well supported by the available data. We understand and acknowledge the reviewer's concerns regarding the strength of the evidence linking NAA60's role in platinum drug uptake specifically to its N-terminal acetylation activity. In response to this concern, as specified above, we have revised multiple key sections of the manuscript to avoid overstatements and ensure that our claims more accurately reflect the limitations of our data.

R1: Concerning Fig. 6 there is no change in data and only very minor changes in the text, meaning that overall there is not much improvement to ensure that the conclusions are well supported by the available data.

Concerning Fig. 3, the in vitro Nt-acetylation assay (3a) does not provide any conclusive data, however the IP-MS and IP-gel cutting-MS may be used as evidence indicating that LRRC8A and LRRC8D are less Nt-acetylated in NAA60 KO cells.

Reviewer #2 (Remarks to the Author):

The authors did an excellent job in addressing the extensive comments of the reviewers. Although my comments and suggestions did not result in any changes in the revised manuscript, I find the manuscript ready to be published. Publication of the reviewer comments and the author rebuttal letters in a supplementary file is strongly recommended.

Answer: Thank you very much.